# Sample-Efficient Multi-Agent RL: An Optimization Perspective

**Nuoya Xiong**[*]
IIIS, Tsinghua University
xiongny20@mails.tsinghua.edu.cn

**Zhihan Liu**[*]
Northwestern University
zhihanliu2027@u.northwestern.edu

**Zhaoran Wang**
Northwestern University
zhaoranwang@gmail.com

**Zhuoran Yang**
Yale University
zhuoran.yang@yale.edu

## Abstract

We study multi-agent reinforcement learning (MARL) for the general-sum Markov Games (MGs) under general function approximation. In order to find the minimum assumption for sample-efficient learning, we introduce a novel complexity measure called the Multi-Agent Decoupling Coefficient (MADC) for general-sum MGs. Using this measure, we propose the first unified algorithmic framework that ensures sample efficiency in learning Nash Equilibrium, Coarse Correlated Equilibrium, and Correlated Equilibrium for both model-based and model-free MARL problems with low MADC. We also show that our algorithm provides comparable sublinear regret to the existing works. Moreover, our algorithm only requires an equilibrium-solving oracle and an oracle that solves regularized supervised learning, and thus avoids solving constrained optimization problems within data-dependent constraints (Jin et al., 2020a; Wang et al., 2023) or executing sampling procedures with complex multi-objective optimization problems (Foster et al., 2023). Moreover, the model-free version of our algorithms is the first provably efficient model-free algorithm for learning Nash equilibrium of general-sum MGs.

## 1 Introduction

Multi-agent reinforcement learning (MARL) has achieved remarkable empirical successes in solving complicated games involving sequential and strategic decision-making across multiple agents (Vinyals et al., 2019; Brown & Sandholm, 2018; Silver et al., 2016). These achievements have catalyzed many research efforts focusing on developing efficient MARL algorithms in a theoretically principled manner. Specifically, a multi-agent system is typically modeled as a general-sum Markov Game (MG) (Littman, 1994), with the primary aim of efficiently discerning a certain equilibrium notion among multiple agents from data collected via online interactions. Some popular equilibrium notions include Nash equilibrium (NE), correlated equilibrium (CE), and coarse correlated equilibrium (CCE).

However, multi-agent general-sum Markov Games (MGs) bring forth various challenges. In particular, empirical application suffers from the large state space. Such a challenge necessitates the use of the function approximation as an effective way to extract the essential features of RL problems and avoid dealing directly with the large state space. Yet, adopting function approximation in a general-sum MG brings about additional complexities not found in single-agent RL or a zero-sum MG. Many prevailing studies on single-agent RL or two-agent zero-sum MGs with the function approximation leverage the special relationships between the optimal policy and the optimal value function (Jin et al., 2021a; Du et al., 2021; Zhong et al., 2022; Jin et al., 2022; Huang et al., 2021; Xiong et al., 2022). In particular, in single-agent RL, the optimal policy is the greedy policy with respect to the optimal value function. Whereas in a two-agent zero-sum MG, the Nash equilibrium

---
[*]Equal contribution

is obtained by solving a minimax estimation problem based on the optimal value function. Contrastingly, in a general-sum MG, individual agents possess distinct value functions, and thus there exists no unified optimal value function that characterizes the equilibrium behavior. Moreover, unlike a zero-sum MG, a general-sum MG can admit diverse equilibrium notions, where each corresponds to a set of policies. Consequently, methodologies developed for single-agent RL or zero-sum MGs cannot be directly extended to general-sum MGs.

Recently, several works propose sample-efficient RL algorithms for general-sum MGs. In particular, Chen et al. (2022b); Foster et al. (2023) propose model-based algorithms for learning NE/CCE/CE based on multi-agent extensions of the Estimation-to-Decision algorithm (Foster et al., 2021), and they establish regret upper bounds in terms of complexity metrics that extend Decision-Estimation Coefficient (Foster et al., 2021) to MGs. In addition, Wang et al. (2023) study model-free RL for general-sum MGs with the general function approximation. They focus on developing a decentralized and no-regret algorithm that finds a CCE. Thus, it seems unclear how to design a provably sample-efficient MARL algorithm for NE/CCE/CE for general-sum MGs in a *model-free* manner. Furthermore, motivated by the recent development in single-agent RL (Jin et al., 2021a; Du et al., 2021; Zhong et al., 2022; Foster et al., 2021; Liu et al., 2023), we aim to develop a unified algorithmic framework for MARL that covers both model-free and model-based approaches. Thus, we aim to address the following questions:

> Can we design a unified algorithmic framework for general-sum MGs such that (i) it is provably sample-efficient in learning NE/CCE/CE in the context of the function approximation and (ii) it covers both model-free and model-based MARL approaches?

In this paper, we provide an affirmative answer to the above questions. Specifically, we propose a unified algorithmic framework named **M**ulti-**A**gent **M**aximize-to-**EX**plore (MAMEX) for general-sum MGs with the general function approximation. MAMEX extends the framework of Maximize-to-Explore (Liu et al., 2023) to general-sum MGs by employing it together with an equilibrium solver for general-sum normal-form games defined over the policy space.

Maximize-to-Explore (MEX) is a class of RL algorithms for single-agent MDP and two-agent zero-sum MGs where each new policy is updated by solving an optimization problem involving a hypothesis $f$, which can be regarded as the action-value function in the model-free version and the transition model in the model-based version. The optimization objective of MEX contains two terms — (a) the optimal value with respect to the hypothesis $f$ and (b) a loss function computed from data that quantifies how far $f$ is from being the true hypothesis. Here, the term (a) reflects the planning part of online RL and leverages the fact that the optimal policy is uniquely characterized by the given hypothesis. On the other hand, the term (b), which can be the mean-squared Bellman error or log-likelihood function, reflects the estimation part of online RL. By optimizing the sum of (a) and (b) over the space of hypotheses *without any data-dependent constraints*, MEX balances exploitation with exploration in the context of the function approximation.

However, the first term in MEX's optimization objective leverages the fact that the optimal policy can be uniquely constructed from the optimal value function or the true model, using a greedy step or dynamic programming. Such a nice property cannot be extended to general-sum MGs, where the relationship between the equilibrium policies and value function is more complicated, and each agent has its own value function. As a result, it is impractical to construct a single-objective optimization problem in the style of MEX over the hypothesis space for general-sum MGs.

Instead of optimizing over the spaces of hypotheses, MAMEX optimizes over the policy space. Specifically, in each iteration, MAMEX updates the joint policy of all agents by solving for a desired equilibrium (NE/CCE/CE) of a *normal-form game*, where the pure strategies are a class of joint policies of the $n$ agents, e.g., the class of deterministic joint policies. Besides, for each pure strategy of this normal form game, the corresponding payoff function is obtained by solving a regularized optimization problem over the hypothesis space à la MEX. Thus, policy updates in MAMEX involve the following two steps:

(i) For each pure strategy $\pi$, construct the payoff function $\overline{V}_i(\pi)$ for each agent $i$ by solving an unconstrained and regularized optimization problem;

(ii) Compute the NE/CCE/CE of the normal-form game over the space of pure strategies with payoff functions $\{\overline{V}_i(\pi)\}_{i=1}^n$, where $n$ is the number of agents.

The implementation of MAMEX only requires an oracle for solving a single-objective and unconstrained optimization problem and an oracle for solving NE/CCE/CE of a normal-form game. Compared to existing works that either solve constrained optimization subproblems within data-dependent constraints (Wang et al., 2023), or complex multi-objective or minimax optimization subproblems (Foster et al., 2023; Chen et al., 2022b), MAMEX is more amenable to practical implementations. Furthermore, step (i) of MAMEX resembles MEX, which enables both model-free and model-based instantiations.

We prove that MAMEX is provably sample-efficient in a rich class of general-sum MGs. To this end, we introduce a novel complexity measure named **M**ulti-**A**gent **D**ecoupling **C**oefficient (MADC) to capture the exploration-exploitation tradeoff in MARL. Compared to the decoupling coefficient and its variants (Dann et al., 2021; Agarwal & Zhang, 2022; Zhong et al., 2022) proposed for the single-agent setting, MADC characterize the hardness of exploration in MGs in terms of the discrepancy between the out-of-sample *prediction error* and the in-sample *training error* incurred by minimizing a discrepancy function $\ell$ on the historical data. MADC is defined based on the intuition that if a hypothesis attains a small training error on a well-explored dataset, it would also incur a small prediction error. When the MADC of an MG instance is small, achieving a small training error ensures a small prediction error, and thus exploration is relatively easy. We prove that MAMEX achieves a sublinear regret for learning NE/CCE/CE in classes with small MADCs, which includes multi-agent counterparts of models with low Bellman eluder dimensions (Jin et al., 2021a; 2022; Huang et al., 2021), Bilinear Classes (Du et al., 2021), and models with low witness ranks (Sun et al., 2019; Huang et al., 2021). When specialized to specific members within these classes, MAMEX yields comparable regret upper bounds to existing works.

**Our Contributions.** In summary, our contributions are two-fold.

• First, we provide a unified algorithmic framework named **M**ulti-**A**gent **M**aximize-to-**EX**plore (MAMEX) for both model-free and model-based MARL, which is sample-efficient in finding the NE/CCE/CE in general-sum MGs with small MADCs. Moreover, MAMEX leverages an equilibrium-solving oracle for normal-form games defined over a class of joint policies for policy updates, and a single-objective optimization procedure that solves for the payoff functions of these normal-form games. To our best knowledge, the model-free version of MAMEX is the first model-free algorithm for general-sum MGs that learns all three equilibria NE, CCE, and CE with sample efficiency.

• Second, we introduce a complexity measure, **M**ulti-**A**gent **D**ecoupling **C**oefficient (MADC), to quantify the hardness of exploration in a general-sum MG in the context of the function approximation. The class of MGs with low MADCs includes a rich class of MG instances, such as multi-agent counterparts of models with low Bellman eluder dimensions (Jin et al., 2021a; 2022; Huang et al., 2021), Bilinear Classes (Du et al., 2021), and models with low witness ranks (Sun et al., 2019; Huang et al., 2021). When specialized to specific MG instances in these classes, we achieve comparable regret upper bounds to existing works.

**Related Works.** Our paper is closely related to the prior research on Markov Games and MARL with the function approximation. A comprehensive summary of the related literature is in §A.

## 2 MODELS AND PRELIMINARIES

### 2.1 MARKOV GAMES

For clarity, certain mathematical notations are provided in Appendix §B.

**General-Sum Markov Games** In this work, we consider general-sum Markov Games (MGs) in the episodic setting, which is denoted by a tuple $(\mathcal{S}, H, \mathcal{A}, \{r_h^{(i)}\}_{i \in [n], h \in [H]}, \{\mathbb{P}_h\}_{h \in [H]}, \rho)$, where $n$ is the number of agents, $H$ is the length of one episode, $\mathcal{S}$ is the state set, and $\mathcal{A} = \otimes_{i=1}^{n} \mathcal{A}_i$ is the joint action set. Here, $\mathcal{A}_i$ is the action set of the agent $i$. Moreover, $r_h^{(i)} : \mathcal{S} \times \mathcal{A} \mapsto \mathbb{R}$ is the known reward function[1] of the agent $i$ at step $h$, $\mathbb{P}_h : \mathcal{S} \times \mathcal{A} \to \Delta(\mathcal{S})$ is the transition kernel at the $h$-th step, and $\rho \in \Delta(\mathcal{S})$ is the distribution of the initial state $s_1$. We assume the $n$ agents

---

[1]Our results can be extended to the unknown stochastic reward case (Agarwal & Zhang, 2022; Zhong et al., 2022). Note that learning the transition kernel is more difficult than learning the reward.

observe the same state at each step and each agent $i$ chooses an action within its own action set $\mathcal{A}_i$ simultaneously. In each episode, starting from $s_1 \sim p_0$, for each $h \in [H]$, the agents choose their joint action $a_h \in \mathcal{A}$ in state $s_h$, where $a_h = (a_h^{(1)}, \ldots, a_h^{(n)})$. Then, each agent $i$ receives its own reward $r_h^{(i)}(s_h, a_h)$, and the game move to the next state $s_{h+1} \sim \mathbb{P}_h(s_{h+1} \mid s_h, a_h)$. Moreover, we assume $\sum_{h=1}^{H} r_h^{(i)}(s_h, a_h) \in [0, R]$ for any possible state-action sequences for some $1 \le R \le H$.

In MGs, the agents' policy can be stochastic and correlated. To capture such a property, we introduce the notion of *pure policy* and *joint policy* as follows. For each agent $i$, its local (Markov) policy maps a state $s$ to a distribution over the local action space $\mathcal{A}_i$. We let $\Pi_{h,i}^{\mathrm{pur}} \subseteq \{\pi : \mathcal{S} \mapsto \Delta(\mathcal{A}_i)\}$ denote a subset of the agent $i$'s local policies for step $h \in [H]$, which is called the set of Markov *pure policies*. We then define $\Pi_i^{\mathrm{pur}} = \cup_{h \in [H]} \Pi_{h,i}^{\mathrm{pur}}$. We assume the agent $i$'s policy is a random variable taking values in $\Pi_i^{\mathrm{pur}}$. Specifically, let $\omega \in \Omega$ be the random seed. The *random* policy $\pi^{(i)} = \{\pi_h^{(i)}\}_{h \in [H]}$ for the agent $i$ contains $H$ mappings $\pi_h^{(i)} : \Omega \mapsto \Pi_{h,i}^{\mathrm{pur}}$ such that $\pi_h^{(i)}(\omega) \in \Pi_{h,i}^{\mathrm{pur}}$ is a pure policy. To execute $\pi^{(i)}$, the agent $i$ first samples a random seed $\omega \in \Omega$, and then follows the policy $\pi_h^{(i)}(\omega)$ for all $h \in [H]$. The *joint policy* $\pi$ of the $n$ agents is a set of policies $\{\pi^{(i)}\}_{i=1}^{n}$ where $\omega = (\omega_1, \cdots, \omega_n)$ are joint variables. In other words, $\{\pi_h^{(i)}(\omega)\}_{i \in [n]} \in \otimes_{i=1}^{n} \Pi_{h,i}^{\mathrm{pur}}$ are random policies of the $n$ agents whose randomness is correlated by the random seed $\omega$. Equivalently, we can regard $\pi$ as a random variable over $\otimes_{i=1}^{n} \Pi_{h,i}^{\mathrm{pur}}$. Furthermore, a special class of the joint policy is the *product policy*, where each agent executes their own policies independently. In other words, we have $\omega = (\omega_1, \ldots, \omega_n)$, where $\omega_1, \ldots, \omega_n$ are independent, and each $\pi^{(i)}$ depends on $\omega_i$ only. We let $\pi_h(a \mid s)$ denote the probability of taking action $a$ in the state $s$ at step $h$. As a result, we have $\pi_h(a \mid s) = \prod_{i=1}^{n} \pi_h^{(i)}(a^{(i)} \mid s)$ for any product policy $\pi$.

Furthermore, using the notion of pure policy and joint policy, we can equivalently view the MG as a normal form game over $\Pi^{\mathrm{pur}} = \otimes_{i=1}^{n} \Pi_i^{\mathrm{pur}}$. That is, each pure policy can be viewed as a pure strategy of the normal form game, and each joint policy can be viewed as a mixed strategy. Such a view is without loss of generality, because we can choose $\Pi_i^{\mathrm{pur}}$ to be the set of all possible deterministic policies of the agent $i$. Meanwhile, using a general $\Pi_i^{\mathrm{pur}}$, we can also incorporate parametric policies as the pure policies, e.g., log-linear policies (Xie et al., 2021; Yuan et al., 2022; Cayci et al., 2021).

The value function $V_h^{(i),\pi}$ is the expected cumulative rewards received by the agent $i$ from step $h$ to step $H$, when all the agents follow a joint policy $\pi$, which is defined as

$$V_h^{(i),\pi}(s) = \mathbb{E}_\pi \Big[ \sum_{h'=h}^{H} r_{h'}^{(i)}(s_{h'}, a_{h'}) \Big| s_h = s \Big].$$

We let $V^{(i),\pi}(\rho) = \mathbb{E}_{s \sim \rho}[V_1^{(i),\pi}(s)]$ denote the agent $i$'s expected cumulative rewards within the whole episode. Besides, the corresponding $Q$-function (action-value function) can be written as

$$Q_h^{(i),\pi}(s, a) = \mathbb{E}_\pi \Big[ \sum_{h'=h}^{H} r_{h'}^{(i)}(s_{h'}, a_{h'}) \Big| s_h = s, a_h = a \Big]. \tag{2.1}$$

For a joint policy $\pi$ and any agent $i$, we let $\pi^{(-i)}$ denote the joint policy excluding the agent $i$. Given $\pi^{(-i)}$, the *best response* of the agent $i$ is defined as $\pi^{(i),\dagger} = \arg\max_{\nu \in \Delta(\Pi_i^{\mathrm{pur}})} V^{(i),\nu \times \pi^{(-i)}}(\rho)$, which is random policy of the agent $i$ that maximizes its expected rewards when other agents follow $\pi^{(-i)}$. Besides, we denote $\mu^{(i),\pi} = (\pi^{(i),\dagger}, \pi^{(-i)})$.

**Online Learning and Solution Concepts** We focus on three common equilibrium notions in the game theory: Nash Equilibrium (NE), Coarse Correlated Equilibrium (CCE) and Correlated Equilibrium (CE).

First, a NE of a game is a *product* policy that no individual player can improve its expected cumulative rewards by unilaterally deviating its local policy.

**Definition 2.1** ($\varepsilon$-Nash Equilibrium)**.** A *product policy* $\pi$ is an $\varepsilon$-Nash Equilibrium if $V^{(i),\mu^{(i),\pi}}(\rho) \le V^{(i),\pi}(\rho) + \varepsilon$ for all $i \in [n]$, where $\mu^{(i),\pi} = (\pi^{(i),\dagger}, \pi^{(-i)})$ and $\pi^{(i),\dagger}$ is the best response policy with respect to $\pi^{(-i)}$.

In other words, a product policy $\pi$ is an $\varepsilon$-Nash Equilibrium if and only if

$$\max_{i \in [n]} \left\{ \max_{\nu \in \Delta(\Pi_i^{\mathrm{pur}})} V^{(i), \nu \times \pi^{(-i)}}(\rho) - V^{(i), \pi}(\rho) \right\} \leq \varepsilon.$$

In this work, we design algorithms for the online and self-play setting. That is, we control the joint policy all agents, interact with the environment over $K$ episodes, and aim to learn the desired equilibrium notion from bandit feedbacks. To this end, let $\pi^k$ denote the joint policy that the agents execute in the $k$-th episode, $k \in [K]$. We define the Nash-regret as the cumulative suboptimality across all agents with respect to NE.

**Definition 2.2** (Nash-Regret). For all $k \in [K]$, let $\pi^k$ denote the *product policy* deployed in the $k$-th episode, then the Nash-regret is defined as

$$\mathrm{Reg}_{\mathrm{NE}}(K) = \sum_{k=1}^{K} \sum_{i=1}^{n} \left( V^{(i), \mu^{(i), \pi^k}}(\rho) - V^{(i), \pi^k}(\rho) \right).$$

By replacing the concept of NE to CCE and CE, we can define CCE-regret and CE-regret in a similar way. The detailed definitions are provided in §C.

We note that the definitions of NE, CCE, and CE align with those defined on the normal form game defined on the space of pure policies. That is, each agent $i$'s "pure strategy" is a pure policy $\pi^{(i)} \in \Pi_i^{\mathrm{pur}}$, and the "payoff" of the agent $i$ when the "mixed strategy" is $\pi$ is given by $V^{(i), \pi}(\rho)$.

## 2.2 FUNCTION APPROXIMATION

To handle the large state space in MARL, we assume the access to a hypothesis class $\mathcal{F}$, which captures the $Q$ function in the model-free setting and the transition kernel in the model-based setting.

**Model-Based Function Approximation** In the model-based setting, the hypothesis class $\mathcal{F}$ contains the model (transition kernel) of MGs. Specifically, we let $\mathbb{P}_f = \{\mathbb{P}_{1,f} \cdots, \mathbb{P}_{H,f}\}$ denote the transition kernel parameterized by $f \in \mathcal{F}$. When the model parameters are $f$ and the joint policy is $\pi$, we denote the value function and $Q$-function of the agent $i$ at the $h$-th step as $V_{h,f}^{(i), \pi}(s)$ and $Q_{h,f}^{(i), \pi}(s, a)$ respectively. We have the Bellman equation $Q_{h,f}^{(i), \pi}(s, a) = r_h^{(i)}(s, a) + \mathbb{E}_{s' \sim \mathbb{P}_{h,f}(\cdot | s, a)}[V_{h+1,f}^{(i), \pi}(s')]$.

**Model-Free Function Approximation** In the model-free setting, we let $\mathcal{F} = \otimes_{i=1}^{n} \mathcal{F}^{(i)} = \otimes_{i=1}^{n}(\otimes_{h=1}^{H} \mathcal{F}_h^{(i)})$ be a class of $Q$-functions of the $n$ agents, where $\mathcal{F}_h^{(i)} = \{f_h^{(i)} : \mathcal{S} \times \mathcal{A} \mapsto \mathbb{R}\}$ is a class of $Q$-functions of the agent $i$ at the $h$-th step. For any $f \in \mathcal{F}$, we denote $Q_{h,f}^{(i)}(s, a) = f_h^{(i)}(s, a)$ for all $i \in [n]$ and $h \in [H]$. Meanwhile, for any joint policy $\pi$ and any $f \in \mathcal{F}$, we define

$$V_{h,f}^{(i), \pi}(s) = \mathbb{E}_{a \sim \pi(s)}[f_h^{(i)}(s, a)] = \langle f_h^{(i)}(s, \cdot), \pi_h(\cdot \mid s) \rangle_{\mathcal{A}}.$$

For any joint policy $\pi$, agent $i$, and step $h$, we define the Bellman operator $\mathcal{T}_h^{(i), \pi}$ by letting

$$(\mathcal{T}_h^{(i), \pi}(f_{h+1}))(s, a) = r_h^{(i)}(s, a) + \mathbb{E}_{s' \sim \mathbb{P}_h(s' | s, a)} \langle f_{h+1}(s', \cdot), \pi_{h+1}(\cdot \mid s') \rangle_{\mathcal{A}}, \quad \forall f \in \mathcal{F}^{(i)}. \quad (2.2)$$

Note that the Bellman operator depends on the index $i$ of the agent because the reward functions of the agents are different. Such a definition is an extension of the Bellman evaluation operator in the single-agent setting (Puterman, 2014) to the multi-agent MGs. By definition, $\{Q_h^{(i), \pi}\}$ defined in (2.1) is the fixed point of $\mathcal{T}_h^{(i), \pi}$, i.e., $Q_h^{(i), \pi} = \mathcal{T}_h^{(i), \pi}(Q_{h+1}^{(i), \pi})$ for all $h \in [H]$.

For both the model-based and the model-free settings, we impose the realizability assumption, which requires that the hypothesis space $\mathcal{F}$ is sufficiently expressive such that it contains the true transition model or the true $Q$-functions. Besides, for the model-free setting, we also require that the hypothesis classes be closed with respect to the Bellman operator.

**Assumption 2.3** (Realizability and Completeness). For the model-based setting, we assume the true transition model $f^*$ lies in the hypothesis class $\mathcal{F}$. Besides, for the model-free setting, for any *pure policy* $\pi$ and any $i \in [n]$, we assume that $Q^{(i), \pi} \in \mathcal{F}^{(i)}$ and $\mathcal{T}_h^{(i), \pi} \mathcal{F}_{h+1}^{(i)} \subseteq \mathcal{F}_h^{(i)}$ for all $h \in [H]$.

**Covering Number and Bracketing Number.** When a function class $\mathcal{F}$ is infinite, the $\delta$-covering number $\mathcal{N}_{\mathcal{F}}(\delta)$ and the $\delta$-bracketing number $\mathcal{B}_{\mathcal{F}}(\delta)$ serve as surrogates of the cardinality of $\mathcal{F}$. We put the definitions in §C.2.

**Multi-Agent Decoupling Coefficient** Now we introduce a key complexity measure — multi-agent decoupling coefficient (MADC) — which captures the hardness of exploration in MARL. Such a notion is an extension of the decoupling coefficient (Dann et al., 2021) to general-sum MGs.

**Definition 2.4** (Multi-Agent Decoupling Coefficient). The Multi-Agent Decoupling Coefficient of a MG is defined as the smallest constant $d_{\mathrm{MADC}} \geq 1$ such that for any $i \in [n]$, $\mu > 0$, $\{f^k\}_{k \in [K]} \subseteq \mathcal{F}^{(i)}$, and $\{\pi^k\}_{k \in [K]} \subseteq \Pi^{\mathrm{pur}}$ the following inequality holds:

$$\underbrace{\sum_{k=1}^{K}(V_{f^k}^{(i),\pi^k}(\rho) - V^{(i),\pi^k}(\rho))}_{\text{prediction error}} \leq \frac{1}{\mu}\underbrace{\sum_{k=1}^{K}\sum_{s=1}^{k-1}\ell^{(i),s}(f^k,\pi^k)}_{\text{training error}} + \underbrace{\mu \cdot d_{\mathrm{MADC}} + 6d_{\mathrm{MADC}}H}_{\text{gap}}, \quad (2.3)$$

where we define $V_{f^k}^{(i),\pi^k}(\rho) = \mathbb{E}_{s_1 \sim \rho}[V_{1,f^k}^{(i),\pi^k}(s_1)]$, and $\ell^{(i),s}(f^k,\pi^k)$ is a discrepancy function that measures the inconsistency between $f^k$ and $\pi^k$, on the historical data. The specific definitions of $\{\ell^{(i),s}\}_{i \in [n], s \in [K-1]}$ under the model-free and model-based settings are given in (2.4) and (2.5).

**Model-Free RL** In the model-free setting, for $\{\pi^k\}_{k \in [K]} \subseteq \Pi^{\mathrm{pur}}$ in (2.3), the discrepancy function $\ell^{(i),s}(f,\pi)$ for $\pi \in \Pi^{\mathrm{pur}}$ is defined as

$$\ell^{(i),s}(f,\pi) = \sum_{h=1}^{H}\mathbb{E}_{(s_h,a_h)\sim\pi_h^s}((f_h - \mathcal{T}_h^{(i),\pi}(f_{h+1}))(s_h,a_h))^2, \quad \forall f \in \mathcal{F}^{(i)}, \forall s \in [K]. \quad (2.4)$$

That is, $\ell^{(i),s}(f,\pi)$ measures agent $i$'s mean-squared Bellman error for evaluating $\pi$, when the trajectory is sampled by letting all agents follow policy $\zeta^s$.

**Model-Based RL** We choose the discrepancy function $\ell^{(i),s}$ in Assumption 2.5 as

$$\ell^{(i),s}(f^k,\pi^k) = \sum_{h=1}^{H}\mathbb{E}_{(s_h,a_h)\sim\pi_h^s}D_{\mathrm{H}}^2(\mathbb{P}_{h,f^k}(\cdot \mid s_h,a_h)\|\mathbb{P}_{h,f^*}(\cdot \mid s_h,a_h)), \quad (2.5)$$

where $D_{\mathrm{H}}$ denotes the Hellinger distance and $\mathbb{E}_{(s_h,a_h)\sim\pi_h^s}$ means that the expectation is taken with respect to the randomness of the trajectory induced by $\pi^s$ on the true model $f^*$. Intuitively, it represents the expected in-sample distance of model $f^k$ and true model $f^*$.

Note that the discrepancy between $f^k, \pi^k$ in (2.3) is summed over $s \in [k-1]$. Thus, in both the model-free and model-based settings, the training error can be viewed as the in-sample error of $f^k$ on the historical data collected before the $k$-th episode. Thus, for an MG with a finite MADC, the prediction error is small whenever the training error is small. Specifically, when the training error is $\mathcal{O}(K^\alpha)$ for some $\alpha \in (0,2)$, then by choosing a proper $\mu$, we know that the prediction error grows as $\mathcal{O}(\sqrt{K^\alpha \cdot d_{\mathrm{MADC}}}) = o(K)$. In other words, as $K$ increases, the average prediction error decays to zero. In single-agent RL, when we adopt an optimistic algorithm, the prediction error serves as an upper bound of the regret (Dann et al., 2021; Zhong et al., 2022; Jin et al., 2021a). Therefore, by quantifying how the prediction error is related to the training error, the MADC can be used to characterize the hardness of exploration in MARL.

Compared to the decoupling coefficient and its variants for the single-agent MDP or the two-player zero-sum MG Dann et al. (2021); Agarwal & Zhang (2022); Zhong et al. (2022); Xiong et al. (2022), MADC selects the policy $\pi^k$ in a different way. In the single-agent setting, the policy $\pi^k$ is always selected as the greedy policy of $f^k$, hence $V_{1,f^k}^{\pi^k}(\rho)$ is equivalent to the optimal value function. In the zero-sum MG, the policy pair $\pi^k$ is always selected as the Nash policy and the best response (Xiong et al., 2022). On the contrary, in our definition, the policy $\pi^k$ is not necessarily the greedy policy of $f^k$. In fact, $\{\pi^k\}_{k \in [K]}$ can be any pure policy sequence that is unrelated to $\{f^k\}_{k \in [K]}$.

**Assumption 2.5** (Finite MADC). We assume that the MADC of the general-sum MG of interest is finite, denoted by $d_{\mathrm{MADC}}$. As we will show in Section D, the class of MGs with low MADCs include a rich class of MG instances, including multi-agent counterparts of models with low Bellman eluder dimensions (Jin et al., 2021a; 2022; Huang et al., 2021), bilinear classes (Du et al., 2021), and models with low witness ranks (Sun et al., 2019; Huang et al., 2021).

# 3 ALGORITHM AND RESULTS

In this section, we first introduce a unified algorithmic framework called **M**ulti-**A**gent **M**aximize-to-**EX**plore (MAMEX). Then, we present the regret and sample complexity upper bounds of MAMEX, showing that both the model-free and model-based versions of MAMEX are sample-efficient for learning NE/CCE/CE under the general function approximation.

## 3.1 ALGORITHM

---

**Algorithm 1 M**ulti-**A**gent **M**aximize-to-**EX**plore (MAMEX)

1: **Input:** Hypothesis class $\mathcal{F}$, parameter $\eta > 0$, and an equilibrium solving oracle EQ.
2: **for** $k = 1, 2, \cdots, K$ **do**
3:     Compute $\overline{V}_i^k(\pi)$ defined in (3.1) for all $\pi \in \Pi^{\mathrm{pur}}$ and all $i \in [n]$.
4:     Compute the NE/CCE/CE of the normal-form game defined on $\Pi^{\mathrm{pur}}$ with payoff functions $\{\overline{V}_i^k(\pi)\}_{i=1}^n$: $\pi^k \leftarrow \mathsf{EQ}(\overline{V}_1^k, \overline{V}_2^k, \cdots, \overline{V}_n^k)$.
5:     Sample a pure joint policy $\zeta^k \sim \pi^k$, and collect a trajectory $\{s_h^k, a_h^k\}_{h \in [H]}$ following $\zeta^k$.
6:     Update $\{L^{(i),k}\}_{i=1}^n$ according to (3.2) (model-free) or (3.3) (model-based).
7: **end for**

---

In this subsection, we provide the MAMEX algorithm for multi-agent RL under the general function approximation, which extends the MEX algorithm (Liu et al., 2023) to general-sum MGs. Recall that the definitions of NE/CCE/CE of general-sum MGs coincide with those defined in the normal-form game with pure strategies being the pure policies in $\Pi^{\mathrm{pur}}$. Thus, when we know the payoffs $\{V^{(i),\pi}(\rho)\}_{i\in[n]}$ for all $\pi \in \Pi^{\mathrm{pur}}$, we can directly compute the desired NE/CCE/CE given an equilibrium solving oracle for the normal-form game. However, each $V^{(i),\pi}(\rho)$ is unknown and has to be estimated from data via online learning. Thus, in a nutshell, MAMEX is an iterative algorithm that consists of the following two steps:

(a) Policy evaluation: For each $k \in [K]$, construct an estimator $\overline{V}_i^k(\pi)$ of $V^{(i),\pi}(\rho)$ for each pure policy $\pi \in \Pi^{\mathrm{pur}}$ and the agent $i \in [n]$ in each episode based on the historical data collected in the previous $k-1$ episodes. Here, the policy evaluation subproblem can be solved in both the model-free and model-based fashion.

(b) Equilibrium finding: Compute an equilibrium (NE/CCE/CE) for the normal-form game over the space of pure policies with the estimated payoff functions $\{\overline{V}_i^k(\pi)\}_{i=1}^n$. The joint policy returned by the equilibrium finding step is then executed in the next episode to generate a new trajectory.

By the algorithmic design, to strike a balance between exploration and exploitation, it is crucial to construct $\{\overline{V}_i^k(\pi)\}_{i=1}^n$ in such a way that promotes exploration. To this end, we solve a regularized optimization problem over the hypothesis class $\mathcal{F}^{(i)}$ to obtain $\overline{V}_i^k(\pi)$, where the objective function balances exploration with exploitation. We introduce the details of MAMEX as follows.

**Policy Evaluation.** For each $k \in [K]$, before the $k$-th episode, we have collected $k - 1$ trajectories $\tau^{1:k-1} = \cup_{t=1}^{k-1}\{s_1^t, a_1^t, r_1^t, \cdots, s_H^t, a_H^t, r_H^t\}$. For any $i \in [n]$, $\pi \in \Pi^{\mathrm{pur}}$ and $f \in \mathcal{F}^{(i)2}$, we can define a data-dependent discrepancy function $L^{(i),k-1}(f, \pi, \tau^{1:k-1})$. Such a function measures the in-sample error of the hypothesis $f$ with respect a policy $\pi$, evaluated on the historical data $\tau^{1:k-1}$. The specific form of such a function differs under the model-free and model-based settings. In particular, as we will show in (3.2) and (3.3) below, under the model-free setting, $L^{(i),k-1}(f, \pi, \tau^{1:k-1})$ is constructed based on the mean-squared Bellman error with respect to the Bellman operator $\mathcal{T}_h^{(i),\pi}$ in (2.2), while under the model-based setting, $L^{(i),k-1}(f, \pi, \tau^{1:k-1})$ is constructed based on the negative log-likelihood loss. Then, for each $\pi \in \Pi^{\mathrm{pur}}$ and $i \in [n]$, we define $\overline{V}_i^k(\pi)$ as

$$\overline{V}_i^k(\pi) = \sup_{f \in \mathcal{F}^{(i)}} \Big\{ \widehat{V}^{(i),\pi,k}(f) := \underbrace{V_f^{(i),\pi}(\rho)}_{(a)} \underbrace{-\eta \cdot L^{(i),k-1}(f, \pi, \tau^{1:k-1})}_{(b)} \Big\}. \tag{3.1}$$

---

[2]For ease of notation, under the model-based setting, we denote $\mathcal{F}^{(i)} = \mathcal{F}$ for all agent $i \in [n]$.

**Equilibrium Finding.** Afterwards, the algorithm utilizes the equilibrium oracle EQ (Line 4 of Algorithm 1) to compute an equilibrium (NE/CCE/CE) for the normal-form game over $\Pi^{\mathrm{pur}}$ with payoff functions $\{\overline{V}_i^k(\pi)\}_{i=1}^n$. The solution to the equilibrium oracle is a mixed strategy $\pi^k$, i.e., a probability distribution over $\Pi^{\mathrm{pur}}$.

Finally, we sample a random pure policy $\zeta^k$ from $\pi^k$ and execute $\zeta^k$ in the $k$-th episode to generate a new trajectory. See Algorithm 1 for the details of MAMEX. Here, we implicitly assume that $\Pi^{\mathrm{pur}}$ is finite for ease of presentation. For example, $\Pi^{\mathrm{pur}}$ is the set of all deterministic policies. When $\Pi^{\mathrm{pur}}$ is infinite, we can replace $\Pi^{\mathrm{pur}}$ by a $1/K$-cover of $\Pi^{\mathrm{pur}}$ with respect to the distance $d^{(i)}(\pi^{(i)}, \widetilde{\pi}^{(i)}) = \max_{s \in \mathcal{S}} \|\pi^{(i)}(\cdot \mid s) - \widetilde{\pi}^{(i)}(\cdot \mid s)\|_1$.

Furthermore, the objective $\widehat{V}^{(i),\pi,k}(f)$ in (3.1) is constructed by a sum of (a) the value function $V_f^{(i),\pi}(\rho)$ of $\pi$ under the hypothesis $f$ and (b) a regularized term $-\eta \cdot L^{(i),k-1}(f, \pi, \tau^{1:k-1})$, and the payoff function $\overline{V}_i^k(\pi)$ is obtained by solving a maximization problem over $\mathcal{F}^{(i)}$. The two terms (a) and (b) represent the "exploration" and "exploitation" objectives, respectively, and the parameter $\eta > 0$ controls the trade-off between them. To see this, consider the case where we only have the term (b) in the objective function. In the model-based setting, (3.1) reduces to the maximum likelihood estimation (MLE) of the model $f$ given the historical data $\tau^{1:k-1}$. Then $\pi^k$ returned by Line 4 is the equilibrium policy computed from the MLE model. Thus, without term (a) in $\widehat{V}^{(i),\pi,k}(f)$, the algorithm only performs exploitation. In addition to fitting the model, the term (a) also encourages the algorithm to find a model with a large value function under the given policy $\pi$, which promotes exploration. Under the model-free setting, only having term (b) reduces to least-squares policy evaluation (LSPE) (Sutton & Barto, 2018), and thus term (b) also performs exploitation only.

**Comparison with Single-Agent MEX (Liu et al., 2023).** When reduced to the single-agent MDP, MAMEX can be further simplified to the single-agent MEX algorithm (Liu et al., 2023). In particular, when $n = 1$, equilibrium finding is reduced to maximizing the function defined in (3.1) over single-agent policies, i.e., $\max_\pi \max_{f \in \mathcal{F}} \widehat{V}^{\pi,k}(f)$. By exchanging the order of the two maximizations, we obtain an optimization problem over the hypothesis class $\mathcal{F}$, which recovers the single-agent MEX (Liu et al., 2023). In contrast, in general-sum MGs, the equilibrium policy can no longer be obtained by a single-objective optimization problem. Hence, it is unviable to directly extend MEX to optimize over hypothesis space in MARL. Instead, MAMEX solves an optimization over $\mathcal{F}$ in the style of MEX for each pure policy $\pi \in \Pi^{\mathrm{pur}}$, and then computes the NE/CCE/CE of the normal-form game over the space of pure policies.

**Comparison with Existing MARL Algorithms with Function Approximation** Previous RL algorithms for MGs with the general function approximation usually require solving minimax optimization (Chen et al., 2022b; Zhan et al., 2022a; Foster et al., 2023) or constrained optimization subproblems within data-dependent constraints (Wang et al., 2023). In comparison, the optimization subproblems of MEX are single-objective and do not have data-dependent constraints, and thus seem easier to implement. For example, in practice, the inner problem can be solved by a regularized version of TD learning (Liu et al., 2023), and the outer equilibrium finding can be realized by any fast method to calculate equilibrium (Hart & Mas-Colell, 2000; Anagnostides et al., 2022).

In the following, we instantiate the empirical discrepancy function $L^{(i),k-1}$ for both the model-free setting and the model-based setting.

**Model-Free Algorithm** Under the model-free setting, we define the empirical discrepancy function $L$ as follows. For any $h \in [H]$ and $k \in [K]$, let $\xi_h^k = \{s_h^k, a_h^k, s_{h+1}^k\}$. For any $i \in [n]$, $\pi \in \Pi^{\mathrm{pur}}$ and $f \in \mathcal{F}^{(i)}$, we define

$$L^{(i),k-1}(f, \pi, \tau^{1:k-1}) = \sum_{h=1}^{H} \sum_{j=1}^{k-1} \left[ \left( l_h^{(i)}(\xi_h^j, f, f, \pi) \right)^2 - \inf_{f_h' \in \mathcal{F}_h^{(i)}} \left( l_h^{(i)}(\xi_h^j, f', f, \pi) \right)^2 \right], \qquad (3.2)$$

where $l_h^{(i)}(\xi_h^j, f, g, \pi) = (f_h(s_h^j, a_h^j) - r_h^{(i)}(s_h^j, a_h^j) - \langle g_{h+1}(s_{h+1}^j, \cdot), \pi_{h+1}(\cdot \mid s_{h+1}^j) \rangle_{\mathcal{A}})^2$ is the mean-squared Bellman error involving $f_h$ and $g_{h+1}$.

In Lemma E.1, we can show that $L^{(i),k-1}(f,\pi,\tau^{1:k-1})$ is an upper bound of $\sum_{s=1}^{k-1}\ell^{(i),s}(f,\pi)$, where $\ell^{(i),s}$ is defined in (2.4). Thus, the function $L^{(i),k-1}$ can be used to control the training error in the definition of MADC.

**Model-Based Algorithm** For the model-based setting, we define $L^{(i),k-1}$ as the negative log-likelihood:

$$L^{(i),k-1}(f,\pi,\tau^{1:k-1}) = \sum_{h=1}^{H}\sum_{j=1}^{k-1} -\log \mathbb{P}_{h,f}(s_{h+1}^{j} \mid s_{h}^{j}, a_{h}^{j}). \tag{3.3}$$

As we will show in Lemma E.3, the function $L^{(i),k-1}$ can be used to control the training error in (2.3), where $\ell^{(i),s}$ is defined in (2.5).

## 3.2 THEORETICAL RESULTS

In this subsection, we present our main theoretical results and show that MAMEX (Algorithm 1) is sample-efficient for learning NE/CCE/CE in the context of general function approximation.

**Theorem 3.1.** Let the discrepancy function $\ell^{(i),s}$ in (2.3) be defined in (2.4) and (2.5) for model-free and model-based settings, respectively. Suppose Assumptions 2.3 and 2.5 hold. By setting $K \geq 16$ and $\eta = 4/\sqrt{K} \leq 1$, with probability at least $1 - \delta$, the regret of Algorithm 1 after $K$ episodes is upper bounded by

$$\mathrm{Reg}_{\mathrm{NE,CCE,CE}}(K) \leq \widetilde{\mathcal{O}}\Big(nH\sqrt{K}\Upsilon_{\mathcal{F},\delta} + nd_{\mathrm{MADC}}\sqrt{K} + nd_{\mathrm{MADC}}H\Big),$$

where $\widetilde{\mathcal{O}}(\cdot)$ hides absolute constants and polylogarithmic terms in $H$ and $K$, and $\Upsilon_{\mathcal{F},\delta}$ is a term that quantifies the complexity of the hypothesis class $\mathcal{F}$. In particular, we have $\Upsilon_{\mathcal{F},\delta} = R^2 \log(\max_{i\in[n]}\mathcal{N}_{\mathcal{F}^{(i)}}(1/K) \cdot |\Pi^{\mathrm{pur}}|/\delta)$ in the model-free setting and $\Upsilon_{\mathcal{F},\delta} = \log(\mathcal{B}_{\mathcal{F}}(1/K)/\delta)$ in the model-based setting.

Theorem 3.1 shows that our MAMEX achieves a sublinear $\sqrt{K}$-regret for learning NE/CCE/CE, where the multiplicative factor depends polynomially on the number of agents $n$ and horizon $H$. Thus, MAMEX is sample-efficient in the context of the general function approximation. Moreover, the regret depends on the complexity of the hypothesis class via two quantifies – the MADC $d_{\mathrm{MADC}}$, which captures the inherent challenge of exploring the dynamics of the MG, and the quantity $\Upsilon_{\mathcal{F},\delta}$, which characterizes the complexity of estimating the true hypothesis $f^*$ based on data. To be more specific, in the model-free setting, since we need to evaluate each pure policy, $\Upsilon_{\mathcal{F},\delta}$ contains $\log|\Pi^{\mathrm{pur}}|$ due to uniform concentration. When reduced to the tabular setting, we can choose $\Pi^{\mathrm{pur}}$ to be the set of deterministic policies, and both $\Upsilon_{\mathcal{F},\delta}$ and $d_{\mathrm{MADC}}$ are polynomials of $|\mathcal{S}|$ and $|\mathcal{A}|$. Furthermore, when specialized to tractable special cases with function approximation and some special pure policy class such as log-linear policy class Cayci et al. (2021), we show in §D that Theorem D.8 yields regret upper bounds comparable to existing works. Moreover, using the standard online-to-batch techniques, we can transform the regret bound into a sample complexity result. We defer the details to §E.3.

## 4 CONCLUSION

In this paper, we study multi-player general-sum MGs under the general function approximation. We propose a unified algorithmic framework MAMEX for both model-free and model-based RL problems with the general function approximation. Compared with previous works that either solve constrained optimization subproblems within data-dependent sub-level sets (Wang et al., 2023), or complex multi-objective minimax optimization subproblems (Chen et al., 2022b; Foster et al., 2023), the implementation of MAMEX requires only an oracle for solving a single-objective unconstrained optimization problem with an equilibrium oracle of a normal-form game, thus being more amenable to empirical implementation. Moreover, we introduce a complexity measure MADC to capture the exploration-exploitation tradeoff for general-sum MGs. We prove that MAMEX is provably sample-efficient in learning NE/CCE/CE on RL problems with small MADCs, which covers a rich class of MG models. When specialized to the special examples with small MADCs, the regret of MAMEX is comparable to existing algorithms that are designed for specific MG subclasses.

ACKNOWLEDGEMENTS

Zhaoran Wang acknowledges National Science Foundation (Awards 2048075, 2008827, 2015568, 1934931), Simons Institute (Theory of Reinforcement Learning), Amazon, J.P. Morgan, and Two Sigma for their supports. Zhuoran Yang acknowledges Simons Institute (Theory of Reinforcement Learning) for its support.

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

# Supplementary Material

## CONTENTS

# Appendix

## A    RELATED WORK

**Markov Games**    Markov Game (MG) (Littman, 1994) is a popular model of multi-agent reinforcement learning, which generalizes the Markov decision process to multiple agents. A series of recent works design the sample-efficient algorithm for two-agent zero-sum games (Wei et al., 2017; Zhang et al., 2020; Xie et al., 2020; Bai et al., 2020; Bai & Jin, 2020; Bai et al., 2021; Zhao et al., 2021; Huang et al., 2021; Jin et al., 2022; Chen et al., 2022b;d). For instance, Bai & Jin (2020) provide a sample-efficient algorithm in an episodic MG based on optimistic value iteration. Xie et al. (2020); Chen et al. (2022d) mainly focus on zero-sum MGs with a linear structure. Huang et al. (2021); Jin et al. (2022); Chen et al. (2022b) further consider the two-player zero-sum MGs under general function approximation, and provide algorithms with a sublinear regret. Another line of research focuses on general-sum MGs with multiple players (Jin et al., 2020a; Liu et al., 2021; Tian et al., 2021; Jin et al., 2021b; Song et al., 2021; Liu et al., 2022b; Daskalakis et al., 2022; Zhan et al., 2022a; Cui et al., 2023; Wang et al., 2023). Some of previous works (Liu et al., 2021; Tian et al., 2021; Liu et al., 2022b) consider learning all three equilibrium notions — NE, CCE, and CE — and their regret or sample complexity results are exponential in the number of agents. To break this exponential curse, some existing works propose decentralized algorithms for learning CCE or CE rather than NE (Jin et al., 2021b; Daskalakis et al., 2022; Zhan et al., 2022a; Cui et al., 2023; Wang et al., 2023).

Moreover, there are also some works to study how to learn the equilibrium from a practical perspective. The EGTA (Wellman, 2006) uses a graph to represent the deviation for all profiles and identify the Nash equilibrium and whether a profile is relatively stable. Note if we want to evaluate all profiles and construct the whole graph, we need many samples to estimate the payoff functions for each profile, and then identify all the deviations and construct the graph. The author also mentions that one may apply machine learning techniques to fit a payoff function over the entire profile space given the available data. In fact, we use the function approximation technique to derive an estimate of all the profiles without estimating all the profiles. Thus, our approach can be considered as a learning approach to evaluate all the profiles.

Some previous works (Lanctot et al., 2017; Marris et al., 2021) propose a practical algorithm PSRO to compute the equilibrium in Markov Games, which also needs an equilibrium-solving oracle to learn the equilibrium. To be more specific, the PSRO learns the equilibrium in the following way: At first, every player chooses a uniform policy as their strategy. The algorithm then calculates the equilibrium by a meta-solver and trains an oracle that outputs the best response $\pi_i$ of the equilibrium for player $i$. After that, the algorithm adds $\pi_i$ into the strategy space of the player $i$. Last, the algorithm simulates all the new joint policy and construct a new normal-form game for the next iteration. However, in each iteration, it should simulate all the new joint policies and estimate the return. Consequently, the sample complexity increases exponentially as the iteration rounds increase. Different from PSRO, MAMEX utilizes the function approximation technique to the value function. The precise characterization of the structure of the value function can help us to evaluate the policy without actually simulating the environment with more samples. To be more specific, at each round, instead of simulating the environment and getting a Monte-Carlo return of each joint policy, MAMEX only needs to solve a regularized optimization problem over the function space of the value function. The solution of the optimization problem is used to be a payoff for the normal-form game. Since solving this optimization subproblem does not need to additional samples, MAMEX bypasses the requirement for exponential samples to simulate the environment and estimate the value for each joint policy $\pi$. This characteristic enhances its sample efficiency in comparison to PSRO.

**MARL with Function Approximation**    There are many papers working on multi-player general-sum MGs with the function approximation (Zhan et al., 2022a; Ni et al., 2022; Chen et al., 2022b; Wang et al., 2023; Cui et al., 2023; Foster et al., 2023) that build upon previous works for function approximation in the single-agent setting (Jiang et al., 2017; Sun et al., 2019; Jin et al., 2020b; Wang et al., 2020b; Dann et al., 2021; Du et al., 2021; Jin et al., 2021a; Foster et al., 2021; Chen et al., 2022c; Agarwal & Zhang, 2022; Zhong et al., 2022; Liu et al., 2023). In recent years, Xiong

et al. (2022) consider the multi-agent decoupling coefficient in the two-player zero-sum MGs, and provide the posterior sampling algorithm. However, unlike a zero-sum MG, a general-sum MG can have various equilibrium concepts, each of which aligns with a specific set of policies. Hence, their definition of the multi-agent decoupling coefficient cannot be extended to the general-sum setting. Chen et al. (2022b) and Foster et al. (2023) generalize the complexity measure Decision-Estimation Coefficient (DEC), and learn the equilibria in model-based general-sum MGs. Ni et al. (2022) provide both a model-based algorithm and a model-free algorithm for the low-rank MGs. Some previous works (Zhan et al., 2022a; Wang et al., 2023; Cui et al., 2023) provide model-free algorithms that learn CCE and CE with polynomial sample complexity. Compared to their works, this paper provides a unified algorithmic framework for both model-free and model-based MARL problems, which learns NE/CCE/CE efficiently under general function approximation and provides comparable regret to existing works. In particular, our work provides the first model-free algorithm for learning NE/CCE/CE of general-sum MGs in the context of the general function approximation.

## B  NOTATION

For $n$ sets $\mathcal{F}_1, \cdots, \mathcal{F}_n$, we let $\otimes_{i=1}^n \mathcal{F}_i$ denote $\mathcal{F}_1 \times \cdots \times \mathcal{F}_n$. For a set $\mathcal{A}$, we denote $\Delta(\mathcal{A})$ as a set of probability distributions over $\mathcal{A}$. For a vector $x \in \mathbb{R}^n$, we denote $\|x\|_1 = \sum_{i=1}^n |x_i|$, $\|x\|_2 = \sqrt{\sum_{i=1}^n x_i^2}$ and $\|x\|_\infty = \max_{i=1}^n |x_i|$. For a function $f : \mathcal{X} \mapsto \mathcal{Y}$, we denote $\|f\|_\infty = \sup_{x \in \mathcal{X}} |f(x)|$ as the infinity norm. For two functions $f, g : \mathcal{A} \mapsto \mathbb{R}$, we denote $\langle f, g \rangle_{\mathcal{A}} = \mathbb{E}_{a \in \mathcal{A}}[f(x)g(x)]$ as the inner product with respect to the set $\mathcal{A}$. For a Hilbert space $\mathcal{V}$ and $f, g \in \mathcal{V}$, we denote $\langle f, g \rangle_{\mathcal{V}}$ as the inner product defined in the Hilbert space $\mathcal{V}$, and $\|f\|_{\mathcal{V}}$ is the norm defined in Hilbert space $\mathcal{V}$. For two distributions over $P, Q \in \Delta(\mathcal{X})$, the Hellinger distance is defined as $D_{\mathrm{H}}^2(P\|Q) = \frac{1}{2}\mathbb{E}_{x \sim P}[(\sqrt{dP(x)/dQ(x)} - 1)^2]$. For a vector $x \in \mathbb{R}^d$, the softmax mapping is denoted by $\mathrm{Softmax}(x) \in \mathbb{R}^d$ with $\big(\mathrm{Softmax}(x)\big)_i = e^{x_i} / \sum_{i \in [d]} e^{x_i}$.

## C  ADDITIONAL DEFINITIONS

### C.1  NE/CCE/CE-REGRET

In the following, we provide the definitions of Coarse Correlated Equilibrium (CCE), Correlated Equilibrium (CE), and the corresponding regret CCE-regret and CE-regret. A Coarse Correlated Equilibrium is a *joint* policy $\pi$ such that no agent can achieve higher rewards by only changing its local policy. Compared with a NE, a CCE allows different agents to be correlated, while NE only considers product policies.

**Definition C.1** ($\varepsilon$-Coarse Correlated Equilibrium). A *joint policy* $\pi$ is a $\varepsilon$-Coarse Correlated Equilibrium if $V^{(i),\mu^{(i)},\pi}(\rho) \leq V^{(i),\pi}(\rho) + \varepsilon$ for all $i \in [n]$.

Here, the definition of $\varepsilon$-CCE is similar to that of an $\varepsilon$-NE. But here $\pi$ is a joint policy, i.e., the randomness of the local policies of the $n$ agents can be coupled together. As a result, CCE is a more general equilibrium notion than NE. Similarly, we can define the CCE-regret, which represents the cumulative suboptimality across all agents with respect to CCE.

**Definition C.2** (CCE-Regret). For all $k \in [K]$, let $\pi^k$ denote the *joint* policy that is deployed in the $k$-th episode, then the CCE-regret is defined as

$$\mathrm{Reg}_{\mathrm{CCE}}(K) = \sum_{k=1}^K \sum_{i=1}^n \big(V^{(i),\mu^{(i),\pi^k}}(\rho) - V^{(i),\pi^k}(\rho)\big).$$

Last, the Correlated Equilibrium has been extensively studied in previous works for MARL (Jin et al., 2020a; Chen et al., 2022b; Cui et al., 2023; Wang et al., 2023). To introduce the concept of CE, we need first to introduce the *strategy modification*. A strategy modification for the $i$-th agent is a mapping $\phi_i : \Pi_i^{\mathrm{pur}} \to \Pi_i^{\mathrm{pur}}$. Given any random policy $\pi$, the best strategy modification for $i$-th agent is defined as $\arg\max_{\phi_i} \mathbb{E}_{\upsilon \sim \pi}[V^{\phi_i(\upsilon^{(i)}) \times \upsilon^{(-i)}}(\rho)]$. A CE is a joint policy $\pi$ such that no agent can achieve higher rewards by only changing its local policy through strategic modification.

**Definition C.3** ($\varepsilon$-Correlated Equilibrium). A *joint policy* $\pi$ is a $\varepsilon$-Correlated Equilibrium if $\max_{\phi_i} \mathbb{E}_{\upsilon \sim \pi}[V^{\phi_i(\upsilon^{(i)}) \times \upsilon^{(-i)}}(\rho)] \leq V^\pi(\rho) + \varepsilon$ for any agent $i \in [n]$.

We can similarly define CE-regret as the sum of suboptimality terms with respect to CE.

**Definition C.4** (CE-Regret). For any $k \in [K]$, let $\pi^k$ denote the joint policy that is deployed in the $k$-th episode, the CE-regret is defined as

$$\text{Reg}_{\text{CE}}(K) = \sum_{k=1}^{K} \sum_{i=1}^{n} \left( \max_{\phi_i} \mathbb{E}_{\upsilon \sim \pi^k} \left( V^{(i), \phi_i(\upsilon^{(i)}) \times \upsilon^{(-i)}}(\rho) \right) - V^{(i), \pi^k}(\rho) \right).$$

Compared to the NE/CCE regret, the strategy modification of one agent in CE can be correlated to the policies of other agents. Instead, the best response is independent of the other agents.

## C.2 COVERING NUMBER AND BRACKETING NUMBER

When a function class $\mathcal{F}$ is infinite, the $\delta$-covering number and the $\delta$-bracketing number serve as surrogates of the cardinality of $\mathcal{F}$. Intuitively, the $\delta$-covering number is the minimum number of balls of radius $\delta$ required to cover a set.

**Definition C.5** ($\delta$-Covering Number). The $\delta$-covering number of a function class $\mathcal{F}$ with respect to distance metric $d$, denoted as $\mathcal{N}_{\mathcal{F}}(\delta, d)$, is the minimum integer $q$ satisfying the following property: there exists a subset $\mathcal{F}' \subseteq \mathcal{F}$ with $|\mathcal{F}'| = q$ such that for any $f_1 \in \mathcal{F}$ we can find $f_2 \in \mathcal{F}'$ with $d(f_1, f_2) \leq \delta$. To simplify the notation, we write $\mathcal{N}_{\mathcal{F}}(\delta, \| \cdot \|_\infty)$ as $\mathcal{N}_{\mathcal{F}}(\delta)$.

**Definition C.6** ($\delta$-Bracketing Number). A $\delta$-bracket of size $N$ is a bracket $\{g_1^i, g_2^i\}_{i=1}^N$, where $g_1^i$ and $g_2^i$ are functions mapping any policy $\pi$ and trajectory $\tau$ to $\mathbb{R}$, such that for all $i \in [N], \pi \in \Pi$ we have $\|g_1^i(\pi, \cdot) - g_2^i(\pi, \cdot)\| \leq \delta$. Also, for any $f \in \mathcal{F}$, there must exist an $i \in [N]$ such that $g_1^i(\pi, \tau_H) \leq \mathbb{P}_f^\pi(\tau_H) \leq g_2^i(\pi, \tau_H)$ for all possible $\tau_H$ and $\pi$. The $\delta$-bracketing number of $\mathcal{F}$, denoted by $\mathcal{B}_{\mathcal{F}}(\delta)$, is the minimum size of a $\delta$-bracket.

# D RELATIONSHIPS BETWEEN MADC AND TRACTABLE RL PROBLEMS

In this section, we show that the class of MGs with finite MADCs contains a rich class of models. Thus, when applied to these concrete MARL models, Theorem 3.1 shows that MAMEX learns NE/CCE/CE with provable sample efficiency.

In the sequel, we instantiate the discrepancy function $\ell^{(i),s}$ for both model-free and model-based MARL, and introduce some concrete general-sum MG models that satisfy Assumption 2.5.

## D.1 MODEL-FREE MARL PROBLEMS

Now we provide function classes with small MADCs including multi-agent counterparts of models with low Bellman eluder dimensions (Jin et al., 2021a; Huang et al., 2021) and Bilinear Classes (Du et al., 2021). Then, we introduce some concrete examples in these members and show that the regret upper bound of MAMEX in Theorem 3.1, when specialized to these special cases, are comparable to existing works.

**Multi-Agent Bellman Eluder Dimension** Recently, Jin et al. (2021a) introduce a model-free complexity measure called Bellman Eluder dimension (BE dimension) and show that function classes with low BE dimensions contain a wide range of RL problems such as linear MDP (Jin et al., 2020b), kernel MDP (Jin et al., 2021a) and function classes with low eluder dimension (Wang et al., 2020a). In this subsection, we extend the notion of BE dimension to MARL. First, we introduce the definition of $\varepsilon$-independence between distributions and the concept of distribution eluder dimension.

**Definition D.1** ($\varepsilon$-Independent Distributions). Let $\mathcal{G}$ be a function class on $\mathcal{X}$, and $\upsilon, \mu_1, \cdots, \mu_n$ are probability distributions over $\mathcal{X}$. We called $\upsilon$ is $\varepsilon$-independent of $\{\mu_1 \cdots \mu_n\}$ with respect to $\mathcal{G}$ if there exists a function $g \in \mathcal{G}$ such that $\sqrt{\sum_{i=1}^n (\mathbb{E}_{\mu_i}[g])^2} \leq \varepsilon$ and $|\mathbb{E}_\upsilon[g]| > \varepsilon$.

By this definition, if $\nu$ is $\varepsilon$-dependent of $\{\mu_1, \cdots, \mu_n\}$, then whenever we have $\sqrt{\sum_{i=1}^n (\mathbb{E}_{\mu_i}[g])^2} \leq \varepsilon$ for some $g \in \mathcal{G}$, we also have $|\mathbb{E}_\upsilon[g]| \leq \varepsilon$.

**Definition D.2** (Distribution Eluder Dimension). Let $\mathcal{G}$ be a function class on $\mathcal{X}$ and $\mathcal{D}$ be a family of probability measures over $\mathcal{X}$. The distributional eluder dimension $\dim_{\mathrm{DE}}(\mathcal{G}, \mathcal{D}, \varepsilon)$ is the length of the longest sequence $\rho_1, \cdots, \rho_n \subseteq \mathcal{D}$ such that there exists $\varepsilon' \geq \varepsilon$ where $\rho_i$ is $\varepsilon'$-independent of $\{\rho_1, \cdots, \rho_{i-1}\}$ for all $i \in [n]$.

In other words, distributional eluder dimension $\dim_{\mathrm{DE}}(\mathcal{G}, \mathcal{D}, \varepsilon)$ is the length of the longest sequences of distributions in $\mathcal{D}$ such that each element is $\varepsilon'$-independent of its predecessors with respect to $\mathcal{G}$, from some $\varepsilon' \geq \epsilon$. Such a notion generalizes the standard eluder dimension Russo & Van Roy (2013) to the distributional setting. When we set $\mathcal{D}$ to be the set of Dirac measures $\{\delta_x(\cdot)\}_{x \in \mathcal{X}}$, the distributional eluder dimension $\dim_{\mathrm{DE}}(\mathcal{G} - \mathcal{G}, \mathcal{D}, \varepsilon)$ reduces to the standard eluder dimension introduced in Russo & Van Roy (2013). Here, $\mathcal{G} - \mathcal{G} = \{g_1 - g_2 \colon g_1, g_2 \in \mathcal{G}\}$.

For any agent $i$ and any pure policy $\pi \in \Pi^{\mathrm{pur}}$, we denote the function class of the Bellman residual as $\mathcal{F}_h^{(i),\pi} = \{f_h - \mathcal{T}^{(i),\pi} f_{h+1} \mid f \in \mathcal{F}^{(i)}\}$. Now we introduce the definition of the multi-agent BE dimension with respect to a class of distributions.

**Definition D.3** (Multi-Agent Bellman Eluder Dimension). Let $D = \{D_h\}_{h \in [H]}$ be a set of $H$ classes of distributions over $\mathcal{S} \times \mathcal{A}$, one for each step of an episode. The multi-agent Bellman eluder (BE) dimension with respect to $D$ is defined as

$$\dim_{\mathrm{MABE}}(\mathcal{F}, \mathcal{D}, \varepsilon) = \max_{h \in [H]} \max_{i \in [n]} \left\{ \dim_{\mathrm{DE}}\left( \bigcup_{\pi \in \Pi^{\mathrm{pur}}} \mathcal{F}_h^{(i),\pi}, \mathcal{D}_h, \varepsilon \right) \right\}. \qquad \text{(D.1)}$$

In other words, the multi-agent BE dimension is defined as the maximum of the distribution eluder dimensions with respect to $D_h$, based on the agent-specific Bellman residue classes $\bigcup_{\pi \in \Pi^{\mathrm{pur}}} \mathcal{F}_h^{(i),\pi}$. Compared with the BE dimension for single-agent RL (Jin et al., 2021a), the multi-agent version takes the maximum over the agent index $i \in [n]$, and the function class involves the union of the function class $\mathcal{F}_h^{(i),\pi}$ for all $\pi \in \Pi^{\mathrm{pur}}$. In comparison, leveraging the facts that the optimal policy is the greedy policy of the optimal value function, and that the optimal value function is the fixed point of the Bellman optimality operator, it suffices to only consider residues of the Bellman optimality operator in the definition of single-agent BE dimension. In contrast, for general-sum MGs, finding the desired equilibrium policies is not a single-objective policy optimization problem, and the notion of the Bellman optimality operator is not well-defined. As a result, to extend the concept of Bellman eluder dimension to general-sum MGs, in the function class, we take into account $\mathcal{F}_h^{(i),\pi}$ for all $\pi \in \Pi^{\mathrm{pur}}$, which correspond to evaluating the performance of all the pure policies. Besides, in (D.1), we also take the maximum over all agents $i \in [n]$ and all steps $h \in [H]$, which aligns with the definition of single-agent BE dimension.

Furthermore, in the definition of multi-agent BE dimension, we need to specify a set of distributions $D = \{D_h\}_{h \in [H]}$ over $\mathcal{S} \times \mathcal{A}$. We consider two classes. First, let $D_\Delta = \{D_{\Delta,h}\}_{h \in [H]}$ denote a class of probability measures over $\mathcal{S} \times \mathcal{A}$ with $D_{\Delta,h} = \{\delta_{(s,a)}(\cdot) \mid (s,a) \in \mathcal{S} \times \mathcal{A}\}$, which contains all the Dirac measures that put mass one to a state-action pair at step $h$. Second, given the set of pure policies $\Pi^{\mathrm{pur}}$, we let $D_\Pi = \{D_{\Pi,h}\}_{h \in [H]}$ denote a class of probability measures induced $\Pi^{\mathrm{pur}}$ as follows. For any $\pi \in \Pi^{\mathrm{pur}}$, when all the agents follow $\pi$ on the true MG model, they generate a Markov chain $\{s_h, a_h\}_{h \in [H]}$ whose joint distribution is determined by $\pi$, denoted by $\mathbb{P}^\pi$. Then, for any $h \in [H]$, we define $D_{\Pi,h} = \{\rho \in \Delta(\mathcal{S} \times \mathcal{A}) \mid \rho(\cdot) = \mathbb{P}^\pi((s_h, a_h) = \cdot), \pi \in \Pi^{\mathrm{pur}}\}$, i.e., $D_{\Pi,h}$ denotes the collection of all marginal distributions of $(s_h, a_h)$ induced by pure policies.

In the following, to simplify the notation, we denote

$$\dim_{\mathrm{MABE}}(\mathcal{F}, \varepsilon) = \min\{\dim_{\mathrm{MABE}}(\mathcal{F}, \mathcal{D}_\Delta, \varepsilon), \ \dim_{\mathrm{MABE}}(\mathcal{F}, \mathcal{D}_\Pi, \varepsilon)\}. \qquad \text{(D.2)}$$

The following theorem shows that, when $\mathcal{F}$ satisfies realizability and completeness (Assumption 2.3), for a general-sum MG with a finite multi-agent BE dimension given by (D.2), its multi-agent decoupling coefficient (Definition 2.4) is also bounded. In other words, Assumption 2.5 holds any general-sum MG model with a low multi-agent BE dimension. As a result, the class of MGs with finite multi-agent BE dimensions is a subclass of MGs with finite multi-agent decoupling coefficients.

**Theorem D.4** (Low Multi-Agent BE Dimension $\subseteq$ Low MADC). Let $K$ any integer and let $\mathcal{F}$ be a hypothesis class under the model-free setting, i.e., a class of $Q$-functions. Assume that $\mathcal{F}$ satisfy the realizability and completeness condition specified in Assumption 2.3. Suppose that $\mathcal{F}$ has a finite multi-agent BE dimension $d = \dim_{\mathrm{MABE}}(\mathcal{F}, 1/K)$, then with the discrepancy function $\ell^{(i),s}$ given

in (2.4), the multi-agent decoupling coefficient of $\mathcal{F}$ satisfies $d_{\mathrm{MADC}} = \mathcal{O}(dH \log K)$, where $\mathcal{O}(\cdot)$ omits absolute constants.

*Proof.* See §F.1 for a detailed proof. □

Combining Theorem 3.1 and Theorem D.4, we obtain that MAMEX achieves a sublinear $\widetilde{\mathcal{O}}(ndH\sqrt{K} + ndH^2 + nHR^2\sqrt{K}\log \Upsilon_{\mathcal{F},\delta})$ regret for function classes with a finite multi-agent BE dimension $d$. It remains to see that that function classes with low multi-agent BE dimensions contain a wide range of RL problems. To this end, we prove that if the eluder dimension (Russo & Van Roy, 2013) of the function class $\mathcal{F}_h^{(i)}$ is small for all $h \in [H]$ and $i \in [n]$, then $\mathcal{F} = \otimes_{i=1}^{n}(\otimes_{h=1}^{H}\mathcal{F}_h^{(i)})$ has a low multi-agent BE dimension. Function classes with finite eluder dimension contains linear, generalized linear, and kernel functions (Russo & Van Roy, 2013), and thus contains a wide rage of MG models. On these MG problems, the model-free version of MAMEX achieve sample efficiency provably.

**Theorem D.5.** Suppose $\mathcal{F}$ satisfies Assumption 2.3. For any $i \in [n]$ and $h \in [H]$, let $\dim_{\mathrm{E}}(\mathcal{F}_h^{(i)}, \varepsilon)$ denote the eluder dimension of $\mathcal{F}_h^{(i)}$, which is a special case of the distributional eluder dimension introduced in Definition D.2. That is, $\dim_{\mathrm{E}}(\mathcal{F}_h^{(i)}, \varepsilon)$ is equal to $\dim_{\mathrm{DE}}(\mathcal{F}_h^{(i)} - \mathcal{F}_h^{(i)}, D_\Delta, \varepsilon)$, where $\mathcal{F}_h^{(i)} - \mathcal{F}_h^{(i)} = \{g\colon g = f_1 - f_2, f_1, f_2 \in \mathcal{F}_h^{(i)}\}$ and $D_\Delta$ contains the class of dirac measures on $\mathcal{S} \times \mathcal{A}$. Then, the multi-agent BE dimension defined in (D.2) satisfy

$$\dim_{\mathrm{MABE}}(\mathcal{F}, \varepsilon) \leq \max_{h \in [H]} \max_{i \in [n]} \dim_{\mathrm{E}}(\mathcal{F}_h^{(i)}, \varepsilon).$$

*Proof.* See §F.2 for a detailed proof. □

**Multi-Agent Bilinear Classes** Bilinear Classes (Du et al., 2021) consists of MDP models where the Bellman error admits a bilienar structure. On these models, Du et al. (2021) propose online RL algorithms that are provably sample-efficient. Thus, Bilinear Classes is a family of tractable MDP models with general function approximation. In the sequel, we extend Bilinear Classes to general-sum MGs and show that such an extension covers some notable special cases studied in the existing works. Then, we prove that multi-agent Bilinear Classes have a small MADC, thus satisfying the Assumption 2.5. Therefore, when applied to these problems, MAMEX provably achieves sample efficiency.

**Definition D.6** (Multi-Agent Bilinear Classes). Let $\mathcal{V}$ be a Hilbert space and let $\langle \cdot, \cdot \rangle_\mathcal{V}$ and $\|\cdot\|_\mathcal{V}$ denote the inner product and norm on $\mathcal{V}$. Given a multi-agent general-sum MG with a hypothesis class $\mathcal{F}$ satisfying Assumption 2.3, it belongs to multi-agent Bilinear Classes if there exist $H$ functions $\{W_h^{(i)} : \mathcal{F}^{(i)} \times \Pi^{\mathrm{pur}} \mapsto \mathcal{V}\}_{h=1}^{H}$ for each agent $i \in [n]$ and $\{X_h : \Pi^{\mathrm{pur}} \mapsto \mathcal{V}\}_{h=1}^{H}$ such that the Bellman error of each agent $i$ can be factorized using $W_h^{(i)}$ and $X_h$. That is, for each $i \in [n]$, $f \in \mathcal{F}^{(i)}, h \in [H], \pi, \pi' \in \Pi^{\mathrm{pur}}$, we have

$$\left| \mathbb{E}_{(s_h, a_h) \sim \pi'}\left[ f_h(s_h, a_h) - r_h^{(i)}(s_h, a_h) - \mathbb{E}_{s' \sim \mathbb{P}_h(s'|s_h, a_h)}\langle f_{h+1}(s', \cdot), \pi_{h+1}(\cdot \mid s')\rangle_\mathcal{A} \right] \right|$$
$$= \left| \langle W_h^{(i)}(f, \pi) - W_h^{(i)}(f^{(i),\mu^{(i),\pi}}, \mu^{(i),\pi}), X_h(\pi') \rangle_\mathcal{V} \right|, \tag{D.3}$$

where $\mu^{(i),\pi} = (\pi^{(i),\dagger}, \pi^{(-i)})$ is the best response for $i$-th agent given that the other agents all follow $\pi$. Here, the function $f^{(i),\mu^{(i),\pi}}$ is the fixed point of $\mathcal{T}^{(i),\mu^{(i),\pi}}$, i.e.,

$$f_h^{(i),\mu^{(i),\pi}} = \mathcal{T}^{(i),\mu^{(i),\pi}} f_{h+1}^{\mu^{(i),\pi}}. \tag{D.4}$$

Moreover, we require that $\{W_h^{(i)}, X_h\}_{h \in [H]}$ satisfy a regularity condition

$$\sup_{\pi \in \Pi^{\mathrm{pur}}, h \in [H]} \|X_h(\pi)\|_\mathcal{V} \leq 1, \qquad \sup_{i \in [n], f \in \mathcal{F}^{(i)}, \pi \in \Pi^{\mathrm{pur}}, h \in [H]} \|W_h^{(i)}(f, \pi)\|_\mathcal{V} \leq B_W, \tag{D.5}$$

where $B_W$ is a constant.

In this definition, for any $\pi \in \Pi^{\mathrm{pur}}$ and $f \in \mathcal{F}^{(i)}$,

$$f_h(s_h, a_h) - r_h^{(i)}(s_h, a_h) - \mathbb{E}_{s' \sim \mathbb{P}_h(s'|s_h, a_h)} \langle f_{h+1}(s', \cdot), \pi_{h+1}(\cdot \mid s') \rangle_{\mathcal{A}}$$

is the Bellman error of $f$ at $(s_h, a_h)$ for evaluating policy $\pi$ on behalf of agent $i$. On the left-hand side of (D.3), we evaluate such a Bellman error with respect to the distribution induced by another policy $\pi'$. Equation (D.3) shows that this error can be factorized into the inner product between $W_h^{(i)}$ and $X_h^{(i)}$, where both $W_h^{(i)}$ only involves $(f, \pi)$ while $X_h^{(i)}$ only involves $\pi'$. Thus, multi-agent Bilinear Classes specifies a family of Markov games whose Bellman error satisfies a factorization property. Furthermore, recall that the best response $\pi^{(i),\dagger} = \max_{\nu \in \Delta(\Pi_i^{\mathrm{pur}})} V^{\nu, \pi^{(-i)}}$ is attained at some pure policy, thus we have $\mu^{(i),\pi} \in \Pi^{\mathrm{pur}}$. Under Assumption 2.3, the fixed point $f^{(i),\mu^{(i),\pi}}$ in (D.4) is guaranteed to exist and belongs to $\mathcal{F}$.

We define $\mathcal{X}_h = \{X_h(\pi) : f \in \mathcal{F}, \pi \in \Pi^{\mathrm{pur}}\}$ and $\mathcal{X} = \bigcup_{h=1}^{H} \mathcal{X}_h$. The complexity of the multi-agent bilinear class essentially is determined by the complexity of the Hilbert space $\mathcal{V}$. To allow $\mathcal{V}$ be infinite-dimensional, we introduce the notion of information gain, which characterizes the intrinsic complexity of $\mathcal{V}$ in terms of exploration.

**Definition D.7** (Information Gain). Suppose $\mathcal{V}$ is a Hilbert space and $\mathcal{X} \subseteq \mathcal{V}$. For $\varepsilon > 0$ and integer $K > 0$, the information gain $\gamma_K(\varepsilon, \mathcal{X})$ is defined by

$$\gamma_K(\varepsilon, \mathcal{X}) = \max_{x_1, \cdots, x_K \in \mathcal{X}} \log \det \left( I + \frac{1}{\varepsilon} \sum_{k=1}^{K} x_k x_k^\top \right).$$

The following theorem shows that multi-agent Bilinear Classes with small information gain have low MADCs.

**Theorem D.8** (Multi-Agent Bilinear Classes $\subseteq$ Low MADC). For a general-sum MG in the multi-agent bilinear class with a hypothesis class $\mathcal{F}$, let $\gamma_K(\varepsilon, \mathcal{X}) = \sum_{h=1}^{H} \gamma_K(\varepsilon, \mathcal{X}_h)$ be the information gain. Then, Assumption 2.5 holds with the discrepancy function $\ell^{(i),s}$ given in (2.4). In particular, we have

$$d_{\mathrm{MADC}} \leq \max \left\{ 1, 8R^2 \cdot \gamma_K(1/(KB_W^2), \mathcal{X}) \right\},$$

where $B_W$ is given in (D.5) and $R \in (0, H]$ is an upper bound on $\sum_h^H r_h$.

*Proof.* See §F.3 for a detailed proof. $\qquad\square$

Now we introduce some concrete members of multi-agent Bilinear Classes, which are general-sum MGs with linear function approximation. In single-agent RL, linear Bellman complete MDPs (Wang et al., 2019) assume that the MDP model satisfies the Bellman completeness condition with respect to linear $Q$-functions. We can extend such a model to general-sum MGs.

**Example D.9** (Linear Bellman Complete MGs). We say a Markov Game is a *linear Bellman complete MG* of dimension $d$, if for any step $h \in [H]$ there exists a known feature $\phi_h : \mathcal{S} \times \mathcal{A} \mapsto \mathbb{R}^d$ with $\|\phi_h(s, a)\| \leq 1$ for all $(s, a) \in \mathcal{S} \times \mathcal{A}$ such that Assumption 2.3 holds for linear functions of $\phi_h$. In other words, the Markov game satisfies Assumption 2.3 with $\mathcal{F}_h^{(i)} \subseteq \{\phi_h^\top \theta \mid \theta \in \mathbb{R}^d, \|\theta\|_2 \leq \sqrt{d_\theta}\}$ for all $i \in [n]$ and $h \in [H]$, where $d_\theta > 0$ is a parameter.

It is easy to see that Linear Bellman complete MGs belong to multi-agent Bilinear Classes by choosing

$$X_h(\pi) = \mathbb{E}_\pi[\phi(s_h, a_h)] \in \mathbb{R}^d, \qquad W_h^{(i)}(f, \pi) = \theta_{f,h} - w_{f,h}^{(i)},$$

where $\theta_{f,h}$ satisfies that $f(s_h, a_h) = \theta_{f,h}^\top \phi_h(s_h, a_h)$, and $w_{f,h}^{(i)}$ satisfies that[3]

$$(w_{f,h}^{(i)})^\top \phi_h(s_h, a_h) = r_h^{(i)}(s_h, a_h) + \mathbb{E}_{s' \sim \mathbb{P}_h(\cdot|s_h, a_h)} \langle f_{h+1}(s', \cdot), \pi_{h+1}(\cdot \mid s') \rangle_{\mathcal{A}}$$
$$= \mathcal{T}^{(i),\pi}(f_{h+1}) \in \mathcal{F}_h^{(i)}.$$

---

[3]If there are multiple $\theta$ satisfying the requirement, we can break the tie arbitrarily.

Then, we have $\mathcal{X}_h \subseteq \mathcal{V} = \{\phi \in \mathbb{R}^d : \|\phi\|_2 \le 1\}$ for all $h \in [H]$ and $B_W = 2\sqrt{d}$. It can be shown that the logarithm of $1/K$-covering number of $\mathcal{F}$ is $\log(\mathcal{N}_{\mathcal{F}}(1/K)) = \widetilde{\mathcal{O}}(d)$, and the information gain can bounded by

$$\gamma_K(1/B_W^2 K, \mathcal{X}) = \sum_{h=1}^H \gamma_K(1/B_W^2 K, \mathcal{X}_h) \le \sum_{h=1}^H \gamma_K(1/4dK, \mathcal{X}_h) = \widetilde{\mathcal{O}}(Hd),$$

where $\widetilde{\mathcal{O}}$ omits absolute constants and logarithmic factors (Du et al., 2021; Wang et al., 2020b). Thus, by Theorem 3.1, MAMEX achieves a $\widetilde{\mathcal{O}}(ndHR^2\sqrt{K} + nHR^2\sqrt{K}\log|\Pi^{\mathrm{pur}}| + ndH^2)$ regret. For the single-agent setting, comparing to the state-of-the-art $\widetilde{\mathcal{O}}(dH\sqrt{K})$ regret when $R = 1$ (Zanette et al., 2020; Chen et al., 2022c), our result matches their results in terms of $d, H$ and $K$ with an extra factor $|\Pi^{\mathrm{pur}}|$ in the logarithmic term. Note that when the pure policy set of $i$-th agent is selected as some particular policy classes such as log-linear policy

$$\Pi_{h,i}^{\mathrm{pur}} = \{\pi_\vartheta : \pi_\vartheta(\cdot \mid s) = \mathrm{Softmax}(\vartheta^\top \psi(s, \cdot)), \|\vartheta\|_2 \le 1, \|\psi(\cdot, \cdot)\| \le 1, \vartheta \in \mathbb{R}^{d_\pi}\},$$

we can select a cover by

$$\widehat{\Theta} = \{\widehat{\vartheta} : \widehat{\vartheta}_i = \lfloor \vartheta_i/\varepsilon \rfloor \times \varepsilon, \|\vartheta\|_2 \le 1, \vartheta \in \mathbb{R}^{d_\pi}\}.$$

Zanette et al. (2021) prove that the logarithm of cardinality of the induced covering $\{\pi_\vartheta : \vartheta \in \widehat{\Theta}\}^H$ is bounded by $\widetilde{\mathcal{O}}(nHd_\pi)$, and then MAMEX provides a $\widetilde{\mathcal{O}}((nd+n^2d_\pi)H^2R^2\sqrt{K}+ndH^2)$ regret.

In particular, as one of the examples of Linear Bellman Complete MGs, Xie et al. (2020) consider a similar linear structure for two-player zero-sum games.

**Example D.10** (Zero-Sum Linear MGs (Xie et al., 2020))**.** In a zero-sum linear MG, for each $(s, a, b) \in \mathcal{S} \times \mathcal{A} \times \mathcal{B}$ and $h \in [H]$, we have reward $r_h(s, a, b) \in [0, 1]$, and there is a known feature map $\phi : \mathcal{S} \times \mathcal{A} \times \mathcal{B} \to \mathbb{R}^d$, $H$ known vectors $\theta_h \in \mathbb{R}^d$ and a vector of $d$ unknown measures $\mu_h = \{\mu_{h,d'}\}_{d' \in [d]}$ on $\mathcal{S}$ such that $\|\phi(\cdot, \cdot, \cdot)\|_2 \le 1$, $\|\theta_h\|_2 \le \sqrt{d}$, $\|\mu_h(\mathcal{S})\|_2 \le \sqrt{d}$ and

$$r_h(s, a, b) = \phi(s, a, b)^\top \theta_h, \ \ \mathbb{P}_h(\cdot \mid s, a, b) = \phi(s, a, b)^\top \mu_h(\cdot).$$

Zero-sum linear MGs is a special case of linear Bellman complete MG with two players and $d_\theta = 2H\sqrt{d}$, and our algorithm provides a $\widetilde{\mathcal{O}}(dH^3\sqrt{K}+H^3\sqrt{K}\log(|\Pi^{\mathrm{pur}}|))$ regret by choosing $R = H$ and the fact that $\log \mathcal{N}_{\mathcal{F}}^{(i)}(1/K) = \widetilde{\mathcal{O}}(d)$. The previous work provides a $\widetilde{\mathcal{O}}(d^{3/2}H^2\sqrt{K})$ sublinear regret (Xie et al., 2020) and a $\Omega(dH^{3/2}\sqrt{K})$ information-theoretic lower bound (Chen et al., 2022d) for zero-sum linear MGs. Thus, our regret matches the lower bound in terms of $d$, has a higher order in $H$ compared to Xie et al. (2020) and an extra factor $\log|\Pi^{\mathrm{pur}}|$. Again, we can adopt the class of log-linear policies with a policy cover, which leads to $\log|\Pi^{\mathrm{pur}}| = \widetilde{\mathcal{O}}(d_\pi)$. Thus, MAMEX yields a $\widetilde{\mathcal{O}}((dH^3 + d_\pi H^4)\sqrt{K})$ regret.

### D.2 Model-Based RL Problems

Sun et al. (2019) provide a complexity measure — *witness rank* — to characterize the exploration hardness of the model-based RL problems. In the following, we extend the notion of the witness rank to MARL.

**Example D.11** (Multi-Agent Witness Rank)**.** Let $\mathcal{V} = \{\mathcal{V}_h : \mathcal{S} \times \mathcal{A} \times \mathcal{S} \mapsto [0, 1]\}_{h \in [H]}$ denote a class of discriminators and let $\mathcal{F}$ be a hypothesis class such that the true model, denoted by $f^*$, belongs to $\mathcal{F}$. We say a multi-agent witness rank of a general-sum MG is at most $d$, if for any model $f \in \mathcal{F}$ and any policy $\pi \in \Pi^{\mathrm{pur}}$ there exist mappings $\{X_h : \Pi^{\mathrm{pur}} \to \mathbb{R}^d\}_{h=1}^H$ and $\{W_h : \mathcal{F} \to \mathbb{R}^d\}_{h=1}^H$

$$\max_{v \in \mathcal{V}_h} \mathbb{E}_{(s_h, a_h) \sim \pi}[(\mathbb{E}_{s' \sim \mathbb{P}_{h,f}(\cdot \mid s_h, a_h)} - \mathbb{E}_{s' \sim \mathbb{P}_{h,f^*}(\cdot \mid s_h, a_h)})v(s_h, a_h, s')] \ge \langle W_h^{(i)}(f), X_h(\pi)\rangle,$$
(D.6)

$$\kappa_{\mathrm{wit}} \cdot \mathbb{E}_{(s_h, a_h) \sim \pi}[(\mathbb{E}_{s' \sim \mathbb{P}_{h,f}(\cdot \mid s_h, a_h)} - \mathbb{E}_{s' \sim \mathbb{P}_{h,f^*}(\cdot \mid s_h, a_h)})V_{h+1,f}^{(i),\pi}(s')] \le \langle W_h^{(i)}(f), X_h(\pi)\rangle \quad \text{(D.7)}$$

for all $h \in [H]$, where $\kappa_{\mathrm{wit}}$ is a parameter. Here, $V_{h+1,f}^{(i),\pi}$ is the value function of $\pi$ associated with agent $i$ under model $f$. Moreover, these mappings satisfy the following regularity condition:

$$\sup_{h \in [H], \pi \in \Pi^{\mathrm{pur}}} \|X_h(\pi)\| \le 1, \qquad \sup_{h \in [H], f \in \mathcal{F}, i \in [n]} \|W_h^{(i)}(f)\| \le B_W.$$

Compared with the single-agent witness rank (Sun et al., 2019), the policy $\pi$ in the mapping $X_h(\pi)$ and the expectation $\mathbb{E}_{(s_h, a_h) \sim \pi}$ in (D.6) and (D.7) can be an arbitrary pure policy instead of the optimal policy $\pi_f$ of the model $f$. This stricter assumption is essential for general-sum MGs because we are interested in various equilibrium notions and each equilibrium can be non-unique. The following theorem shows that model classes with small multi-agent witness ranks have small MADCs.

**Theorem D.12** (Multi-Agent Witness Rank $\subseteq$ Low MADC)**.** Let $\mathcal{F}$ be a class of general-sum MGs whose multi-agent witness rank is no more than $d$. Then, for any $f^* \in \mathcal{F}$, we have $d_{\mathrm{MADC}} = \widetilde{\mathcal{O}}(Hd/\kappa_{\mathrm{wit}}^2)$, where $d_{\mathrm{MADC}}$ is the multi-agent decoupling coefficient of $f^*$.

*Proof.* See §F.4 for detailed proof. $\qquad\qquad\square$

This theorem shows that the multi-agent decoupling coefficient is upper bounded by the multi-agent witness rank, which shows that the class of MG models with a finite multi-agent decoupling coefficient contains models with a finite multi-agent witness rank. Hence, many concrete MG models such as the multi-agent version of factor MDP and linear kernel MDP all have finite multi-agent decoupling coefficients. Therefore, applying Theorem 3.1 to models with a finite Multi-Agent witness rank, the model-based version of MAMEX achieves a $\widetilde{\mathcal{O}}(nHd\sqrt{K}/\kappa_{\mathrm{wit}}^2 + nH\sqrt{K})$ regret with witness rank $d$. Note that for the model-based RL problems, our regret does not have the term $\log(|\Pi^{\mathrm{pur}}|)$, because the discrepancy function $\ell^{(i),s}$ in 2.5 is independent with $\pi^k$. When applying our results to the single-agent setting, Theorem D.12 provides a similar regret result as in previous works (Sun et al., 2019; Zhong et al., 2022).

Another example of model-based RL problems is the linear mixture MG (Chen et al., 2022d), which assumes that the transition kernel $\mathbb{P}(s' \mid s, a)$ is a linear combination of $d$ feature mappings $\{\phi_i(s', s, a)\}_{i \in [d]}$, i.e. $\mathbb{P}(s' \mid s, a) = \sum_{i=1}^{d} \theta_i \phi_i(s', s, a)$, where $a$ is a joint action.

**Example D.13** (Multi-Agent Linear Mixture MGs)**.** We call one general-sum MG is a *linear mixture MG* with dimension $d$, if there exist $h$ vectors $\{\theta_h \in \mathbb{R}^d\}_{h \in [H]}$ and a known feature $\phi(s' \mid s, a) \in \mathbb{R}^d$ such that $\|\theta_h\|_2 \leq \sqrt{d}$ and $\mathbb{P}_h(s' \mid s, a) = \langle \theta_h, \phi(s' \mid s, a) \rangle$ for any state-action pair $(s', s, a) \in \mathcal{S} \times \mathcal{S} \times \mathcal{A}$.

The following theorem shows that a linear mixture general-sum MG has a finite multi-agent decoupling coefficient. Thus, MAMEX can be readily applied to these models with sample efficiency.

**Theorem D.14** (Multi-Agent Linear Mixture MGs $\subset$ Low MADC)**.** For a linear mixture MG with dimension $d$, we have $d_{\mathrm{MADC}} = \widetilde{\mathcal{O}}(dHR^4)$, where $R$ is an upper bound on $\sum_{h=1}^{H} r_h$.

*Proof.* See §F.5 for a detailed proof. $\qquad\qquad\square$

Chen et al. (2022d) provides a minimax-optimal $\widetilde{\mathcal{O}}(dH\sqrt{K})$ regret for two-player zero-sum MGs for $r_h \in [0, 1]$. Now choose $\mathcal{F}_h = \{\theta_h \in \mathbb{R}^d\}$. Combining with Theorem D.14 and Theorem 3.1, and the fact that $\log(\mathcal{B}_{\mathcal{F}}(1/K)) = \widetilde{\mathcal{O}}(Hd)$ (Liu et al., 2022a), MAMEX achieves a $\widetilde{\mathcal{O}}(ndH^5\sqrt{K} + ndH^4)$ regret, where we set $R = H$. Compared with their regret upper bound, when applying our result to two-player zero-sum MGs by choosing $n = 2$, the leading term of our regret $\widetilde{\mathcal{O}}(dH^5\sqrt{K})$ matches the minimax-optimal result in terms of $d$ and $K$ but with an extra multiplicative factor $H^2$.

# E   PROOF OF MAIN RESULTS

## E.1   PROOF OF MODEL-FREE VERSION OF THEOREM 3.1

*Proof.* We first consider learning Nash equilibrium and coarse correlated equilibrium.

**NE/CCE**   First, by Assumption 2.3, for any pure joint policy $\upsilon$, there exists a function $f^{(i),\upsilon} \in \mathcal{F}^{(i)}$ satisfies that it has no Bellman error with Bellman operator $\mathcal{T}^{(i),\upsilon}$ for any pure joint policy $\upsilon$, i.e.

$$\mathcal{T}_h^{(i),\upsilon} f_{h+1}^{(i),\upsilon} = f_h^{(i),\upsilon}. \tag{E.1}$$

Hence, $\{f_h^{(i),\upsilon}\}_{h\in[H]}$ is the $Q$-function of the agent $i$ when all agents follow the policy $\upsilon$. Thus, we have

$$V_{f^{(i),\upsilon}}^{(i),\upsilon}(\rho) = \mathbb{E}_{s_1\sim\rho,a\sim\upsilon(s_1)}[f_1^{(i),\upsilon}(s,a)] = \mathbb{E}_{s_1\sim\rho,a\sim\upsilon(s_1)}[Q_1^{(i),\upsilon}(s,a)] = V^{(i),\upsilon}(\rho). \qquad \text{(E.2)}$$

Also, denote $\widehat{f}^{(i),\upsilon} = \arg\sup_{f\in\mathcal{F}^{(i)}}\widehat{V}_i^\upsilon(f)$ as the optimal function with respect to the regularized value $\widehat{V}^{(i),\pi}(f)$ for the pure joint policy $\pi$ and agent $i$. Now we have

$$\mathbb{E}_{\upsilon\sim\pi^k}\left[V_{\widehat{f}^{(i),\upsilon}}^{(i),\upsilon}(\rho) - \eta L^{(i),k-1}(\widehat{f}^{(i),\upsilon}, \upsilon, \tau^{1:k-1})\right] = \mathbb{E}_{\upsilon\sim\pi^k}\left[\sup_{f\in\mathcal{F}^{(i)}}\widehat{V}^{(i),\upsilon}(f)\right]$$

$$\geq \max_{\upsilon^{(i)}\in\Pi^{\mathrm{pur}}}\mathbb{E}_{\upsilon\sim\upsilon^{(i)}\times\pi^{(-i),k}}\left[\sup_{f\in\mathcal{F}^{(i)}}\widehat{V}^{(i),\upsilon}(f)\right]. \qquad \text{(E.3)}$$

The inequality holds because of the property of Nash Equilibrium or Coarse Correlated Equilibrium. Then, since the best response $\pi^{(i),k,\dagger}$ is a pure policy, we have

$$\max_{\upsilon^{(i)}\in\Pi^{\mathrm{pur}}}\mathbb{E}_{\upsilon\sim\upsilon^{(i)}\times\pi^{(-i),k}}\left[\sup_{f\in\mathcal{F}^{(i)}}\widehat{V}^{(i),\upsilon}(f)\right]$$

$$\geq \mathbb{E}_{\upsilon\sim\pi^{(i),k,\dagger}\times\pi^{(-i),k}}\left[\sup_{f\in\mathcal{F}^{(i)}}\widehat{V}^{(i),\upsilon}(f)\right] = \mathbb{E}_{\upsilon\sim\pi^{(i),k,\dagger}\times\pi^{(-i),k}}\left[\widehat{V}^{(i),\upsilon}(f^{(i),\upsilon})\right]$$

$$\geq \mathbb{E}_{\upsilon\sim\mu^{(i),\pi^k}}\left[V_{f^{(i),\upsilon}}^{(i),\upsilon}(\rho) - \eta L^{(i),k-1}(f^{(i),\upsilon}, \upsilon, \tau^{1:k-1})\right], \qquad \text{(E.4)}$$

where $\upsilon \in \Pi_i^{\mathrm{pur}}$, $\mu^{(i),\pi^k} = (\pi^{(i),k,\dagger}, \pi^{(-i),k})$ and $\pi^{(i),k,\dagger}$ is the best response given the action of other agents $\pi^{(-i),k}$. Thus, combining (E.3) and (E.4), we can derive

$$\mathbb{E}_{\upsilon\sim\mu^{(i),\pi^k}}\left[V_{f^{(i),\upsilon}}^{(i),\upsilon}(\rho)\right] - \mathbb{E}_{\upsilon\sim\pi^k}\left[V_{\widehat{f}^{(i),\upsilon}}^{(i),\upsilon}(\rho)\right]$$

$$\leq \eta\mathbb{E}_{\upsilon\sim\mu^{(i),\pi^k}}\left[L^{(i),k-1}(f^{(i),\upsilon}, \upsilon, \tau^{1:k-1})\right] - \eta\mathbb{E}_{\upsilon\sim\pi^k}\left[L^{(i),k-1}(\widehat{f}^{(i),\upsilon}, \upsilon, \tau^{1:k-1})\right]. \qquad \text{(E.5)}$$

Now we provide the concentration lemma, which shows that the empirical discrepancy function $L^{(i),k}(f, \pi, \tau^{1:k})$ is an estimate of the true discrepancy function $\sum_{s=0}^{k-1}\ell^{(i),s}(f, \pi)$.

**Lemma E.1** (Concentration Lemma). For any $k \in [K]$ pure joint policy $\pi$, and $\{\zeta^s\}_{s=1}^{k-1} \in \Pi$ that be executed in Algorithm 1 in the first $k-1$ episodes, with probability at least $1 - \delta$,

$$L^{(i),k-1}(f, \pi, \tau^{1:k-1}) - \frac{1}{4}\left(\sum_{s=0}^{k-1}\ell^{(i),s}(f, \pi)\right) \geq -\varepsilon_{\mathrm{conc}},$$

where $\varepsilon_{\mathrm{conc}} = \max\{\mathcal{O}(HR^2\log(HK\max_{i\in[n]}\mathcal{N}_{\mathcal{F}^{(i)}}(1/K)|\Pi^{\mathrm{pur}}|/\delta)), H\}$ and

$$\ell^{(i),s}(f, \pi) = \sum_{h=1}^{H}\mathbb{E}_{(s_h,a_h)\sim\zeta_h^s}\left[((f_h - \mathcal{T}_h^{(i),\pi}f_{h+1})(s_h, a_h))^2\right].$$

*Proof.* See §F.6 for a detailed proof. □

In other words, if we define the event as

$$\mathcal{E}_1 = \left\{L^{(i),k}(f, \pi, \tau^{1:k}) - \frac{1}{4}\left(\sum_{s=0}^{k-1}\ell^{(i),s}(f, \pi)\right) \geq \varepsilon_{\mathrm{conc}}, \forall f \in \mathcal{F}^{(i)}, \pi \in \Pi^{\mathrm{pur}}, k \in [K]\right\},$$

we have $\Pr\{\mathcal{E}_1\} \geq 1 - \delta$. Note that the $\varepsilon_{\mathrm{conc}}$ contains $\log(|\Pi^{\mathrm{pur}}|/\delta)$ in the logarithmic term, which arises from our policy-search style algorithm.

**Lemma E.2** (Optimal Concentration Lemma). For all index $i \in [n]$, all $\pi \in \Pi^{\mathrm{pur}}$ and function $f^{(i),\pi} \in \mathcal{F}^{(i)}$ such that $\mathcal{T}^{(i),\pi}f^{(i),\pi} = f^{(i),\pi}$, with probability at least $1 - \delta$, we have

$$L^{(i),k}(f^{(i),\pi}, \pi, \tau^{1:k}) \leq \varepsilon_{\mathrm{conc}}.$$

*Proof.* See §F.7 for a detailed proof. □

In other words, if we define the event as

$$\mathcal{E}_2 = \{\forall\, i \in [n], \pi \in \Pi^{\mathrm{pur}}, L^{(i),k}(f^{(i),\pi}, \pi, \tau^{1:k}) \le \varepsilon_{\mathrm{conc}}\},$$

we have $\Pr\{\mathcal{E}_2\} \ge 1 - \delta$. Lemma E.2 shows that the empirical discrepancy function $L^{(i),k}(f, \pi, \tau^{1:k})$ is small if the function $f$ and the policy $\pi$ are consistent, i.e. $f = f^{(i),\pi}$. Now by (E.5) and Lemma E.2, for any $i \in [n]$, under the event $\mathcal{E}_2$,

$$\mathbb{E}_{\upsilon \sim \mu^{(i)}, \pi^k}\left[ V^{(i),\upsilon}(\rho) \right] - \mathbb{E}_{\upsilon \sim \pi^k}\left[ V^{(i),\upsilon}(\rho) \right]$$

$$= \mathbb{E}_{\upsilon \sim \mu^{(i)}, \pi^k}\left[ V^{(i),\upsilon}_{f^{(i),\upsilon}}(\rho) \right] - \mathbb{E}_{\upsilon \sim \pi^k}\left[ V^{(i),\upsilon}(\rho) \right]$$

$$= \underbrace{\mathbb{E}_{\upsilon \sim \mu^{(i)}, \pi^k}\left[ V^{(i),\upsilon}_{f^{(i),\upsilon}}(\rho) \right] - \mathbb{E}_{\upsilon \sim \pi^k}\left[ V^{(i),\upsilon}_{\widehat{f}^{(i),\upsilon}}(\rho) \right]}_{(a)} + \mathbb{E}_{\upsilon \sim \pi^k}\left[ V^{(i),\upsilon}_{\widehat{f}^{(i),\upsilon}}(\rho) \right] - \mathbb{E}_{\upsilon \sim \pi^k}\left[ V^{(i),\upsilon}(\rho) \right].$$

By (E.5) and Lemma E.2, under event $\mathcal{E}_2$, $(a)$ can be bounded by

$$(a) \le \eta \mathbb{E}_{\upsilon \sim \mu^{(i)}, \pi^k}\left[ L^{(i),k-1}(f^{(i),\upsilon}, \upsilon, \tau^{1:k-1}) \right] - \eta \mathbb{E}_{\upsilon \sim \pi^k}\left[ L^{(i),k-1}(\widehat{f}^{(i),\upsilon}, \upsilon, \tau^{1:k-1}) \right] \quad \text{(E.6)}$$

$$\le \eta \varepsilon_{\mathrm{conc}} - \eta \mathbb{E}_{\upsilon \sim \pi^k}\left[ L^{(i),k-1}(\widehat{f}^{(i),\upsilon}, \upsilon, \tau^{1:k-1}) \right]. \quad \text{(E.7)}$$

Now by Assumption 2.5, on the events $\mathcal{E}_1$ and $\mathcal{E}_2$ we have

$$\mathrm{Reg}(K) = \sum_{k=1}^{K} \sum_{i=1}^{n} \left( V^{(i),\mu^{(i)},\pi^k}(\rho) - V^{(i),\pi^k}(\rho) \right)$$

$$= \sum_{k=1}^{K} \sum_{i=1}^{n} \left( \mathbb{E}_{\upsilon \sim \mu^{(i)}, \pi^k}\left[ V^{(i),\upsilon}(\rho) \right] - \mathbb{E}_{\upsilon \sim \pi^k}\left[ V^{(i),\upsilon}(\rho) \right] \right)$$

$$\le \sum_{k=1}^{K} \sum_{i=1}^{n} \left( \eta \varepsilon_{\mathrm{conc}} - \eta \mathbb{E}_{\upsilon \sim \pi^k}\left[ L^{(i),k-1}(\widehat{f}^{(i),\upsilon}, \upsilon, \tau^{1:k-1}) \right] \right.$$

$$\left. + \mathbb{E}_{\upsilon \sim \pi^k}\left[ V^{(i),\upsilon}_{\widehat{f}^{(i),\upsilon}}(\rho) \right] - \mathbb{E}_{\upsilon \sim \pi^k}\left[ V^{(i),\upsilon}(\rho) \right] \right). \quad \text{(E.8)}$$

Now since $\widehat{f}^{(i),\upsilon} = \arg\max_{f \in \mathcal{F}^{(i)}} \left[ V^{(i),\upsilon}_f(\rho) - \eta L^{(i),k-1}(f, \upsilon, \tau^{1:k-1}) \right]$ is the optimal function with respect to the regularized value, under the event $\mathcal{E}_2$ we have

$$V^{(i),\upsilon}_{\widehat{f}^{(i),\upsilon}}(\rho) - \eta L^{(i),k-1}(\widehat{f}^{(i),\upsilon}, \upsilon, \tau^{1:k-1}) \ge V^{(i),\upsilon}_{f^{(i),\upsilon}}(\rho) - \eta L^{(i),k-1}(f^{(i),\upsilon}, \upsilon, \tau^{1:k-1}),$$

then we have $\eta L^{(i),k-1}(\widehat{f}^{(i),\upsilon}, \upsilon, \tau^{1:k-1}) \ge 0$ and by $\eta \le 1$,

$$\eta L^{(i),k-1}(\widehat{f}^{(i),\upsilon}, \upsilon, \tau^{1:k-1}) \le V^{(i),\upsilon}_{\widehat{f}^{(i),\upsilon}}(\rho) - V^{(i),\upsilon}_{f^{(i),\upsilon}}(\rho) + \eta L^{(i),k-1}(f^{(i),\upsilon}, \upsilon, \tau^{1:k-1})$$

$$\le R + \eta \varepsilon_{\mathrm{conc}} \le 2\varepsilon_{\mathrm{conc}},$$

where the last inequality follows the Lemma E.2. If we define

$$L^{(i),k-1}_{2\varepsilon_{\mathrm{conc}}}(\widehat{f}^{(i),\upsilon}, \upsilon, \tau^{1:k-1}) = L^{(i),k-1}(\widehat{f}^{(i),\upsilon}, \upsilon, \tau^{1:k-1}) \cdot \mathbb{I}\{\eta L^{(i),k-1}(\widehat{f}^{(i),\upsilon}, \upsilon, \tau^{1:k-1}) \le 2\varepsilon_{\mathrm{conc}}\}$$

and the event as

$$\mathcal{E}_3 = \left\{ \forall\, i \in [n], \upsilon \in \Pi^{\mathrm{pur}}, L^{(i),k-1}_{2\varepsilon_{\mathrm{conc}}}(\widehat{f}^{(i),\upsilon}, \upsilon, \tau^{1:k-1}) = L^{(i),k-1}(\widehat{f}^{(i),\upsilon}, \upsilon, \tau^{1:k-1}) \right\},$$

we will have $\mathcal{E}_3 \subseteq \mathcal{E}_2$. Since the policy $\zeta^k$ that algorithm executes is sampled from $\pi^k$, then the sequence $\{Y_k\}_{k=1}^{K}$ that is defined by

$$Y_k = \mathbb{E}_{\upsilon \sim \pi^k}\left[ V^{(i),\upsilon}_{\widehat{f}^{(i),\upsilon}}(\rho) - V^{(i),\upsilon}(\rho) - \eta L^{(i),k-1}_{2\varepsilon_{\mathrm{conc}}}(\widehat{f}^{(i),\upsilon}, \upsilon, \tau^{1:k-1}) \right]$$

$$- \left( V^{(i),\zeta^k}_{\widehat{f}^{(i),\zeta^k}}(\rho) - V^{(i),\upsilon}(\rho) - \eta L^{(i),k-1}_{2\varepsilon_{\mathrm{conc}}}(\widehat{f}^{(i),\zeta^k}, \zeta^k, \tau^{1:k-1}) \right)$$

is a martingale difference sequence. Now by Azuma-Hoeffding's inequality and $Y_k \le R + 2\varepsilon_{\mathrm{conc}} \le 3\varepsilon_{\mathrm{conc}}$, with probability at least $1 - \delta$ we have

$$\left| \sum_{k=1}^{K} \left[ \mathbb{E}_{\upsilon \sim \pi^k} \left[ V_{\widehat{f}^{(i),\upsilon}}^{(i),\upsilon}(\rho) - V^{(i),\upsilon}(\rho) - \eta L_{2\varepsilon_{\mathrm{conc}}}^{(i),k-1}(\widehat{f}^{(i),\upsilon}, \upsilon, \tau^{1:k-1}) \right] \right. \right.$$
$$\left. \left. - \left( V_{\widehat{f}^{(i),\zeta^k}}^{(i),\zeta^k}(\rho) - V^{(i),\upsilon}(\rho) - \eta L_{2\varepsilon_{\mathrm{conc}}}^{(i),k-1}(\widehat{f}^{(i),\zeta^k}, \zeta^k, \tau^{1:k-1}) \right) \right] \right| \le \mathcal{O}(\varepsilon_{\mathrm{conc}}\sqrt{K}). \quad \text{(E.9)}$$

Define the event $\mathcal{E}_4$ as the (E.9) holds. Now by choosing $\frac{\eta}{4} = \frac{1}{\mu} = \frac{1}{\sqrt{K}}$ and taking the union bound over the event $\mathcal{E}_1, \mathcal{E}_2, \mathcal{E}_3$ and $\mathcal{E}_4$, with probability at least $1 - 4\delta$, we can get

$\mathrm{Reg}(K)$

$$\le \sum_{i=1}^{n} \sum_{k=1}^{K} \left( \eta\varepsilon_{\mathrm{conc}} - \eta\mathbb{E}_{\upsilon \sim \pi^k} \left[ L^{(i),k-1}(\widehat{f}^{(i),\upsilon}, \upsilon, \tau^{1:k-1}) \right] + \mathbb{E}_{\upsilon \sim \pi^k} \left[ V_{\widehat{f}^{(i),\upsilon}}^{(i),\upsilon}(\rho) \right] - \mathbb{E}_{\upsilon \sim \pi^k} \left[ V^{(i),\upsilon}(\rho) \right] \right)$$

$$= \sum_{i=1}^{n} \sum_{k=1}^{K} \left( \eta\varepsilon_{\mathrm{conc}} - \eta\mathbb{E}_{\upsilon \sim \pi^k} \left[ L_{2\varepsilon_{\mathrm{conc}}}^{(i),k-1}(\widehat{f}^{(i),\upsilon}, \upsilon, \tau^{1:k-1}) \right] + \mathbb{E}_{\upsilon \sim \pi^k} \left[ V_{\widehat{f}^{(i),\upsilon}}^{(i),\upsilon}(\rho) \right] - \mathbb{E}_{\upsilon \sim \pi^k} \left[ V^{(i),\upsilon}(\rho) \right] \right)$$

$$\le \underbrace{\sum_{i=1}^{n} \sum_{k=1}^{K} \left( \eta\varepsilon_{\mathrm{conc}} - \eta L^{(i),k-1}(\widehat{f}^{(i),\zeta^k}, \zeta^k, \tau^{1:k-1}) + V_{\widehat{f}^{(i),\zeta^k}}^{(i),\zeta^k}(\rho) - V^{(i),\zeta^k}(\rho) \right)}_{\text{(b)}} + \widetilde{\mathcal{O}}(n\varepsilon_{\mathrm{conc}}\sqrt{K}).$$

$$\text{(E.10)}$$

The first inequality holds because of Eq (E.8). The equality in the second line holds under Lemma E.2 (event $\mathcal{E}_3 \subseteq \mathcal{E}_2$). The second inequality is derived from Azuma-Hoeffding's inequality (event $\mathcal{E}_4$). Now using Lemma E.1 and MADC assumption, we can get

$$\text{(b)} \le -\sum_{i=1}^{n} \sum_{k=1}^{K} \left( \frac{\eta}{4} \left( \sum_{s=0}^{k-1} \ell^{(i),s}(f, \zeta^k) \right) \right) + \sum_{i=1}^{n} \sum_{k=1}^{K} \left( V_{\widehat{f}^{(i),\zeta^k}}^{(i),\zeta^k}(\rho) - V^{(i),\zeta^k}(\rho) \right) + 4n\sqrt{K} \cdot \eta\varepsilon_{\mathrm{conc}}$$

$$\le n\mu \cdot d_{\mathrm{MADC}} + 6d_{\mathrm{MADC}}H + 4n\sqrt{K}\varepsilon_{\mathrm{conc}}.$$

The second inequality uses Assumption 2.5. Now the regret can be bounded by

$$\mathrm{Reg}(K) \le n\sqrt{K} \cdot d_{\mathrm{MADC}} + 6d_{\mathrm{MADC}}H + 4n\sqrt{K}\varepsilon_{\mathrm{conc}} + \mathcal{O}(n\varepsilon_{\mathrm{conc}}\sqrt{K})$$
$$= \mathcal{O}(n\varepsilon_{\mathrm{conc}}\sqrt{K} + nd_{\mathrm{MADC}}H + nd_{\mathrm{MADC}}\sqrt{K}).$$

Hence, we complete the proof by noting that $\varepsilon_{\mathrm{conc}} = \widetilde{\mathcal{O}}(HR^2 \log \Upsilon_{\mathcal{F},\delta})$.

**CE**  By changing the best response to the strategy modification, we can derive a proof for Correlated Equilibrium (CE). We simplify the notation of strategy modification as $\phi_i(\upsilon^{(i)}) \times \upsilon^{(-i)}$ as $\phi_i(\upsilon)$. Now we have

$$\mathbb{E}_{\upsilon \sim \pi^k} \left[ V_{\widehat{f}^{(i),\upsilon}}^{(i),\upsilon}(\rho) - \eta L^{(i),k-1}(\widehat{f}^{(i),\upsilon}, \upsilon, \tau^{1:k-1}) \right]$$
$$= \mathbb{E}_{\upsilon \sim \pi^k} \left[ \sup_{f \in \mathcal{F}^{(i)}} \widehat{V}^{(i),\upsilon}(f) \right]$$
$$= \max_{\phi_i} \mathbb{E}_{\upsilon \sim \pi^k} \left[ \sup_{f \in \mathcal{F}^{(i)}} \widehat{V}^{(i),\phi_i(\upsilon^{(i)}) \times \upsilon^{(-i)}}(f) \right]. \quad \text{(E.11)}$$

The second equality holds because of the property of Correlated Equilibrium. Now we have

$$\max_{\phi_i} \mathbb{E}_{\upsilon \sim \pi^k} \left[ \sup_{f \in \mathcal{F}^{(i)}} \widehat{V}^{(i),\phi_i(\upsilon^{(i)}) \times \upsilon^{(-i)}}(f) \right]$$
$$\ge \max_{\phi_i} \mathbb{E}_{\upsilon \sim \pi^k} \left[ V_{f^{(i),\phi_i(\upsilon)}}^{(i),\phi_i(\upsilon)}(\rho) - \eta L^{(i),k-1}(f^{(i),\phi_i(\upsilon)}, \phi_i(\upsilon), \tau^{1:k-1}) \right]$$
$$\ge \max_{\phi_i} \mathbb{E}_{\upsilon \sim \pi^k} \left[ V_{f^{(i),\phi_i(\upsilon)}}^{(i),\phi_i(\upsilon)}(\rho) - \eta\varepsilon_{\mathrm{conc}} \right]. \quad \text{(E.12)}$$

The first equality holds by $f^{(i),\phi_i(v)} \in \mathcal{F}^{(i)}$ in (E.1), and the last inequality is derived from Lemma E.2 and $\phi_i(v)$ is a pure joint policy. Then, by combining (E.11) and (E.12), we can get

$$\max_{\phi_i} \mathbb{E}_{v\sim\pi^k}\left[V^{(i),\phi_i(v)}_{f^{(i)},\phi_i(v)}(\rho)\right] - \mathbb{E}_{v\sim\pi^k}\left[V^{(i),v}_{\widehat{f}^{(i)},v}(\rho)\right]$$

$$\leq \eta\varepsilon_{\text{conc}} - \eta\mathbb{E}_{v\sim\pi^k}\left[L^{(i),k-1}(\widehat{f}^{(i)},v, v, \tau^{1:k-1})\right].$$

Hence, we can upper bound the regret of the agent $i$ at $k$-th episode as

$$\max_{\phi_i} \mathbb{E}_{v\sim\pi^k}\left[V^{(i),\phi_i(v^{(i)})\times v^{(-i)}}(\rho)\right] - \mathbb{E}_{v\sim\pi^k}\left[V^{(i),v}(\rho)\right]$$

$$= \max_{\phi_i} \mathbb{E}_{v\sim\pi^k}\left[V^{(i),\phi_i(v)}_{f^{(i)},\phi_i(v)}(\rho)\right] - \mathbb{E}_{v\sim\pi^k}\left[V^{(i),v}(\rho)\right]$$

$$= \max_{\phi_i} \mathbb{E}_{v\sim\pi^k}\left[V^{(i),\phi_i(v)}_{f^{(i)},\phi_i(v)}(\rho)\right] - \mathbb{E}_{v\sim\pi^k}\left[V^{(i),v}_{\widehat{f}^{(i)},v}(\rho)\right] + \mathbb{E}_{v\sim\pi^k}\left[V^{(i),v}_{\widehat{f}^{(i)},v}(\rho)\right] - \mathbb{E}_{v\sim\pi^k}\left[V^{(i),v}(\rho)\right]$$

$$\leq \eta\varepsilon_{\text{conc}} - \eta\mathbb{E}_{v\sim\pi^k}L^{(i),k-1}(\widehat{f}^{(i)},v, v, \tau^{1:k-1}) + \mathbb{E}_{v\sim\pi^k}\left[V^{(i),v}_{\widehat{f}^{(i)},v}(\rho)\right] - \mathbb{E}_{v\sim\pi^k}\left[V^{(i),v}(\rho)\right].$$

The rest of the proof is the same as in NE/CCE after (E.7). □

### E.2 Proof of Model-Based Version of Theorem 3.1

*Proof.* We first consider NE/CCE.

**NE/CCE** Denote $\widehat{f}^{(i),\pi} = \arg\sup_{f\in\mathcal{F}} \widehat{V}^{\pi}_i(f)$ as the optimal model with respect to the regularized value $\widehat{V}^{(i),\pi}(f)$. Since for model-based RL problems, the empirical discrepancy function $L(f,\pi,\tau)$ and $\ell^{(i),s}(f,\pi)$ is independent with policy $\pi$, we simplify it as $L(f,\tau)$ and $\ell^{(i),s}(f)$. Then, from the definition of regularized value function $\widehat{V}^{(i),\pi}(f)$, we have

$$\mathbb{E}_{v\sim\pi^k}\left[V^{(i),v}_{\widehat{f}^{(i)},v}(\rho) - \eta L^{(i),k-1}(\widehat{f}^{(i)},v, \tau^{1:k-1})\right]$$

$$= \mathbb{E}_{v\sim\pi^k}\left[\sup_{f\in\mathcal{F}}\widehat{V}^{(i),v}(f)\right] \geq \max_{v^{(i)}\in\Pi^{\text{pur}}}\mathbb{E}_{v\sim v^{(i)}\times\pi^{(-i),k}}\left[\sup_{f\in\mathcal{F}}\widehat{V}^{(i),v}(f)\right]. \tag{E.13}$$

The inequality holds by the fact that $\pi^k$ is the NE/CCE of the regularized value function $\widehat{V}^{(i),\pi}(f)$. Now since the best response $\pi^{(i),k,\dagger}$ is a pure policy, we have

$$\max_{v^{(i)}\in\Pi^{\text{pur}}}\mathbb{E}_{v\sim v^{(i)}\times\pi^{(-i),k}}\left[\sup_{f\in\mathcal{F}}\widehat{V}^{(i),v}(f)\right]$$

$$\geq \mathbb{E}_{v\sim\pi^{(i),k,\dagger}\times\pi^{(-i),k}}\left[\sup_{f\in\mathcal{F}}\widehat{V}^{(i),v}(f)\right]$$

$$\geq \mathbb{E}_{v\sim\mu^{(i),\pi^k}}\left[V^{(i),v}_{f^*}(\rho) - \eta L^{(i),k-1}(f^*, \tau^{1:k-1})\right]. \tag{E.14}$$

Thus, by combining E.13 and E.14, we have

$$\mathbb{E}_{v\sim\mu^{(i),\pi^k}}\left[V^{(i),v}_{f^*}(\rho)\right] - \mathbb{E}_{v\sim\pi^k}\left[V^{(i),v}_{\widehat{f}^{(i)},v}(\rho)\right]$$

$$\leq \eta L^{(i),k-1}(f^*, \tau^{1:k-1}) - \eta\mathbb{E}_{v\sim\pi^k}\left[L^{(i),k-1}(\widehat{f}^{(i)},v, \tau^{1:k-1})\right]. \tag{E.15}$$

Now we provide our concentration lemma for model-based RL problems.

**Lemma E.3** (Concentration Lemma for Model-Based RL Problems). With probability at least $1-\delta$, for any $k \in [K], f \in \mathcal{F}$, for the executed policy $\{\zeta^s\}_{s=1}^{k-1}$ in Algorithm 1, we have

$$L^{(i),k-1}(f^*, \tau^{1:k-1}) - L^{(i),k-1}(f, \tau^{1:k-1}) \leq -\sum_{s=1}^{k-1}\ell^{(i),s}(f) + \kappa_{\text{conc}}, \tag{E.16}$$

where $\kappa_{\text{conc}} = \max\{2H\log\frac{H\mathcal{B}_{\mathcal{F}}(1/K)}{\delta}, H\}$, where $\mathcal{B}_{\mathcal{F}}(1/K)$ is the $1/K$-bracketing number of the model class $\mathcal{F}$. We also define the event $\mathcal{E}_5$ as the situation when (E.16) holds.

*Proof.* See §F.8 for detailed proof. □

By Lemma E.3, for any $i \in [n]$,

$$
\mathbb{E}_{\upsilon \sim \mu^{(i)}, \pi^k}\left[V^{(i),\upsilon}(\rho)\right] - \mathbb{E}_{\upsilon \sim \pi^k}\left[V^{(i),\upsilon}(\rho)\right]
$$
$$
= \underbrace{\mathbb{E}_{\upsilon \sim \mu^{(i)}, \pi^k}\left[V_{f^*}^{(i),\upsilon}(\rho)\right] - \mathbb{E}_{\upsilon \sim \pi^k}\left[V_{\widehat{f}^{(i)},\upsilon}^{(i),\upsilon}(\rho)\right]}_{(a)} + \mathbb{E}_{\upsilon \sim \pi^k}\left[V_{\widehat{f}^{(i)},\upsilon}^{(i),\upsilon}(\rho)\right] - \mathbb{E}_{\upsilon \sim \pi^k}\left[V^{(i),\upsilon}(\rho)\right]. \quad \text{(E.17)}
$$

Now substitute into equation (E.15),

$$
(a) \leq \eta L^{(i),k-1}(f^*, \tau^{1:k-1}) - \eta \mathbb{E}_{\upsilon \sim \pi^k}\left[L^{(i),k-1}(\widehat{f}^{(i)},\upsilon, \tau^{1:k-1})\right]
$$
$$
= \mathbb{E}_{\upsilon \sim \pi^k}\left[\eta L^{(i),k-1}(f^*, \tau^{1:k-1}) - \eta L^{(i),k-1}(\widehat{f}^{(i)},\upsilon, \tau^{1:k-1})\right]. \quad \text{(E.18)}
$$

Hence, combining with (E.17) and (E.18), we can get

$$
\mathbb{E}_{\upsilon \sim \mu^{(i)}, \pi^k}\left[V^{(i),\upsilon}(\rho)\right] - \mathbb{E}_{\upsilon \sim \pi^k}\left[V^{(i),\upsilon}(\rho)\right]
$$
$$
\leq (a) + \mathbb{E}_{\upsilon \sim \pi^k}\left[V_{\widehat{f}^{(i)},\upsilon}^{(i),\upsilon}(\rho)\right] - \mathbb{E}_{\upsilon \sim \pi^k}\left[V^{(i),\upsilon}(\rho)\right]
$$
$$
\leq \mathbb{E}_{\upsilon \sim \pi^k}\left[\eta L^{(i),k-1}(f^*, \tau^{1:k-1}) - \eta L^{(i),k-1}(\widehat{f}^{(i)},\upsilon, \tau^{1:k-1}) + V_{\widehat{f}^{(i)},\upsilon}^{(i),\upsilon}(\rho) - V^{(i),\upsilon}(\rho)\right]. \quad \text{(E.19)}
$$

By summing over $k \in [K]$ and $I \in [n]$, the regret can be obtained by

$$
\text{Reg}(K)
$$
$$
\leq \sum_{i=1}^{n} \sum_{k=1}^{K} \mathbb{E}_{\upsilon \sim \pi^k}\left[\eta L^{(i),k-1}(f^*, \tau^{1:k-1}) - \eta L^{(i),k-1}(\widehat{f}^{(i)},\upsilon, \tau^{1:k-1}) + V_{\widehat{f}^{(i)},\upsilon}^{(i),\upsilon}(\rho) - V^{(i),\upsilon}(\rho)\right]. \quad \text{(E.20)}
$$

Now we want to use Azuma-Hoeffding's inequality to transform $\upsilon \sim \pi^k$ to executed policy $\zeta^k$. To achieve this goal, note that by Lemma E.3, under event $\mathcal{E}_5$, we have

$$
L^{(i),k-1}(f^*, \tau^{1:k-1}) - L^{(i),k-1}(\widehat{f}^{(i)},\upsilon, \tau^{1:k-1}) \leq \kappa_{\text{conc}}. \quad \text{(E.21)}
$$

Moreover, since $\widehat{f}^{(i)},\upsilon$ achieves the maximum value of the regularized value function $\widehat{V}^{(i),\pi}(f) = V_f^{(i),\upsilon}(\rho) - L^{(i),k-1}(f^*, \tau^{1:k-1})$, we have

$$
L^{(i),k-1}(f^*, \tau^{1:k-1}) - L^{(i),k-1}(\widehat{f}^{(i)},\upsilon, \tau^{1:k-1}) \geq \mathbb{E}_{\upsilon \sim \mu^{(i)}, \pi^k}\left[V_{f^*}^{(i),\upsilon}(\rho)\right] - \mathbb{E}_{\upsilon \sim \pi^k}\left[V_{f^{(i)},\upsilon}^{(i),\upsilon}(\rho)\right]
$$
$$
\geq -R \geq -\kappa_{\text{conc}}.
$$

Thus, if we define

$$
\mathcal{L}_{\varepsilon}^{(i),\upsilon} = \left(L^{(i),k-1}(f^*, \tau^{1:k-1}) - L^{(i),k-1}(\widehat{f}^{(i)},\upsilon, \tau^{1:k-1})\right)
$$
$$
\cdot \mathbb{I}\left\{|L^{(i),k-1}(f^*, \tau^{1:k-1}) - L^{(i),k-1}(\widehat{f}^{(i)},\upsilon, \tau^{1:k-1})| \leq \varepsilon\right\},
$$

we can have $|\mathcal{L}_{\kappa_{\text{conc}}}^{(i),\upsilon}| \leq \kappa_{\text{conc}}$ is bounded under event $\mathcal{E}_5$. Then, with probability at least $1 - \delta$, $\mathcal{L}_{\kappa_{\text{conc}}}^{(i),\upsilon} = L^{(i),k-1}(f^*, \tau^{1:k-1}) - L^{(i),k-1}(\widehat{f}^{(i)},\upsilon, \tau^{1:k-1})$. Then, we can apply Azuma-Hoeffding's inequality to transform the expectation to the executed policy $\zeta^k$.

$$
\left|\sum_{k=1}^{K} \mathcal{L}_{\kappa_{\text{conc}}}^{(i),\zeta^k} - \sum_{k=1}^{K} \mathbb{E}_{\upsilon \sim \pi^k}\left[\mathcal{L}_{\kappa_{\text{conc}}}^{(i),\upsilon}\right]\right| = \mathcal{O}(\kappa_{\text{conc}} \cdot \log K). \quad \text{(E.22)}
$$

Now by taking the union bound of Azuma-Hoeffding's inequality and event $\mathcal{E}_5$, with probability at least $1 - 2\delta$,

$$
\begin{aligned}
\text{Reg}(K) &\leq \sum_{i=1}^{n} \sum_{k=1}^{K} \mathbb{E}_{v \sim \pi^k} \left[ \eta L^{(i),k-1}(f^*, \tau^{1:k-1}) - \eta L^{(i),k-1}(\widehat{f}^{(i),v}, \tau^{1:k-1}) + V_{\widehat{f}^{(i),v}}^{(i),v}(\rho) - V^{(i),v}(\rho) \right] \\
&= \sum_{i=1}^{n} \sum_{k=1}^{K} \mathbb{E}_{v \sim \pi^k} \left[ \eta \mathcal{L}_{\kappa_{\text{conc}}}^{v} + V_{\widehat{f}^{(i),v}}^{(i),v}(\rho) - V^{(i),v}(\rho) \right] \\
&\leq \underbrace{\sum_{i=1}^{n} \sum_{k=1}^{K} \left( \eta \mathcal{L}_{\kappa_{\text{conc}}}^{\zeta^k} + V_{\widehat{f}^{(i),\zeta^k}}^{(i),\zeta^k}(\rho) - V^{(i),\zeta^k}(\rho) \right)}_{\text{(b)}} + \widetilde{\mathcal{O}}(n\kappa_{\text{conc}}),
\end{aligned}
$$

where the first inequality holds by (E.20), the equality holds under event $\mathcal{E}_5$, and the last inequality holds by (E.22). Then, by Lemma E.3, under event $\mathcal{E}_5$, we have

$$
\begin{aligned}
\text{(b)} &= \sum_{i=1}^{n} \sum_{k=1}^{K} \left( \eta L^{(i),k-1}(f^*, \tau^{1:k-1}) - \eta L^{(i),k-1}(\widehat{f}^{(i),\zeta^k}, \tau^{1:k-1}) + V_{\widehat{f}^{(i),\zeta^k}}^{(i),v}(\rho) - V^{(i),\zeta^k}(\rho) \right) \\
&\leq \sum_{i=1}^{n} \sum_{k=1}^{K} \left( -\eta \sum_{s=1}^{k-1} \ell^{(i),s}(\widehat{f}^{(i),\zeta^k}) + \eta \kappa_{\text{conc}} + V_{\widehat{f}^{(i),\zeta^k}}^{(i),v}(\rho) - V^{(i),\zeta^k}(\rho) \right).
\end{aligned}
$$

Then, by Assumption 2.5, (b) can be further upper bounded by

$$
\begin{aligned}
\text{(b)} &\leq \sum_{i=1}^{n} \sum_{k=1}^{K} \left( -\eta \sum_{s=1}^{k-1} \ell^{(i),s}(\widehat{f}^{(i),\zeta^k}) + \eta \kappa_{\text{conc}} + V_{\widehat{f}^{(i),\zeta^k}}^{(i),v}(\rho) - V^{(i),\zeta^k}(\rho) \right) \\
&\leq n\eta K \kappa_{\text{conc}} + \frac{n}{\eta} d_{\text{MADC}} + 6n d_{\text{MADC}} H \\
&= \widetilde{\mathcal{O}}(n\kappa_{\text{conc}}\sqrt{K} + n d_{\text{MADC}}\sqrt{K} + n d_{\text{MADC}} H).
\end{aligned}
$$

The first inequality holds by Lemma E.3. The last equality holds by $\eta = 4/\sqrt{K}$. Finally, the regret can be bounded by

$$
\text{Reg}(K) \leq \text{(b)} + \widetilde{\mathcal{O}}(n\kappa_{\text{conc}}) = \widetilde{\mathcal{O}}(n\kappa_{\text{conc}}\sqrt{K} + n d_{\text{MADC}}\sqrt{K} + n d_{\text{MADC}} H).
$$

Thus, we complete the proof by noting that $\kappa_{\text{conc}} = \mathcal{O}(H)$

**Correlated Equilibrium** Similar to model-free problems, we only need to replace the best response with strategy modification.

$$
\begin{aligned}
\mathbb{E}_{v \sim \pi^k} &\left[ V_{\widehat{f}^{(i),v}}^{(i),v}(\rho) - \eta L^{(i),k-1}(\widehat{f}^{(i),v}, \tau^{1:k-1}) \right] \\
&= \mathbb{E}_{v \sim \pi^k} \left[ \sup_{f \in \mathcal{F}} \widehat{V}^{(i),v}(f) \right] \\
&= \max_{\phi_i} \mathbb{E}_{v \sim \pi^k} \left[ \sup_{f \in \mathcal{F}} \widehat{V}^{(i),\phi_i(v)}(f) \right].
\end{aligned}
$$

The last equality uses the property that $\pi^k$ is a CE with respect to the payoff function $\sup_{f \in \mathcal{F}} \widehat{V}^{(i),v}(f)$. Then, since $f^* \in \mathcal{F}$, we can further derive

$$
\begin{aligned}
\mathbb{E}_{v \sim \pi^k} &\left[ V_{\widehat{f}^{(i),v}}^{(i),v}(\rho) - \eta L^{(i),k-1}(\widehat{f}^{(i),v}, \tau^{1:k-1}) \right] \\
&\geq \max_{\phi_i} \mathbb{E}_{v \sim \pi^k} \left[ \widehat{V}^{(i),\phi_i(v)}(f^*) \right] \\
&= \max_{\phi_i} \mathbb{E}_{v \sim \pi^k} \left[ V_{f^*}^{(i),\phi_i(v)}(\rho) - \eta L^{(i),k-1}(f^*, \tau^{1:k-1}) \right].
\end{aligned} \tag{E.23}
$$

The second equality holds by the property of CE. Thus, we have

$$
\begin{aligned}
\max_{\phi_i} \mathbb{E}_{v \sim \pi^k} &[V_{f^*}^{(i),\phi_i(v)}(\rho)] - \mathbb{E}_{v \sim \pi^k}[V_{\widehat{f}^{(i),v}}^{(i),v}(\rho)] \\
&\leq \eta L^{(i),k-1}(f^*, \tau^{1:k-1}) - \eta \mathbb{E}_{v \sim \pi^k} L^{(i),k-1}(\widehat{f}^{(i),v}, \tau^{1:k-1}).
\end{aligned} \tag{E.24}
$$

Hence, combining with (E.23) and (E.24), we can upper bound the regret of the agent $i$ at $k$-th episode as

$$\max_{\phi_i} \mathbb{E}_{\upsilon \sim \pi^k} \left[ V^{(i), \phi_i(\upsilon^{(i)}) \times \upsilon^{(-i)}}(\rho) \right] - \mathbb{E}_{\upsilon \sim \pi^k} \left[ V^{(i), \upsilon}(\rho) \right]$$

$$= \max_{\phi_i} \mathbb{E}_{\upsilon \sim \pi^k} \left[ V^{(i), \phi_i(\upsilon)}_{f^{(i)}, \phi_i(\upsilon)}(\rho) \right] - \mathbb{E}_{\upsilon \sim \pi^k} \left[ V^{(i), \upsilon}(\rho) \right]$$

$$= \max_{\phi_i} \mathbb{E}_{\upsilon \sim \pi^k} \left[ V^{(i), \phi_i(\upsilon)}_{f^{(i)}, \phi_i(\upsilon)}(\rho) \right] - \mathbb{E}_{\upsilon \sim \pi^k} \left[ V^{(i), \upsilon}_{\widehat{f}^{(i)}, \upsilon}(\rho) \right] + \mathbb{E}_{\upsilon \sim \pi^k} \left[ V^{(i), \upsilon}_{\widehat{f}^{(i)}, \upsilon}(\rho) \right] - \mathbb{E}_{\upsilon \sim \pi^k} \left[ V^{(i), \upsilon}(\rho) \right]$$

$$\leq \eta \varepsilon_{\text{conc}} - \eta \mathbb{E}_{\upsilon \sim \pi^k} L^{(i), k-1}(\widehat{f}^{(i), \upsilon}, \upsilon, \tau^{1:k-1}) + \mathbb{E}_{\upsilon \sim \pi^k} \left[ V^{(i), \upsilon}_{\widehat{f}^{(i)}, \upsilon}(\rho) \right] - \mathbb{E}_{\upsilon \sim \pi^k} \left[ V^{(i), \upsilon}(\rho) \right].$$

The rest of the proof is the same as NE/CCE after (E.15). $\square$

### E.3 SAMPLE COMPLEXITY RESULTS

we can utilize the standard online-to-batch techniques to transform the regret result in Theorem 3.1 into the sample complexity result.

**Corollary E.4.** Under the same setting as in Theorem 3.1, with probability at least $1 - \delta$, when $K \geq \widetilde{\mathcal{O}} \left( \left( n^2 H^2 + n^2 d_{\text{MADC}}^2 \Upsilon_{\mathcal{F}, \delta}^2 \right) \cdot \varepsilon^{-2} \right)$, if we output the mixture policy $\pi_{\text{out}} = \text{Unif}(\{\pi^k\}_{k \in [K]})$, the output policy $\pi_{\text{out}}$ is a $\varepsilon$-{NE, CCE, CE}.

*Proof.* See §F.9 for the proof. $\square$

Corollary E.4 shows that MAMEX is sample-efficient for learning all three equilibria of general-sum MGs under general function approximation.

## F PROOF OF THEOREMS AND LEMMAS

### F.1 PROOF OF THEOREM D.4

*Proof.* The proof follows Proposition 3 in Dann et al. (2021). First, we provide the following lemma in Dann et al. (2021).

**Lemma F.1.** For any positive real number sequence $x_1, \cdots, x_n$, we have

$$\frac{\sum_{i=1}^n x_i}{\sqrt{\sum_{i=1}^n i x_i^2}} \leq \sqrt{1 + \log n}.$$

Now denote $E_\varepsilon = \dim_{\text{MABE}}(\mathcal{F}, \varepsilon)$. We fix $i \in [n]$, and ignore both $h$ and $i$ for simplicity. Also denote $\widehat{e}_h^{s,k} = \mathbb{E}_{\pi_s}[\phi_t]$ and $e_h^{s,k} = \widehat{e}_h^{s,k} \cdot \mathbb{I}\{\widehat{e}_h^{s,k} > \varepsilon\}$, where $\phi_t = (I - \mathcal{T}_h^{(i), \pi^k}) f_h \in \mathcal{F}_h^{(i)}$. We initialize $K$ buckets $B_h^0, \cdots, B_h^{K-1}$, and we want to add element $e_h^{k,k}$ for $k \in [K]$ into these buckets one by one. The rule for adding elements is as follows: If $e_h^{k,k} = 0$, we do not add it to any buckets. Otherwise we go through all buckets from $B_h^0$ to $B_h^{K-1}$, and add $e_h^{k,k}$ to $B_h^i$ whenever

$$\sum_{s \leq t-1, s \in B_h^i} (e_h^{s,k})^2 < (e_h^{k,k})^2.$$

Now assume we add $e_h^{k,k}$ into the bucket $b_h^k$. Then, for all $1 \leq i \leq b_h^k - 1$, we have $(e_h^{k,k})^2 \leq \sum_{s \leq k-1, s \in B_h^i} (e_h^{s,k})^2$. Thus,

$$\sum_{k=1}^K \sum_{s=1}^k (e_h^{s,k})^2 \geq \sum_{k=1}^K \sum_{0 \leq i \leq b_h^k - 1} \sum_{s \leq t-1, s \in B_h^i} (e_h^{s,k})^2 \geq \sum_{k=1}^K b_h^k (e_h^{k,k})^2. \tag{F.1}$$

Now note that by the definition of $\varepsilon$-independent sequence, for the measures in $B_h^i$ $\{\pi^{k_1}, \cdots, \pi^{k_j}\}$, $\pi^k$ is a $\varepsilon'$-independent from all predecessors $\pi^{k_1}, \cdots, \pi^{k_{j-1}}$ such that $\varepsilon' > \varepsilon$. (We can choose

$\varepsilon' = e_h^{k,k} - c$ for enough small $c$ such that $\sqrt{\sum_{s \leq t-1, s \in B_h^i}(e_h^{s,k})^2} \leq \varepsilon'$ and $\varepsilon' > \varepsilon$ by $e_h^{k,k} > \varepsilon$.)
Thus, from the definition of BE dimension, the size of each bucket cannot exceed $E_\varepsilon$. Now by Jensen's inequality, we can get

$$\sum_{k=1}^{K} b_h^k (e_h^{k,k})^2 = \sum_{i=1}^{K-1} i \cdot \sum_{s \in B_h^i}(e_h^{s,s})^2 \geq \sum_{i=1}^{K-1} i|B_h^i| \left(\sum_{s \in B_h^i} \frac{e_h^{s,s}}{|B_h^i|}\right)^2 \geq \sum_{i=1}^{K-1} iE_\varepsilon \left(\sum_{s \in B_h^i} \frac{e_h^{s,s}}{E_\varepsilon}\right)^2,$$
(F.2)

where the last inequality uses the fact that $|B_h^i| \leq E_\varepsilon$. Let $x_i = \sum_{s \in B_h^i}(e_h^{s,s})$. By Lemma F.1, we have

$$\sum_{i=1}^{K-1} iE_\varepsilon \left(\sum_{s \in B_h^i} \frac{e_h^{s,s}}{E_\varepsilon}\right)^2 = \frac{1}{E_\varepsilon} \sum_{i=1}^{K-1} i \cdot \left(\sum_{s \in B_h^i} e_h^{s,s}\right)^2 \geq \frac{1}{E_\varepsilon(1 + \log K)} \left(\sum_{s \in [K] \setminus B_h^0} e_h^{s,s}\right)^2.$$
(F.3)

Hence, combining (F.1), (F.2) and (F.3), we can get

$$\sum_{s \in [K] \setminus B_h^0} e_h^{s,s} \leq \left(E_\varepsilon(1 + \log K) \sum_{k=1}^{K} b_h^k (e_h^{k,k})^2\right)^{1/2}$$

$$\leq \left(E_\varepsilon(1 + \log K) \sum_{k=1}^{K} \sum_{s=1}^{k}(e_h^{s,k})^2\right)^{1/2}.$$

Now by the definition $e_h^{s,k} = \hat{e}_h^{s,k} \cdot \mathbb{I}\{\hat{e}_h^{s,k} > \varepsilon\}$ and the fact that $|B_h^0| \leq E_\varepsilon$, we can have

$$\sum_{h=1}^{H} \sum_{k=1}^{K} \hat{e}_h^{k,k} \leq HK\varepsilon + \sum_{h=1}^{H} \sum_{k=1}^{K} e_h^{k,k}$$

$$\leq HK\varepsilon + \min\{HE_\varepsilon, HK\} + \sum_{h=1}^{H} \sum_{s \in [K] \setminus B_h^0} e_h^{s,s}.$$

Then, by (F.3), we can further bounded it by

$$\sum_{h=1}^{H} \sum_{k=1}^{K} \hat{e}_h^{k,k} \leq HK\varepsilon + \min\{HE_\varepsilon, HK\} + \sum_{h=1}^{H} \left(E_\varepsilon(1 + \log K) \sum_{k=1}^{K} \sum_{s=1}^{k-1}(e_h^{s,k})^2\right)^{1/2}$$

$$\leq HK\varepsilon + \min\{HE_\varepsilon, HK\} + \left(E_\varepsilon H(1 + \log K) \sum_{h=1}^{H} \sum_{k=1}^{K} \sum_{s=1}^{k-1}(e_h^{s,k})^2\right)^{1/2}. \quad \text{(F.4)}$$

The last inequality uses the Jensen's inequality Now we can use a similar technique in Xie et al. (2021). Define $(r_h')^{(i)}(s,a) = f_h^k(s,a) - \mathbb{E}_{s' \sim \mathbb{P}_h(\cdot|s,a)} \langle f_{h+1}^k(s',\cdot), \pi_{h+1}^k(\cdot \mid s') \rangle$. Then, we have

$$\mathbb{E}_{s_1 \sim \rho}\left[f_1^k(s_1, \pi_1^k(s_1))\right] = \mathbb{E}_{\pi^k}\left[\sum_{h=1}^{H}\left(f_h^k(s_h, \pi_h^k(s_h)) - f_{h+1}^k(s_{h+1}, \pi_{h+1}^k(s_{h+1}))\right)\right]$$

$$= \mathbb{E}_{\pi^k}\left[\sum_{h=1}^{H}\left(f_h^k(s_h, \pi_h^k(s_h)) - \mathbb{E}_{s' \sim \mathbb{P}_h(\cdot|s,a)} \langle f_{h+1}^k(s',\cdot), \pi_{h+1}^k(\cdot \mid s') \rangle\right)\right]$$

$$= \mathbb{E}_{\pi^k}\left[\sum_{h=1}^{H}(r_h')^{(i)}(s,a)\right].$$

Hence, we can rewrite the regret of the $k$-th episode as

$$\mathbb{E}_{s_1 \sim \rho}\left[(f_1^k(s_1, \pi_1^k(s_1)) - V^{(i), \pi^k}(s_1))\right] = \mathbb{E}_{\pi^k}\left[\sum_{h=1}^{H}((r_h')^{(i)}(s,a) - r_h^{(i)}(s,a))\right]. \quad \text{(F.5)}$$

The last inequality uses the fact that Then, substitute into the definition of $(r'_h)^{(i)}$, we can get

$$\mathbb{E}_{s_1 \sim \rho} \left[ (f_1^k(s_1, \pi_1^k(s_1)) - V^{(i),\pi^k}(s_1)) \right]$$

$$= \mathbb{E}_{\pi^k} \left[ \sum_{h=1}^{H} (f_h^k(s,a) - \mathbb{E}_{s' \sim \mathbb{P}_h(\cdot|s,a)} \langle f_{h+1}^k(s', \cdot), \pi_{h+1}^k(\cdot \mid s') \rangle - r_h^{(i)}(s,a)) \right]$$

$$= \mathbb{E}_{\pi^k} \left[ \sum_{h=1}^{H} (I - \mathcal{T}_h^{(i),\pi^k})(f_h^k) \right] = \sum_{h=1}^{H} \widehat{e}_h^{k,k},$$

where the first equality holds by the definition of $M_k$, the second equality holds by decomposing the value function to the expected cumulative sum of the reward function, and the last equality is derived by the definition of $\widehat{e}_h^{k,k}$. Now we can get

$$\mathrm{Reg}(K) \leq \sum_{k=1}^{K} \sum_{h=1}^{H} \widehat{e}_h^{k,k}$$

$$\leq HK\varepsilon + \min\{HE_\varepsilon, HK\} + \left( (E_\varepsilon H \cdot 2\log K) \sum_{h=1}^{H} \sum_{k=1}^{K} \sum_{s=1}^{k-1} (e_h^{s,k})^2 \right)^{1/2}. \tag{F.6}$$

The last inequality holds by (F.4). Now by the definition $e_h^{s,k} \leq \widehat{e}_h^{s,k} = \mathbb{E}_{\pi^s}[(I - \mathcal{T}_h^{(i),\pi^k})(f_h)]$ and the basic inequality $\sqrt{ab} \leq \mu a + b/\mu$ for $\mu > 0$, we can derive

$\mathrm{Reg}(K)$

$$\leq HK\varepsilon + \min\{HE_\varepsilon, HK\} + \mu \cdot (E_\varepsilon H \cdot 2\log K) + \frac{1}{\mu} \sum_{h=1}^{H} \sum_{k=1}^{K} \sum_{s=1}^{k-1} \left( \mathbb{E}_{\pi^s} \left[ (I - \mathcal{T}_h^{(i),\pi^k})(f_h) \right] \right)^2$$

$$\leq HK\varepsilon + \min\{HE_\varepsilon, HK\} + \mu \cdot (E_\varepsilon H \cdot 2\log K) + \frac{1}{\mu} \sum_{h=1}^{H} \sum_{k=1}^{K} \sum_{s=1}^{k-1} \mathbb{E}_{\pi^s} \left[ \left( (I - \mathcal{T}_h^{(i),\pi^k})(f_h) \right)^2 \right].$$

The last inequality holds by $(\mathbb{E}[X])^2 \leq \mathbb{E}[X^2]$. Thus, by choosing $\varepsilon = 1/K$, we can derive

$$\mathrm{Reg}(K) \leq H + HE_{1/K} + \mu(E_{1/K} H \cdot 2\log K) + \frac{1}{\mu} \sum_{h=1}^{H} \sum_{k=1}^{K} \sum_{s=1}^{k-1} \mathbb{E}_{\pi^s} \left[ \left( (I - \mathcal{T}_h^{(i),\pi^k})(f_h) \right)^2 \right]$$

$$\leq 6d_{\mathrm{MADC}} H + \mu d_{\mathrm{MADC}} + \frac{1}{\mu} \sum_{h=1}^{H} \sum_{k=1}^{K} \sum_{s=1}^{k-1} \mathbb{E}_{\pi^s} \left[ \left( (I - \mathcal{T}_h^{(i),\pi^k})(f_h) \right)^2 \right],$$

where $d_{\mathrm{MADC}} = \max\{2E_{1/K} H \log K, 1\} = \mathcal{O}(2\mathrm{dim}_{\mathrm{MABE}}(\mathcal{F}, 1/K) H \log K)$. The last inequality uses the fact that $\log K \geq 1$ and $H \geq 1$. $\qquad\square$

### F.2 Proof of Theorem D.5

*Proof.* For any policy $\pi$ and $i \in [n]$, assume $\delta_{z_1}, \cdots, \delta_{z_m}$ is an $\varepsilon$-independent sequence with respect to $\bigcup_{\pi \in \Pi^{\mathrm{pur}}} \mathcal{F}_h^{(i),\pi} = \bigcup_{\pi \in \Pi^{\mathrm{pur}}} (I - \mathcal{T}^{(i),\pi})\mathcal{F}$, where $\delta_{z_1}, \delta_{z_2}, \cdots, \delta_{z_m} \in \mathcal{D}_\Delta$, i.e. $\delta_{z_i}$ is a Dirichlet probability measure over $\mathcal{S} \times \mathcal{A}$ that $\delta_{z_i} = \delta_{(s,a)}(\cdot)$. Then, for each $j \in [m]$, there exist function $f^j \in \mathcal{F}^{(i)}$ and policy $\pi^j \in \Pi$ such that $|(I - \mathcal{T}^{(i),\pi^j}) f_j(z_j)| > \varepsilon$ and $\sqrt{\sum_{p=1}^{j-1} |(f_h^p - \mathcal{T}^{(i),\pi^p} f_{h+1}^p)(z_p)|^2} \leq \varepsilon$. Define $g_h^j = \mathcal{T}^{(i),\pi^j} f_{h+1}^j$, by Assumption 2.3, we have $g_h^j \in \mathcal{F}_h^{(i)} \subseteq \mathcal{F}_h$. Thus, $|(f_h^j - g_h^j)(z_j)| > \varepsilon$ and $\sqrt{\sum_{p=1}^{j} |(f_h^p - g_h^p)(z_p)|^2} < \varepsilon$. Thus, by the definition of eluder dimension, we have $m \leq \mathrm{dim}_{\mathrm{E}}(\mathcal{F}_h, \varepsilon)$. Hence, for all $i$ and policy $\pi$,

$$\mathrm{dim}_{\mathrm{E}}(\mathcal{F}_h, \varepsilon) \geq m \geq \max_{h \in [H]} \max_{i \in [n]} \mathrm{dim}_{\mathrm{DE}} \left( \bigcup_{\pi \in \Pi^{\mathrm{pur}}} \mathcal{F}_h^{(i),\pi}, \mathcal{D}_{h,\Delta}, \varepsilon \right),$$

which concludes the proof. $\qquad\square$

### F.3 PROOF OF THEOREM D.8

*Proof.* First, by the elliptical potential lemma introduced in Lemma G.3, if we define $\Lambda_{k,h} = \varepsilon I + \sum_{s=1}^{k-1} x_{k,h} x_{k,h}^T$, for any $\{x_{k,h}\}_{k=1}^K \in \mathcal{X}_h$ we have

$$\sum_{k=1}^K \sum_{h=1}^H \min\left\{1, \|x_{k,h}\|_{\Lambda_{k,h}^{-1}}^2\right\} \leq \sum_{h=1}^H 2\log\det\left(I + \frac{1}{\varepsilon}\sum_{k=1}^K x_{k,h} x_{k,h}^T\right) = 2\gamma_K(\varepsilon, \mathcal{X}). \quad (\text{F.7})$$

Now denote $\Sigma_{k,h} = \varepsilon I + \sum_{s=1}^{k-1} X_h(\pi^s) X_h(\pi^s)^T$. Similar to Section F.1, define $(r_h')^{(i)}(s,a) = f_h^k(s,a) - \mathbb{E}_{s'\sim\mathbb{P}_h(\cdot|s,a)}\langle f_{h+1}^k(s',\cdot), \pi_{h+1}^k(\cdot \mid s')\rangle \in [-1,1]$, then we can have

$$\mathbb{E}_{s_1\sim\rho}\left[f_1^k(s_1, \pi_1^k(s_1)) - V^{(i),\pi^k}(s_1)\right] = \mathbb{E}_{\pi^k}\left[\sum_{h=1}^H ((r_h')^{(i)}(s,a) - r_h^{(i)}(s,a))\right].$$

Then, we can substitute the definition of $(r_h')^{(i)}$ and derive

$$\mathbb{E}_{s_1\sim\rho}\left[f_1^k(s_1, \pi_1^k(s_1)) - V^{(i),\pi^k}(s_1)\right]$$

$$= \mathbb{E}_{\pi^k}\left[\sum_{h=1}^H (f_h^k(s,a) - \mathbb{E}_{s'\sim\mathbb{P}_h(\cdot|s,a)}\langle f_{h+1}^k(s',\cdot), \pi_{h+1}^k(\cdot \mid s')\rangle - r_h^{(i)}(s,a))\right]$$

$$= \sum_{h=1}^H \min\left\{\left|\langle W_h^{(i)}(f^k, \pi^k) - W_h^{(i)}(f^{\mu^{(i),\pi^k}}, \mu^{(i),\pi^k}), X_h(\pi^k)\rangle_\mathcal{V}\right|, 2R\right\}.$$

Then, by $\min\{x, 2R\} \leq 2R\min\{x, 1\}$, we have

$$\text{Reg}(K) = \sum_{k=1}^K \mathbb{E}_{s_1\sim\rho}\left[(f_1^k(s_1, \pi_1^k(s_1)) - V^{(i),\pi^k}(s_1))\right]$$

$$\leq 2R\sum_{k=1}^K \sum_{h=1}^H \min\left\{\left|\langle W_h^{(i)}(f^k, \pi^k) - W_h^{(i)}(f^{\mu^{(i),\pi^k}}, \mu^{(i),\pi^k}), X_h(\pi^k)\rangle_\mathcal{V}\right|, 1\right\}$$

$$= 2R\sum_{k=1}^K \sum_{h=1}^H \min\left\{\left|\langle W_h^{(i)}(f^k, \pi^k) - W_h^{(i)}(f^{\mu^{(i),\pi^k}}, \mu^{(i),\pi^k}), X_h(\pi^k)\rangle_\mathcal{V}\right|, 1\right\}$$

$$\cdot\left(\mathbb{I}\left\{\left\|X_h(\pi^k)\right\|_{\Sigma_{k,h}^{-1}} \leq 1\right\} + \mathbb{I}\left\{\left\|X_h(\pi^k)\right\|_{\Sigma_{k,h}^{-1}} > 1\right\}\right). \quad (\text{F.8})$$

The last inequality is because $1 = \mathbb{I}\{\mathcal{E}\} + \mathbb{I}\{\neg E\}$ for any event. Now we decompose the (F.8) into two terms $A + B$, where

$$A = 2R\sum_{k=1}^K \sum_{h=1}^H \min\left\{\left|\langle W_h^{(i)}(f^k, \pi^k) - W_h^{(i)}(f^{\mu^{(i),\pi^k}}, \mu^{(i),\pi^k}), X_h(\pi^k)\rangle_\mathcal{V}\right|, 1\right\} \cdot \mathbb{I}\left\{\left\|X_h(\pi^k)\right\|_{\Sigma_{k,h}^{-1}} \leq 1\right\},$$

$$\quad (\text{F.9})$$

$$B = 2R\sum_{k=1}^K \sum_{h=1}^H \min\left\{\left|\langle W_h^{(i)}(f^k, \pi^k) - W_h^{(i)}(f^{\mu^{(i),\pi^k}}, \mu^{(i),\pi^k}), X_h(\pi^k)\rangle_\mathcal{V}\right|, 1\right\} \cdot \mathbb{I}\left\{\left\|X_h(\pi^k)\right\|_{\Sigma_{k,h}^{-1}} > 1\right\}.$$

$$\quad (\text{F.10})$$

Now we bound $A$ and $B$ respectively. For $A$, we can use Cauchy's inequality and get

$$A \leq 2R\sum_{k=1}^K \sum_{h=1}^H \left\|W_h^{(i)}(f^k, \pi^k) - W_h^{(i)}(f^{\mu^{(i),\pi^k}}, \mu^{(i),\pi^k})\right\|_{\Sigma_{k,h}} \cdot \left\|X_h(\pi^k)\right\|_{\Sigma_{k,h}^{-1}} \cdot \mathbb{I}\left\{\left\|X_h(\pi^k)\right\|_{\Sigma_{k,h}^{-1}} \leq 1\right\}$$

$$\leq 2R\sum_{k=1}^K \sum_{h=1}^H \left\|W_h^{(i)}(f^k, \pi^k) - W_h^{(i)}(f^{\mu^{(i),\pi^k}}, \mu^{(i),\pi^k})\right\|_{\Sigma_{k,h}} \cdot \min\left\{\left\|X_h(\pi^k)\right\|_{\Sigma_{k,h}^{-1}}, 1\right\}.$$

$$\quad (\text{F.11})$$

The first inequality holds by Cauchy's inequality that $|\langle X, Y\rangle| \leq \|X\|_\Sigma \|Y\|_{\Sigma^{-1}}$.

Now by the definition $\Sigma_{k,h} = \varepsilon I + \sum_{s=1}^{k-1} X_h(\pi^s) X_h(\pi^s)^T$, we expand the term $\|W_h^{(i)}(f^k, \pi^k) - W_h^{(i)}(f^{\mu^{(i),\pi^k}}, \mu^{(i),\pi^k})\|_{\Sigma_{k,h}}$ as

$$
\left\| W_h^{(i)}(f^k, \pi^k) - W_h^{(i)}(f^{\mu^{(i),\pi^k}}, \mu^{(i),\pi^k}) \right\|_{\Sigma_{k,h}}
$$

$$
= \left[ \varepsilon \cdot \left\| W_h^{(i)}(f^k, \pi^k) - W_h^{(i)}(f^{\mu^{(i),\pi^k}}, \mu^{(i),\pi^k}) \right\|_2^2 + \sum_{s=1}^{k-1} \left| \langle W_h^{(i)}(f^k, \pi^k) - W_h^{(i)}(f^{\mu^{(i),\pi^k}}, \mu^{(i),\pi^k}), X_h(\pi^s) \rangle \right|^2 \right]^{1/2}
$$

$$
\leq 2\sqrt{\varepsilon} B_W + \left[ \sum_{s=1}^{k-1} \left| \langle W_h^{(i)}(f^k, \pi^k) - W_h^{(i)}(f^{\mu^{(i),\pi^k}}, \mu^{(i),\pi^k}), X_h(\pi^s) \rangle \right|^2 \right]^{1/2} .
$$

The last inequality holds by $\sqrt{a+b} \leq \sqrt{a} + \sqrt{b}$. Then, we can get

$$
\sum_{k=1}^{K} \sum_{h=1}^{H} \|W_h^{(i)}(f^k, \pi^k) - W_h^{(i)}(f^{\mu^{(i),\pi^k}}, \mu^{(i),\pi^k})\|_{\Sigma_{k,h}} \min\left\{ \|X_h(\pi^k)\|_{\Sigma_{k,h}^{-1}}, 1 \right\}
$$

$$
\leq \sum_{k=1}^{K} \sum_{h=1}^{H} \left( 2\sqrt{\varepsilon} B_W + \left[ \sum_{s=1}^{k-1} |\langle W_h^{(i)}(f^k, \pi^k) - W_h^{(i)}(f^{\mu^{(i),\pi^k}}, \mu^{(i),\pi^k}), X_h(\pi^s) \rangle|^2 \right]^{1/2} \right)
$$

$$
\cdot \min\left\{ \|X_h(\pi^k)\|_{\Sigma_{k,h}^{-1}}, 1 \right\} \leq A_1 + A_2. \tag{F.12}
$$

where $A_1$ and $A_2$ are defined as follows:

$$
A_1 = \left( \sum_{k=1}^{K} \sum_{h=1}^{H} 4\varepsilon B_W^2 \right)^{1/2} \cdot \left( \sum_{k=1}^{K} \sum_{h=1}^{H} \min\left\{ \|X_h(\pi^k)\|_{\Sigma_{k,h}^{-1}}^2, 1 \right\} \right)^{1/2}
$$

$$
A_2 = \left( \sum_{k=1}^{K} \sum_{h=1}^{H} \sum_{s=1}^{k-1} |\langle W_h^{(i)}(f^k, \pi^k) - W_h^{(i)}(f^{\mu^{(i),\pi^k}}, \mu^{(i),\pi^k}), X_h(\pi^s) \rangle|^2 \right)^{1/2}
$$

$$
\cdot \left( \sum_{k=1}^{K} \sum_{h=1}^{H} \min\{ \|X_h(\pi^k)\|_{\Sigma_{k,h}^{-1}}^2, 1 \} \right)^{1/2} ,
$$

Now we bound $A_1$ and $A_2$ respectively. First, for $A_1$, using (F.7), we have

$$
A_1 \leq \sqrt{4\varepsilon K H B_W^2 \cdot 2\gamma_K(\varepsilon, \mathcal{X})}.
$$

Then, for $A_2$, we have

$$
A_2 = \left( \sum_{k=1}^{K} \sum_{h=1}^{H} \sum_{s=1}^{k-1} |\langle W_h^{(i)}(f^k, \pi^k) - W_h^{(i)}(f^{\mu^{(i),\pi^k}}, \mu^{(i),\pi^k}), X_h(\pi^s) \rangle|^2 \right)^{1/2} \sqrt{2\gamma_K(\varepsilon, \mathcal{X})}
$$

$$
= \left( \sum_{k=1}^{K} \sum_{s=1}^{k-1} \ell^{(i),s}(f^k, \pi^k) \right)^{1/2} \sqrt{2\gamma_K(\varepsilon, \mathcal{X})}.
$$

The equality holds by the definition of $\ell^{(i),s}$ and the definition of multi-agent bilinear class (D.3).

Then, since $\sqrt{ab} \leq a\mu + b/\mu$ for any $\mu > 0$, we can further derive

$$
A_2 \leq 2R\mu \cdot 2\gamma_K(\varepsilon, \mathcal{X}) + \frac{1}{2R\mu} \sum_{k=1}^{K} \sum_{s=1}^{k-1} \ell^{(i),s}(f^k, \pi^k).
$$

Now by adding $A_1$ and $A_2$ and combining with (F.11) and (F.12), we can finally get

$$
A \leq 2R(A_1 + A_2) \leq \sqrt{4\varepsilon K H B_W^2 \cdot 8R\gamma_K(\varepsilon, \mathcal{X})} + \mu \cdot 8R^2 \gamma_K(\varepsilon, \mathcal{X}) + \frac{1}{\mu} \sum_{k=1}^{K} \sum_{s=1}^{k-1} \ell^{(i),s}(f^k, \pi^k)
$$

$$
\leq 32R B_W^2 \varepsilon H K + \gamma_K(\varepsilon, \mathcal{X}) + \mu \cdot 8R^2 \gamma_K(\varepsilon, \mathcal{X}) + \frac{1}{\mu} \sum_{k=1}^{K} \sum_{s=1}^{k-1} \ell^{(i),s}(f^k, \pi^k).
$$

Now we have complete the bound of $A$. For $B$, by (F.7), since $\mathbb{I}\{x > 1\} \leq \min\{1, x^2\}$, we know that

$$\sum_{k=1}^{K} \sum_{h=1}^{H} \mathbb{I}\left\{\|X_h(\pi^k)\|_{\Sigma_{k,h}^{-1}} > 1\right\} \leq \sum_{k=1}^{K} \sum_{h=1}^{H} \min\left\{1, \|x_{k,h}\|_{\Sigma_{k,h}^{-1}}^2\right\} \leq 2\gamma_K(\varepsilon, \mathcal{X}). \qquad \text{(F.13)}$$

Thus, by the definition of $B$ in (F.10), we can derive

$$B \leq 2\sum_{k=1}^{K} \sum_{h=1}^{H} \mathbb{I}\left\{\|X_h(\pi^k)\|_{\Sigma_{k,h}^{-1}} > 1\right\} \leq 2\gamma_K(\varepsilon, \mathcal{X}).$$

Now note that $\mathrm{Reg}(K) \leq A + B$, then by choosing $\varepsilon = 1/32RKB_W^2$ and $d_{\mathrm{MADC}} = \max\{1, 8R^2\gamma_K(\varepsilon, \mathcal{X})\}$ we can derive

$$\mathrm{Reg}(K) \leq A + B$$

$$\leq 32RB_W^2 \varepsilon HK + 3\gamma_K(\varepsilon, \mathcal{X}) + \mu \cdot 8R^2\gamma_K(\varepsilon, \mathcal{X}) + \frac{1}{\mu}\sum_{k=1}^{K}\sum_{s=1}^{k-1} \ell^{(i),s}(f^k, \pi^k)$$

$$= H + 3\gamma_K(\varepsilon, \mathcal{X}) + \mu \cdot 8R^2\gamma_K(\varepsilon, \mathcal{X}) + \frac{1}{\mu}\sum_{k=1}^{K}\sum_{s=1}^{k-1} \ell^{(i),s}(f^k, \pi^k)$$

$$\leq 6d_{\mathrm{MADC}}H + \mu d_{\mathrm{MADC}} + \frac{1}{\mu}\sum_{k=1}^{K}\sum_{s=1}^{k-1} \ell^{(i),s}(f^k, \pi^k).$$

The last inequality uses the fact that $d_{\mathrm{MADC}} \geq 1, H \geq 1$. Hence, we complete the proof.

$\square$

## F.4 Proof of Theorem D.12

*Proof.* In this subsection, we give a detailed proof of Theorem D.12. First, similar to the performance difference lemma in Jiang et al. (2017), we have

$$\mathbb{E}_{s_1 \sim \rho}\left[V_{1,f}^{(i),\pi^k}(s_1) - V_1^{(i),\pi^k}(s_1)\right]$$

$$= \mathbb{E}_{\pi^k}\left[Q_{1,f}^{(i),\pi^k}(s_1, a_1)\right] - \mathbb{E}_{\pi^k}\left[\sum_{h=1}^{H} r_h^{(i)}(s_h, a_h)\right]$$

$$= \mathbb{E}_{\pi^k}\left[\sum_{h=1}^{H}\left(Q_{h,f}^{(i),\pi^k}(s_h, a_h) - r_h^{(i)}(s_h, a_h) - Q_{h+1,f}^{(i),\pi^k}(s_{h+1}, a_{h+1})\right)\right]. \qquad \text{(F.14)}$$

The last equality holds by splitting the term. Now, since

$$\mathbb{E}_{\pi^k}\left[Q_{h+1,f}^{(i),\pi^k}(s_{h+1}, a_{h+1})\right] = \mathbb{E}_{\pi^k}\left[V_{h+1,f}^{(i),\pi^k}(s_{h+1})\right] = \mathbb{E}_{\pi^k}\left[\mathbb{E}_{s_{h+1} \sim \mathbb{P}_{h,f^*}(\cdot|s_h, a_h)}\left[V_{h+1,f}^{(i),\pi^k}(s_{h+1})\right]\right],$$

$$\text{(F.15)}$$

we can rewrite (F.14) as

$$\mathbb{E}_{s_1 \sim \rho}\left[V_{1,f}^{(i),\pi^k}(s_1) - V_1^{(i),\pi^k}(s_1)\right]$$

$$= \mathbb{E}_{\pi^k}\left[\sum_{h=1}^{H}\left(Q_{h,f}^{(i),\pi^k}(s_h, a_h) - r_h^{(i)}(s_h, a_h) - \mathbb{E}_{s_{h+1} \sim \mathbb{P}_{h,f^*}(\cdot|s_h, a_h)} V_{h+1,f}^{(i),\pi^k}(s_{h+1})\right)\right]$$

$$= \sum_{k=1}^{K} \mathbb{E}_{\pi^k}\left[\sum_{h=1}^{H}(\mathbb{E}_{s_{h+1} \sim \mathbb{P}_{h,f^k}(\cdot|s_h, a_h)} - \mathbb{E}_{s_{h+1} \sim \mathbb{P}_{h,f^*}(\cdot|s_h, a_h)})\left[V_{h+1,f^k}^{(i),\pi^k}(s_{h+1})\right]\right]. \qquad \text{(F.16)}$$

Then, combining (F.16) and the definition of multi-agent witness rank (D.7), we can derive

$$\sum_{k=1}^{K} \mathbb{E}_{s_1 \sim \rho} \left[ V_{1,f^k}^{(i),\pi^k}(s_1) - V_1^{(i),\pi^k}(s_1) \right]$$

$$\leq \sum_{k=1}^{K} \sum_{h=1}^{H} \min \left\{ R, \frac{1}{\kappa_{\text{wit}}} |\langle W_h(f^k), X_h(\pi^k) \rangle| \right\}$$

$$\leq \sum_{k=1}^{K} \sum_{h=1}^{H} \min \left\{ R, \frac{1}{\kappa_{\text{wit}}} |\langle W_h(f^k), X_h(\pi^k) \rangle| \right\} \left( \mathbb{I}\left\{ \|X_h(\pi^k)\|_{\Sigma_{k,h}^{-1}} \leq 1 \right\} + \mathbb{I}\left\{ \|X_h(\pi^k)\|_{\Sigma_{k,h}^{-1}} \geq 1 \right\} \right). \tag{F.17}$$

Now note that

$$\sum_{k=1}^{K} \min \left\{ 1, \|X_h(\pi^k)\|_{\Sigma_{k,h}^{-1}}^2 \right\} \leq 2d \log \left( \frac{\varepsilon + K}{\varepsilon} \right) \triangleq \mathcal{D}(\varepsilon).$$

and $\mathbb{I}\{x > 1\} \leq \min\{1, x^2\}$, we can derive

$$\sum_{k=1}^{K} \sum_{h=1}^{H} \mathbb{I}\{ \|X_h(\pi^k)\|_{\Sigma_{k,h}^{-1}} > 1 \} \leq \mathcal{D}(\varepsilon) H. \tag{F.18}$$

Then, combining (F.17) and (F.18), we can get

$$\sum_{k=1}^{K} \mathbb{E}_{s_1 \sim \rho} \left[ V_{1,f^k}^{(i),\pi^k}(s_1) - V_1^{(i),\pi^k}(s_1) \right]$$

$$\leq R \sum_{k=1}^{K} \sum_{h=1}^{H} \min \left\{ 1, \frac{1}{\kappa_{\text{wit}}} |\langle W_h(f^k), X_h(\pi^k) \rangle| \right\} \cdot \left( \mathbb{I}\{ \|X_h(\pi^k)\|_{\Sigma_{k,h}^{-1}} \leq 1 \} + \mathbb{I}\{ \|X_h(\pi^k)\|_{\Sigma_{k,h}^{-1}} > 1 \} \right)$$

$$\leq R \sum_{k=1}^{K} \sum_{h=1}^{H} \min \left\{ 1, \frac{1}{\kappa_{\text{wit}}} |\langle W_h(f^k), X_h(\pi^k) \rangle| \right\} \cdot \left( \mathbb{I}\{ \|X_h(\pi^k)\|_{\Sigma_{k,h}^{-1}} \leq 1 \} \right) + \mathcal{D}(\varepsilon) H R$$

$$\leq R \underbrace{\sum_{k=1}^{K} \sum_{h=1}^{H} \frac{1}{\kappa_{\text{wit}}} \|W_h(f^k)\|_{\Sigma_{k,h}} \min \left\{ 1, \|X_h(\pi^k)\|_{\Sigma_{k,h}^{-1}}^2 \right\}}_{(A)} + \mathcal{D}(\varepsilon) H R. \tag{F.19}$$

The last inequality uses the Cauchy's inequality $\langle X, Y \rangle \leq \|X\|_A \|Y\|_{A^{-1}}$ and the fact that $x \cdot \mathbb{I}\{x \leq 1\} \leq \min\{1, x^2\}$. Further, by the definition of $\Sigma_{k,h}$, we decompose the first term as

$$(A) \leq \frac{1}{\kappa_{\text{wit}}} \sum_{k=1}^{K} \sum_{h=1}^{H} \left[ \varepsilon \cdot \|W_h(f^k)\|_2^2 + \sum_{s=1}^{k} |\langle W_h(f^k), X_h(\pi^s) \rangle|^2 \right]^{1/2} \min \left\{ 1, \|X_h(\pi^k)\|_{\Sigma_{k,h}^{-1}} \right\}$$

$$\leq \frac{1}{\kappa_{\text{wit}}} \sum_{k=1}^{K} \sum_{h=1}^{H} \left( \sqrt{\varepsilon} B_W + \left[ \sum_{s=1}^{k} |\langle W_h(f^k), X_h(\pi^s) \rangle|^2 \right]^{1/2} \right) \min \left\{ 1, \|X_h(\pi^k)\|_{\Sigma_{k,h}^{-1}} \right\}.$$

The second inequality is derived by the inequality $\|W_h(f^k)\| \leq B_W$ and $\sqrt{a+b} \leq \sqrt{a} + \sqrt{b}$. Now sum over $k \in [K]$ and $h \in [H]$, we can get

$$(A) \leq \sum_{k=1}^{K} \sum_{h=1}^{H} \frac{1}{\kappa_{\text{wit}}} \left( \sqrt{\varepsilon} B_W + \left[ \sum_{s=1}^{k} |\langle W_h(f^k), X_h(\pi^s) \rangle|^2 \right]^{1/2} \right) \min \left\{ 1, \|X_h(\pi^k)\|_{\Sigma_{k,h}^{-1}} \right\}$$

$$\leq \underbrace{\frac{1}{\kappa_{\text{wit}}} \sum_{k=1}^{K} \sum_{h=1}^{H} \sqrt{\varepsilon} B_W \min \left\{ 1, \|X_h(\pi^k)\|_{\Sigma_{k,h}^{-1}} \right\}}_{(X)}$$

$$+ \underbrace{\frac{1}{\kappa_{\text{wit}}} \sum_{k=1}^{K} \sum_{h=1}^{H} \left[ \sum_{s=1}^{k} |\langle W_h(f^k), X_h(\pi^s) \rangle|^2 \right]^{1/2} \min \left\{ 1, \|X_h(\pi^k)\|_{\Sigma_{k,h}^{-1}} \right\}}_{(Y)}. \tag{F.20}$$

First, we try to give an upper bound for (X). By Cauchy's inequality and (F.18), we can derive

$$
(X) \leq \frac{1}{\kappa_{\text{wit}}} \left( \sum_{k=1}^{K} \sum_{h=1}^{H} \varepsilon B_W^2 \right)^{1/2} \left( \sum_{k=1}^{K} \sum_{h=1}^{H} \min\left\{1, \|X_h(\pi^k)\|_{\Sigma_{k,h}^{-1}}^2\right\} \right)^{1/2}
$$

$$
\leq \frac{1}{\kappa_{\text{wit}}} \sqrt{HK\varepsilon B_W^2 \cdot \mathcal{D}(\varepsilon)H} \leq \frac{HK\varepsilon B_W^2}{\kappa_{\text{wit}}^2} + \mathcal{D}(\varepsilon)H. \tag{F.21}
$$

On the other hand, for (Y), we can bound it using Cauchy's inequality that $\sum_{a,b} \sqrt{ab} \leq \sqrt{(\sum_a a) \cdot (\sum_b b)}$,

$$
(Y) \leq \frac{1}{\kappa_{\text{wit}}} \left( \left( \sum_{k=1}^{K} \sum_{h=1}^{H} \sum_{s=1}^{k} |\langle W_h(f^k), X_h(\pi^s) \rangle|^2 \right) \left( \sum_{k=1}^{K} \sum_{h=1}^{H} \min\left\{1, \|X_h(\pi^k)\|_{\Sigma_{k,h}^{-1}}^2\right\} \right) \right)^{1/2}
$$

$$
\leq \frac{1}{\kappa_{\text{wit}}} \sqrt{\mathcal{D}(\varepsilon)H \left( \sum_{k=1}^{K} \sum_{h=1}^{H} \sum_{s=1}^{k} |\langle W_h(f^k), X_h(\pi^s) \rangle|^2 \right)}.
$$

The last inequality holds by the definition of $\mathcal{D}(\varepsilon)$ in F.18. Now by the definition of multi-agent witness rank D.6, we note that

$$
|\langle W_h(f^k), X_h(\pi^s) \rangle|^2
$$

$$
\leq \left( \max_{v \in \mathcal{V}_h} \mathbb{E}_{(s_h, a_h) \sim \pi} [(\mathbb{E}_{s_{h+1} \sim \mathbb{P}_{h,f^k}(\cdot|s_h,a_h)} - \mathbb{E}_{s_{h+1} \sim \mathbb{P}_{h,f^*}(\cdot|s_h,a_h)}) v(s_h, a_h, s_{h+1})] \right)^2
$$

$$
\leq \max_{v \in \mathcal{V}_h} \mathbb{E}_{(s_h, a_h) \sim \pi^s} \left[ \left( (\mathbb{E}_{s_{h+1} \sim \mathbb{P}_{h,f^k}(\cdot|s_h,a_h)} - \mathbb{E}_{s_{h+1} \sim \mathbb{P}_{h,f^*}(\cdot|s_h,a_h)}) v(s_h, a_h, s_{h+1}) \right)^2 \right]
$$

$$
\leq \mathbb{E}_{(s_h, a_h) \sim \pi^s} \left[ \max_{v \in \mathcal{V}_h} \left( (\mathbb{E}_{s_{h+1} \sim \mathbb{P}_{h,f^k}(\cdot|s_h,a_h)} - \mathbb{E}_{s_{h+1} \sim \mathbb{P}_{h,f^*}(\cdot|s_h,a_h)}) v(s_h, a_h, s_{h+1}) \right)^2 \right]
$$

The last two inequalities use Jensen's inequality. Hence, by the definition of total variation distance, we can get

$$
|\langle W_h(f^k), X_h(\pi^s) \rangle|^2 \leq \text{TV} \left( \mathbb{P}_{h,f^k}(\cdot \mid s_h, a_h), \mathbb{P}_{h,f^*}(\cdot \mid s_h, a_h) \right)^2 \tag{F.22}
$$

$$
\leq 2 D_H^2 \left( \mathbb{P}_{h,f^k}(\cdot \mid s_h, a_h), \mathbb{P}_{h,f^*}(\cdot \mid s_h, a_h) \right), \tag{F.23}
$$

where the $\text{TV}(\cdot, \cdot)$ denotes the total variation distance and $D_H$ denotes the Hellinger divergence. The inequality (F.22) holds by the fact that $v(s_h, a_h, s_{h+1}) \in [0, 1]$, and the (F.23) holds by the relationship between TV distance and Hellinger distance. Then, we can substitute the inequality (F.23) and get

$$
(Y) \leq \frac{1}{\kappa_{\text{wit}}} \sqrt{\mathcal{D}(\varepsilon)H \left( \sum_{k=1}^{K} \sum_{h=1}^{H} \sum_{s=1}^{k} \mathbb{E}_{(s_h,a_h) \sim \pi^s} 2 D_H^2 \left( \mathbb{P}_{h,f^k}(\cdot \mid s_h, a_h), \mathbb{P}_{h,f^*}(\cdot \mid s_h, a_h) \right) \right)}
$$

$$
\leq \mu R \cdot \frac{2\mathcal{D}(\varepsilon)H}{\kappa_{\text{wit}}^2} + \frac{1}{\mu R} \left( \sum_{k=1}^{K} \sum_{h=1}^{H} \sum_{s=1}^{k} \mathbb{E}_{(s_h,a_h) \sim \pi^s} D_H^2 \left( \mathbb{P}_{h,f^k}(\cdot \mid s_h, a_h), \mathbb{P}_{h,f^*}(\cdot \mid s_h, a_h) \right) \right)
$$

$$
\tag{F.24}
$$

Hence, combining (F.19), (F.20), (F.21) and (F.24), we can get

$$
\text{Reg}(K) = \sum_{k=1}^{K} \mathbb{E}_{s_1 \sim \rho} \left[ V_{1,f^k}^{(i),\pi^k}(s_1) - V_1^{(i),\pi^k}(s_1) \right]
$$

$$
\leq R \cdot A + \mathcal{D}(\varepsilon)HR
$$

$$
\leq R(X + Y) + \mathcal{D}(\varepsilon)HR
$$

$$
\leq HKR\varepsilon B_W^2 / \kappa_{\text{wit}}^2 + \mathcal{D}(\varepsilon)HR
$$

$$
+ \mu R^2 \cdot \frac{2\mathcal{D}(\varepsilon)H}{\kappa_{\text{wit}}^2} + \frac{1}{\mu} \left( \sum_{k=1}^{K} \sum_{h=1}^{H} \sum_{s=1}^{k} \mathbb{E}_{(s_h,a_h) \sim \pi^s} D_H^2 \left( \mathbb{P}_{h,f^k}(\cdot \mid s_h, a_h), \mathbb{P}_{h,f^*}(\cdot \mid s_h, a_h) \right) \right).
$$

$$
\tag{F.25}
$$

Now by the definition of $\ell^{(i),s}$ of the model-based problem in (2.5), choosing $\varepsilon = \kappa_{\text{wit}}^2 / HKB_W^2$ and $d_{\text{MADC}} = \frac{2R^2 \mathcal{D}(\varepsilon)H}{\kappa_{\text{wit}}^2}$, we can get

$$\text{Reg}(K) \leq 6 d_{\text{MADC}} H + \mu \cdot d_{\text{MADC}} + \frac{1}{\mu} \sum_{k=1}^{K} \sum_{s=1}^{k-1} \ell^{(i),s}(f^k, \pi^k)$$

complete the proof by $\mathcal{D}(\kappa_{\text{wit}}^2 / HKB_W^2) = \widetilde{\mathcal{O}}(d)$. $\qquad\square$

### F.5 PROOF OF THEOREM D.14

*Proof.* First, we fix an index $i \in [n]$. Similar to Section F.4, we can get

$$\sum_{k=1}^{K} \mathbb{E}_{s_1 \sim \rho} \left[ V_{1,f^k}^{(i),\pi^k}(s_1) - V_1^{(i),\pi^k}(s_1) \right]$$

$$= \sum_{k=1}^{K} \mathbb{E}_{\pi^k} \left[ \sum_{h=1}^{H} (Q_{h,f^k}^{(i),\pi^k}(s_h, a_h) - r_h^{(i)}(s_h, a_h) - \mathbb{E}_{s_{h+1} \sim \mathbb{P}_{h,f^*}(\cdot|s_h,a_h)} V_{h+1,f^k}^{(i),\pi^k}(s_{h+1}, a_{h+1})) \right]$$

$$= \sum_{k=1}^{K} \mathbb{E}_{\pi^k} \left[ \sum_{h=1}^{H} (\mathbb{E}_{s_{h+1} \sim \mathbb{P}_{h,f^k}(\cdot|s_h,a_h)} - \mathbb{E}_{s_{h+1} \sim \mathbb{P}_{h,f^*}(\cdot|s_h,a_h)})[V_{h+1,f^k}^{(i),\pi^k}(s_{h+1})] \right]$$

$$= \sum_{k=1}^{K} \sum_{h=1}^{H} (\theta_{h,f^k} - \theta_h^*)^T \mathbb{E}_{\pi^k} \left[ \int_{\mathcal{S}} \phi_h(s' \mid s, a) V_{h+1,f^k}^{(i),\pi^k}(s') \mathrm{d}s \right],$$

where the last equality is because of the property of the linear mixture MG.

Now we denote

$$W_h(f) = R(\theta_{h,f} - \theta_h^*) \tag{F.26}$$

$$X_h(f, \pi) = \mathbb{E}_\pi \left[ \frac{\int_{\mathcal{S}} \phi_h(s' \mid s, a) V_{h+1,f}^{(i),\pi}(s') \mathrm{d}s}{R} \right]. \tag{F.27}$$

Then, we have $\|W_h(f)\| \leq 2\sqrt{d}$, $\|X_h(f, \pi)\| \leq 1$ and

$$\sum_{k=1}^{K} \mathbb{E}_{s_1 \sim \rho} \left[ V_{1,f^k}^{(i),\pi^k}(s_1) - V_1^{(i),\pi^k}(s_1) \right] \leq \sum_{k=1}^{K} \sum_{h=1}^{H} \min\{\langle W_h(f^k), X_h(f^k, \pi^k) \rangle, R\}.$$

Now similar to Section F.4, if we replace $X_h(\pi^k)$ to $X_h(f^k, \pi^k)$, from (F.21) and (F.24) with $B_W = 2\sqrt{d}R$ we can get

$$\sum_{k=1}^{K} \mathbb{E}_{s_1 \sim \rho} \left[ V_{1,f^k}^{(i),\pi^k}(s_1) - V_1^{(i),\pi^k}(s_1) \right]$$

$$\leq HKR\varepsilon 4dR^2 + \mathcal{D}(\varepsilon)HR + \mu R^4 \cdot 2\mathcal{D}(\varepsilon)H + \frac{1}{\mu R^2} \left( \sum_{k=1}^{K} \sum_{h=1}^{H} \sum_{s=1}^{k} \langle W_h(f^k), X_h(f^s, \pi^s) \rangle^2 \right), \tag{F.28}$$

where $\mathcal{D}(\varepsilon) = 2d \log \left( \frac{\varepsilon + K}{\varepsilon} \right)$. Moreover, by (F.26) and (F.27), note that

$$\langle W_h(f^k), X_h(f^s, \pi^s) \rangle = (\theta_{h,f^k} - \theta_h^*)^T \mathbb{E}_{\pi^s} \left[ \int_{\mathcal{S}} \phi_h(s' \mid s, a) V_{h+1,f^s}^{(i),\pi^s}(s') \mathrm{d}s \right]$$

$$= \mathbb{E}_{\pi^s} \left[ (\mathbb{E}_{s_{h+1} \sim \mathbb{P}_{h,f^k}(\cdot|s_h,a_h)} - \mathbb{E}_{s_{h+1} \sim \mathbb{P}_{h,f^*}(\cdot|s_h,a_h)})[V_{h+1,f^s}^{(i),\pi^s}(s_{h+1})] \right]$$

$$\leq \mathbb{E}_{\pi^s} \left[ 2\|V_{h+1,f^s}^{(i),\pi^s}(\cdot)\|_\infty \cdot d_{\text{TV}}(\mathbb{P}_{h,f^k}(\cdot \mid s_h, a_h) \| \mathbb{P}_{h,f^*}(\cdot \mid s_h, a_h)) \right]$$

$$\leq \mathbb{E}_{\pi^s} \left[ 2\sqrt{2}RD_{\text{H}}(\mathbb{P}_{h,f^k}(\cdot \mid s_h, a_h) \| \mathbb{P}_{h,f^*}(\cdot \mid s_h, a_h)) \right].$$

Hence, from (F.28) and Jensen's inequality that $(\mathbb{E}[X])^2 \le \mathbb{E}[X^2]$, we can have

$$
\begin{aligned}
\mathrm{Reg}(K) &\le \sum_{k=1}^{K} \mathbb{E}_{s_1 \sim \rho} \left[ V_{1,f^k}^{(i),\pi^k}(s_1) - V_1^{(i),\pi^k}(s_1) \right] \\
&\le HKR\varepsilon 4dR^2 + \mathcal{D}(\varepsilon)HR + \mu R^4 \cdot 2\mathcal{D}(\varepsilon)H \\
&\quad + \frac{1}{\mu R^2} \left( \sum_{k=1}^{K} \sum_{h=1}^{H} \sum_{s=1}^{k} \mathbb{E}_{(s_h,a_h)\sim\pi^s} \left[ 8R^2 D_{\mathrm{H}}^2 \left( \mathbb{P}_{h,f^k}(\cdot \mid s_h,a_h), \mathbb{P}_{h,f^*}(\cdot \mid s_h,a_h) \right) \right] \right).
\end{aligned}
$$

By the definition of discrepancy function $\ell^{(i),s}$ in (2.5), and choosing $\varepsilon = 1/HKd$, $d_{\mathrm{MADC}} = HR^4\mathcal{D}(1/HKd) = \widetilde{\mathcal{O}}(HdR^4)$, we can derive

$$
\mathrm{Reg}(K) \le 4R^3 + \mathcal{D}(1/HKd)HR + \mu d_{\mathrm{MADC}} + \frac{1}{\mu} \sum_{k=1}^{K} \sum_{s=1}^{k-1} \ell^{(i),s}(f^k, \pi^k)
$$

$$
\le 6d_{\mathrm{MADC}}H + \mu d_{\mathrm{MADC}} + \frac{1}{\mu} \sum_{k=1}^{K} \sum_{s=1}^{k-1} \ell^{(i),s}(f^k, \pi^k).
$$

Hence, we complete the proof. □

## F.6 Proof of Lemma E.1

*Proof.* The proof is modified from Zhong et al. (2022). Define $\mathcal{W}_{j,h}$ be the filtration induced by $\{s_1^k, a_1^k, r_1^{(i),k}, \cdots, s_H^k, a_H^k, r_H^{(i),k}\}_{k=1}^{j-1}$. First, for $h \in [H], i \in [n], f \in \mathcal{F}^{(i)}$ and $\pi \in \Pi$, we define the random variable

$$
\begin{aligned}
Y_j^{(i)}(h, f, \zeta^k) &= \left( f_h(s_h^j, a_h^j) - r_h^{(i)}(s_h^j, a_h^j) - \langle f_{h+1}(s_{h+1}^j \cdot), \zeta_{h+1}^k(\cdot \mid s_{h+1}^j) \rangle \right)^2 \\
&\quad - \left( \mathcal{T}_h^{(i),\zeta^k}(f)(s_h^j, a_h^j) - r_h^{(i)}(s_h^j, a_h^j) - \langle f_{h+1}(s_{h+1}^j \cdot), \zeta_{h+1}^k(\cdot \mid s_{h+1}^j) \rangle \right)^2.
\end{aligned}
$$

By taking conditional expectation of $Y_j$ with respect to $a_h^j, s_h^j$, we can get

$$
\mathbb{E}[Y_j^{(i)}(h, f, \zeta^k) \mid \mathcal{W}_{j,h}] = \mathbb{E}_{s_h,a_h \sim \zeta^j}[(f_h - \mathcal{T}_h^{(i),\zeta^k}(f))(s_h, a_h)]^2
$$

and

$$
\mathbb{E}[(Y_j^{(i)}(h, f, \zeta^k))^2 \mid \mathcal{W}_{j,h}] \le 2R^2 \mathbb{E}[Y_j^{(i)}(h, f, \zeta^k) \mid \mathcal{W}_{j,h}],
$$

where $3R \ge |f_h(s_h^j, a_h^j) - r_h^{(i)}(s_h^j, a_h^j) - \langle f_{h+1}(s_{h+1}^j \cdot), \zeta_{h+1}^k(\cdot \mid s_{h+1}^j) \rangle)|$ is the constant upper bound. Denote $Z_j = Y_j^{(i)}(h, f, \zeta^k) - \mathbb{E}_{s_{h+1}}[Y_j^{(i)}(h, f, \zeta^k) \mid \mathcal{W}_{j,h}]$ with $|Z_j| \le 4R^2$. By the Freedman inequality, for any $0 < \eta < \frac{1}{4R}$, with probability at least $1 - \delta$,

$$
\begin{aligned}
\sum_{j=1}^{k} Z_j &= \mathcal{O}\left( \eta \sum_{j=1}^{k} \mathrm{Var}[Y_j^{(i)}(h, f, \zeta^k) \mid \mathcal{W}_{j,h}] + \frac{\log(1/\delta)}{\eta} \right) \\
&\le \mathcal{O}\left( \eta \sum_{j=1}^{k} \mathbb{E}[(Y_j^{(i)}(h, f, \zeta^k))^2 \mid \mathcal{W}_{j,h}] + \frac{\log(1/\delta)}{\eta} \right) \\
&\le \mathcal{O}\left( \eta \sum_{j=1}^{k} 2R^2 \mathbb{E}[Y_j^{(i)}(h, f, \zeta^k) \mid \mathcal{W}_{j,h}] + \frac{\log(1/\delta)}{\eta} \right).
\end{aligned}
$$

By choosing $\eta = \min\left\{ \frac{1}{4R}, \frac{\sqrt{\log(1/\delta)}}{\sqrt{2}R\sqrt{\sum_{j=1}^{k} \mathbb{E}[Y_j^{(i)}(h,f,\zeta^k)|\mathcal{W}_{j,h}]}} \right\}$, we will have

$$
\sum_{j=1}^{k} Z_j = \mathcal{O}\left( R\sqrt{\sum_{j=1}^{k} \mathbb{E}[Y_j^{(i)}(h, f, \zeta^k) \mid \mathcal{W}_{j,h}] \log(1/\delta)} + R^2 \log(1/\delta) \right).
$$

Similarly, if we apply the Freedman's inequality with $-\sum_{j=1}^{k} Z_j$, with probability at least $1 - 2\delta$,

$$\left| \sum_{j=1}^{k} Z_j \right| = \mathcal{O}\left( R\sqrt{\sum_{j=1}^{k} \mathbb{E}[Y_j^{(i)}(h, f, \zeta^k) \mid \mathcal{W}_{j,h}] \log(1/\delta) + R^2 \log(1/\delta)} \right).$$

Denote the $\rho$-covering set of $\mathcal{F}^{(i)}$ as $\mathcal{C}_{\mathcal{F}^{(i)}}(\rho)$, then for any $f \in \mathcal{F}^{(i)}, \zeta \in \Pi_i^{\text{pur}}$, there exists a pair $\widetilde{f} \in \mathcal{C}_{\mathcal{F}^{(i)}}(\rho)$ such that

$$\left| \left( f_h(s_h, a_h) - r_h^{(i)}(s_h, a_h) - \langle f_{h+1}(s_{h+1}, \cdot), \zeta_{h+1}(\cdot \mid s_{h+1}) \rangle \right) \right.$$
$$\left. - \left( \widetilde{f}_h(s_h, a_h) - r_h^{(i)}(s_h, a_h) - \langle \widetilde{f}_{h+1}(s_{h+1}, \cdot), \zeta_{h+1}(\cdot \mid s_{h+1}) \rangle \right) \right| \leq 3\rho$$

for all $(s_h, a_h, s_{h+1}) \in \mathcal{S} \times \mathcal{A} \times \mathcal{S}$. Now by taking a union bound over $\mathcal{C}_{\mathcal{F}^{(i)}}(\rho)$, we have that with probability at least $1 - \delta$, for all $\widetilde{f} \in \mathcal{C}_{\mathcal{F}^{(i)}}(\rho)$,

$$\left| \sum_{j=1}^{k} \widetilde{Y}_j^{(i)}(h, \widetilde{f}, \zeta) - \sum_{j=1}^{k} \mathbb{E}[\widetilde{Y}_j^{(i)}(h, \widetilde{f}, \zeta) \mid \mathcal{W}_{j,h}] \right|$$

$$= \mathcal{O}\left( R\sqrt{\sum_{j=1}^{k} \mathbb{E}[\widetilde{Y}_j^{(i)}(h, \widetilde{f}, \zeta) \mid \mathcal{W}_{j,h}] \iota + R^2 \iota} \right), \tag{F.29}$$

where $\iota = 2\log(HK|\mathcal{C}_{\mathcal{F}^{(i)}}(\rho)|/\delta) \leq 2\log(HK\mathcal{N}_{\mathcal{F}^{(i)}}(\rho))$.

Now note that for all $f \in \mathcal{F}^{(i)}, \zeta \in \Pi_i^{\text{pur}}$, we have

$$\sum_{h=1}^{H} \sum_{j=0}^{k-1} Y_j^{(i)}(h, f, \zeta)$$

$$= \sum_{h=1}^{H} \sum_{j=0}^{k-1} (f_h(s_h^j, a_h^j) - r_h^{(i)}(s_h^j, a_h^j) - \langle f_{h+1}(s_{h+1}^j, \cdot), \zeta_{h+1}(\cdot \mid s_{h+1}^j) \rangle)^2$$

$$- (\mathcal{T}_h^{(i),\zeta}(f)(s_h^j, a_h^j) - r_h^{(i)}(s_h^j, a_h^j) - \langle f_{h+1}(s_{h+1}^j, \cdot), \zeta_{h+1}(\cdot \mid s_{h+1}^j) \rangle)^2$$

$$\leq \sum_{h=1}^{H} \sum_{j=0}^{k-1} (f_h(s_h^j, a_h^j) - r_h^{(i)}(s_h^j, a_h^j) - \langle f_{h+1}(s_{h+1}^j, \cdot), \zeta_{h+1}(\cdot \mid s_{h+1}^j) \rangle)^2$$

$$- \inf_{f_h' \in \mathcal{F}_h^{(i)}} (\mathcal{T}_h^{(i),\zeta^k}(f')(s_h^j, a_h^j) - r_h^{(i)}(s_h^j, a_h^j) - \langle f_{h+1}(s_{h+1}^j, \cdot), \zeta_{h+1}(\cdot \mid s_{h+1}^j) \rangle)^2$$

$$= L^{(i),k-1}(f, \zeta, \tau^{1:k-1}).$$

Then, by (F.29) we can get

$$\sum_{h=1}^{H} \sum_{j=0}^{k-1} \mathbb{E}[\widetilde{Y}_j^{(i)}(h, \widetilde{f}, \zeta) \mid \mathcal{W}_{j,h}] \leq 4L^{(i),k-1}(\widetilde{f}, \zeta, \tau^{1:k-1}) + \mathcal{O}(HR^2\iota).$$

Now similar to (Jin et al., 2021a), by the definition of $\rho$-covering number, for any $k \in [K], f \in \mathcal{F}^{(i)}$ and $\zeta \in \Pi_i^{\text{pur}}$,

$$\sum_{h=1}^{H} \sum_{j=0}^{k-1} \mathbb{E}[Y_j^{(i)}(h, f, \zeta) \mid \mathcal{W}_{j,h}] \leq 4L^{(i),k-1}(f, \zeta, \tau^{1:k-1}) + \mathcal{O}(HR^2\iota + HRk\rho).$$

Now since $s_h^j, a_h^j \sim \zeta^j$, we can have

$$\sum_{j=0}^{k-1} \ell^{j,(i)}(f, \zeta^k) = \sum_{j=0}^{k-1} \mathbb{E}[Y_j^{(i)}(h, f, \zeta^k) \mid \mathcal{W}_{j,h}]$$

$$\leq 4L^{(i),k-1}(f, \zeta^k, \tau^{1:k-1}) + \mathcal{O}(HR^2\iota + HRk\rho).$$

We complete the proof by choosing $\rho = 1/K$ and choose $\varepsilon_{\text{conc}} = \mathcal{O}(HR^2\iota + HRk\rho) = \mathcal{O}(HR^2\iota)$.
$\square$

### F.7 PROOF OF LEMMA E.2

*Proof.* First, for any $f \in \mathcal{F}^{(i)}$ and $\pi \in \Pi^{\mathrm{pur}}$ we define the random variable

$$Q_j^{(i)}(h, f, \pi) = (f_h(s_h^j, a_h^j) - r_h^{(i)}(s_h^j, a_h^j) - \langle f_{h+1}^*(s_{h+1}^j, \cdot), \pi_{h+1}(\cdot \mid s_{h+1}^j) \rangle)^2$$
$$- (f_h^*(s_h^j, a_h^j) - r_h^{(i)}(s_h^j, a_h^j) - \langle f_{h+1}^*(s_{h+1}^j, \cdot), \pi_{h+1}(\cdot \mid s_{h+1}^j) \rangle)^2.$$

Then, by similar derivations in Lemma E.1, we can get

$$\mathbb{E}[Q_j^{(i)}(h, f, \pi) \mid \mathcal{W}_{j,h}] = \mathbb{E}_{s_h, a_h \sim \zeta^j}[(f_h - \mathcal{T}^{(i),\pi}(f^*))(s_h, a_h)]^2 \geq 0,$$
$$\mathbb{E}[(Q_j^{(i)}(h, f, \pi))^2 \mid \mathcal{W}_{j,h}] \leq 2R^2 \mathbb{E}[Q_j^{(i)}(h, f, \pi) \mid \mathcal{W}_{j,h}].$$

Then, by Freedman's inequality, with probability at least $1 - \delta$, for all elements in $\widetilde{f} \in \mathcal{C}_{\mathcal{F}^{(i)}}(\rho)$, we have

$$\left| \sum_{j=0}^{k-1} \widetilde{Q}_j^{(i)}(h, \widetilde{f}, \pi) - \sum_{j=0}^{k-1} \mathbb{E}[\widetilde{Q}_j^{(i)}(h, \widetilde{f}, \pi) \mid \mathcal{W}_{j,h}] \right|$$
$$= \mathcal{O}\left( R\sqrt{\sum_{j=0}^{k-1} \mathbb{E}_{s_{h+1}}[\widetilde{Q}_j^{(i)}(h, \widetilde{f}, \pi) \mid \mathcal{W}_{j,h}]\iota + R^2\iota} \right),$$

then we can have

$$\sum_{j=0}^{k-1} \widetilde{Q}_j^{(i)}(h, \widetilde{f}, \pi) \geq -\mathcal{O}(R^2\iota).$$

Thus, by the definition of $\mathcal{C}_{\mathcal{F}^{(i)}}(\rho)$, for all $f \in \mathcal{F}^{(i)}$ and $\pi \in \Pi_i^{\mathrm{pur}}$, we have

$$-\sum_{j=0}^{k-1} Q_j^{(i)}(h, f, \pi) \leq \mathcal{O}(R^2\iota + Rk\rho).$$

Thus,

$$L^{(i),k}(f, \pi) = \sum_{h=1}^{H} \left( -\inf_{f \in \mathcal{F}^{(i)}} \sum_{j=0}^{k-1} Q_j^{(i)}(h, f, \pi) \right) \leq \mathcal{O}(HR^2\iota + HRk\rho) = \mathcal{O}(HR^2\iota).$$

Thus, we complete the proof. $\qquad \square$

### F.8 PROOF OF LEMMA E.3

*Proof.* For simplicity, we first assume $\mathcal{F}$ is a finite class. Given a model $f \in \mathcal{F}$ and $h \in [H]$, we define $X_{h,f}^j = \log \frac{\mathbb{P}_{h,f^*}(s_{h+1}^j | s_h^j, a_h^j)}{\mathbb{P}_{h,f}(s_{h+1}^j | s_h^j, a_h^j)}$. Thus,

$$L^{(i),k}(f^*, \tau^{1:k}) - L^{(i),k}(f, \tau^{1:k}) = -\sum_{h=1}^{H} \sum_{j=1}^{k} X_{h,f}^j. \tag{F.30}$$

Now we define the filtration $\mathcal{G}_j$ as

$$\mathcal{G}_j = \sigma(\{s_h^1, a_h^1, \cdots, s_h^j, a_h^j\}).$$

Then, by Lemma G.1 for all $f \in \mathcal{F}$, with probability at least $1 - \delta$, we have

$$-\sum_{j=1}^{k} X_{h,\bar{f}}^j \leq \sum_{j=1}^{k} \log \mathbb{E}\left[ \exp\left\{ -\frac{1}{2} X_{h,\bar{f}}^j \right\} \,\middle|\, \mathcal{G}_{j-1} \right] + \log(H|\mathcal{F}|/\delta).$$

Now we decompose the first term at the right side as

$$
\mathbb{E}\left[\exp\left\{-\frac{1}{2}X_{h,\bar{f}}^j\right\}\,\middle|\,\mathcal{G}_{j-1}\right]
$$

$$
=\mathbb{E}\left[\sqrt{\left|\frac{\log\mathbb{P}_{h,f}(s_{h+1}^j\mid s_h^j,a_h^j)}{\mathbb{P}_{h,f^*}(s_{h+1}^j\mid s_h^j,a_h^j)}\right|}\,\middle|\,\mathcal{G}_{j-1}\right]
$$

$$
=\mathbb{E}_{(s_h^j,a_h^j)\sim\pi^j}\mathbb{E}_{s_{h+1}\sim\mathbb{P}_{h,f^*}(\cdot\mid s_h^j,a_h^j)}\left[\sqrt{\left|\frac{\mathbb{P}_{h,f}(s_{h+1}^j\mid s_h^j,a_h^j)}{\mathbb{P}_{h,f^*}(s_{h+1}^j\mid s_h^j,a_h^j)}\right|}\,\middle|\,\mathcal{G}_{j-1}\right]
$$

$$
=\mathbb{E}_{(s_h^j,a_h^j)\sim\pi^j}\left[\int\sqrt{\mathbb{P}_{h,f}(s_{h+1}^j\mid s_h^j,a_h^j)\mathbb{P}_{h,f^*}(s_{h+1}^j\mid s_h^j,a_h^j)}d_{s_{h+1}^j}\right]
$$

$$
=1-\frac{1}{2}\mathbb{E}_{(s_h^j,a_h^j)\sim\pi^j}[D_{\mathrm{H}}^2(\mathbb{P}_{h,f}(s_{h+1}^j\mid s_h^j,a_h^j)\|\mathbb{P}_{h,f^*}(s_{h+1}^j\mid s_h^j,a_h^j))].
$$

Now by the inequality $\log x \le x-1$, we have

$$
-\sum_{j=1}^k X_{h,f}^j \le \sum_{j=1}^k\left(1-\frac{1}{2}\mathbb{E}_{(s_h^j,a_h^j)\sim\pi^j}[D_{\mathrm{H}}^2(\mathbb{P}_{h,f}(s_{h+1}^j\mid s_h^j,a_h^j)\|\mathbb{P}_{h,f^*}(s_{h+1}^j\mid s_h^j,a_h^j))]\right)
$$

$$
-1+\log(H|\mathcal{F}|/\delta)
$$

$$
\le-\sum_{j=1}^k\frac{1}{2}\mathbb{E}_{(s_h^j,a_h^j)\sim\pi^j}\left[D_{\mathrm{H}}^2(\mathbb{P}_{h,f}(s_{h+1}^j\mid s_h^j,a_h^j)\|\mathbb{P}_{h,f^*}(s_{h+1}^j\mid s_h^j,a_h^j))\right]
$$

$$
+\log(H|\mathcal{F}|/\delta).
$$

Sum over $h\in[H]$ with (F.30), we can complete the proof by

$$
-\sum_{h=1}^H\sum_{j=1}^k X_{h,f}^j \le -\sum_{j=1}^k\ell^{(i),j}(f)+\kappa_{\mathrm{conc}},
$$

where $\kappa_{\mathrm{conc}}=H\log(H|\mathcal{F}|/\delta)$. For infinite model classes $\mathcal{F}$, we can use $1/K$-bracketing number $\mathcal{B}_{\mathcal{F}}(1/K)$ to replace the cardinality $|\mathcal{F}|$ (Liu et al., 2022a; Zhong et al., 2022; Zhan et al., 2022b). □

## F.9 PROOF OF COROLLARY E.4

*Proof.* We provide the proof for NE. The proof for CCE/CE are the same by replacing the NE-regret to the CCE/CE-regret. By taking the minimum of the index of agents rather than adding them together, we can modify the proof of Theorem 3.1 and derive, with probability at least $1-\delta$,

$$
\frac{1}{K}\left(\sum_{k=1}^K\max_{i\in[n]}\bigl(V^{(i),\mu^{(i),\pi^k}}(\rho)-V^{(i),\pi^k}(\rho)\bigr)\right)\le\widetilde{\mathcal{O}}\left(\frac{H\Upsilon_{\mathcal{F},\delta}}{\sqrt{K}}+\frac{d_{\mathrm{MADC}}}{\sqrt{K}}+\frac{d_{\mathrm{MADC}}H}{K}\right).
$$

Hence, by choosing $K=\widetilde{\mathcal{O}}\left((H^2\Upsilon_{\mathcal{F},\delta}^2+d_{\mathrm{MADC}}^2)\cdot\varepsilon^{-2}+d_{\mathrm{MADC}}H\cdot\varepsilon^{-1}\right)$ with $\varepsilon<1$, we have

$$
\max_{i\in[n]}\bigl(V^{(i),\mu^{(i),\pi_{\mathrm{out}}}}(\rho)-V^{(i),\pi_{\mathrm{out}}}(\rho)\bigr)
$$

$$
=\frac{1}{K}\left(\sum_{k=1}^K\max_{i\in[n]}\bigl(V^{(i),\mu^{(i),\pi^k}}(\rho)-V^{(i),\pi^k}(\rho)\bigr)\right)
$$

$$
\le\varepsilon,
$$

where the second inequality holds from $\pi_{\mathrm{out}}=\mathrm{Unif}(\{\pi^k\}_{k\in[K]})$. Hence, $\pi_{\mathrm{out}}$ is a $\varepsilon$-NE. □

## G  TECHNICAL TOOLS

We provide the following lemma to complete the proof of model-based RL problems. The detailed proof can be found in (Foster et al., 2021).

**Lemma G.1.** For any real-valued random variable sequence $\{X_k\}_{k \in [K]}$ adapted to a filtration $\{\mathcal{G}_k\}_{k \in [K]}$, with probability at least $1 - \delta$, for any $k \in [K]$, we can have

$$-\sum_{s=1}^{k} X_k \le \sum_{s=1}^{k} \log \mathbb{E}[\exp(-X_s) \mid \mathcal{F}_{s-1}] + \log(1/\delta).$$

In the next lemma, we introduce the Freedman's inequality, which has been commonly used in previous RL algorithms. (Jin et al., 2021b; Chen et al., 2022c; Zhong et al., 2022)

**Lemma G.2** (Freedman's Inequality (Agarwal et al., 2014))**.** Let $\{Z_k\}_{k \in [K]}$ be a martingale difference sequence that adapted to filtration $\{\mathcal{F}_k\}_{k \in [K]}$. If $|Z_k| \le R$ for all $k \in [K]$, then for $\eta \in (0, \frac{1}{R})$, with probability at least $1 - \delta$, we can have

$$\sum_{k=1}^{K} X_k = \mathcal{O}\left( \eta \sum_{k=1}^{K} \mathbb{E}[X_k^2 \mid \mathcal{F}_{k-1}] + \frac{\log(1/\delta)}{\eta} \right).$$

The next elliptical potential lemma is first introduced in the linear bandit literature (Dani et al., 2008; Abbasi-Yadkori et al., 2011) and then applied to the RL problems with Bilinear Classes (Du et al., 2021) and the general function approximation (Chen et al., 2022a; Zhong et al., 2022).

**Lemma G.3** (Elliptical Potential Lemma)**.** Let $\{x_k\}_{k=1}^{K}$ be a sequence of real-valued vector, i.e. $x_k \in \mathbb{R}^d$ for any $k \in [K]$. Then, if we define $\Lambda_i = \varepsilon I + \sum_{k=1}^{K} x_k x_k^T$, we can get that

$$\sum_{k=1}^{K} \min\left\{ 1, \|x_i\|_{\Lambda_i^{-1}}^2 \right\} \le 2 \log\left( \frac{\det(\Lambda_{K+1})}{\det(\Lambda_1)} \right) \le 2 \log \det\left( I + \frac{1}{\varepsilon} \sum_{k=1}^{K} x_k x_k^T \right).$$

*Proof.* The proof is provided in Lemma 11 of (Abbasi-Yadkori et al., 2011). $\square$

## H  COMPUTATIONAL EFFICIENCY

Since the normal-form game is defined over the pure policy space, the size of this game can be exponentially large, which makes the algorithm not computationally efficient. Actually, the computational complexity of this oracle depends on the type of equilibrium.

For CCE, the oracle can be approximately implemented by mirror descent (Wang et al., 2023), where the time of calculating $V(\pi)$ depends on the number of iterations. Hence, the computational complexity of this oracle depends only on the number of iterations rather than the pure policy space $|\Pi|^{\mathrm{pur}}$. To be more specific, by replacing the optimistic value function $\overline{V}_i^{(t),\pi}$ in Algorithm 4 (DOPMD) of [1] by the regularized payoff function in 3.1, one can get an approximated CCE by mirror descent using FTRL analysis. The sampling procedure in Line 3 of DOPMD can be implemented by Langevin dynamics (Karagulyan, 2021) as long as we can obtain the gradient of the regularized payoff function.

However, for Nash Equilibrium, calculating the equilibrium is PPAD-complete. So MAMEX is computationally inefficient for learning NE.

