# OpenReview forum: "Sample-Efficient Multi-Agent RL: An Optimization Perspective"
_ICLR.cc/2024/Conference — ICLR 2024 poster_

### Official Review · Reviewer_FdZN · 2023-10-24

**Soundness:** 3 good
**Presentation:** 2 fair
**Contribution:** 3 good
**Rating:** 6
**Confidence:** 2

**Summary:**

This submission introduces a new algorithm dubbed MAMEX which enjoys regret bounds for the Nash Regret in General Sum Markov Games with low MADC. A metric introduced in this work that measures the complexity of a certain Markov Games.

Via an online to offline conversion the regret bounds can be converted into sample complexity guarantees for learning an $\epsilon$-approximate Nash Equilibrium. These guarantees hold in expectation.

**Strengths:**

I think that the paper's idea is well explained in the main text.
Understanding which are the lighter assumptions to enable sample efficient learning in Markov Games is an important theoretical problem which matches the interests of the ICLR community.

**Weaknesses:**

I think that the paper should make it clearer that MAMEX is not computationally efficient.
This is because of (i) the equilibrium computation over the set of pure policies in step 4 of Algorithm 1 is often over combinatorial sets like for example the set of deterministic policies in the tabular setting and (ii) the sup over the function class in the policy evaluation routine.

**Questions:**

Q1) In the tabular case or in the Linear MDP case would there exist an efficient way to implement MAMEX ?

Q2) Do you think that the MADC are necessary quantities that should appear in lower bounds too ?

---

> ### Author Response · Authors · 2023-11-18
> **Response to the Reviewer FdZN**
>
> Thanks for your response. We will address your questions below.
>
> > 1. I think that the paper should make it clearer that MAMEX is not computationally efficient. This is because of (i) the equilibrium computation over the set of pure policies in step 4 of Algorithm 1 is often over combinatorial sets like for example the set of deterministic policies in the tabular setting and (ii) the sup over the function class in the policy evaluation routine. In the tabular case or in the Linear MDP case would there exist an efficient way to implement MAMEX?
>
>
>
> (i). The equilibrium computation over the set of pure policies.
>
> The policy evaluation step refers to Lines 3 and 4 in MAMEX (Algorithm 1). If we regard these two lines as an oracle, MAMEX is oracle-efficient. The computational complexity of this oracle depends on the type of equilibrium. For CCE, the oracle can be approximately implemented by mirror descent [1], where the time of calculating $V(\pi)$ depends on $L$ and $L$ is the number of iterations. Hence, the computational complexity of this oracle depends only on the number of iterations rather than the joint pure policy space $|\Pi|$, and MAMEX can be calculated efficiently. However, for Nash Equilibrium, calculating the equilibrium is PPAD-complete. So MAMEX is computationally inefficient for learning NE.
>
> (ii). The sup over the function class in the policy evaluation routine
>
> We believe that it can be solved by using similar techniques in [2], where they alternately update the policy and the hypothesis function (the model parameter in the model-based version or the Q-function in the model-free setting). Particularly, we apply gradient ascent when updating the hypothesis function. In MAMEX, if we use policy iteration or soft policy iteration to solve a CCE, their alternate updates are also available here.
>
>
>
> > 2. Do you think that the MADC are necessary quantity that should appear in lower bounds too?
>
> We think the MADC is a necessary quantity that should appear in lower bounds. However, even for the single-agent DC ([3]), the lower bound is still lacking. We leave it as a future work.
>
>
> *references:*
>
> [1]. Wang et al.  Breaking the curse of multiagency: Provably efficient decentralized multi-agent rl with function approximation. preprint 2023.
>
> [2]. Liu et al. Maximize to Explore: One Objective Function Fusing Estimation, Planning, and Exploration, NeurIPS 2023.
>
> [3]. Dann et al. A provably efficient model-free posterior sampling method for episodic reinforcement learning. NeurIPS 2021.

---

> > ### Comment · Reviewer_FdZN · 2023-11-20
> >
> > Dear Authors,
> >
> > Thanks for your response. Could you elaborate more on why finding a CCE in line 4 can be implemented with computational complexity independent of $\Pi^{pur}$ ? Unfortunately, I still do not see how this is possible using the mirror descent algorithm you pointed me to.
> >
> > Also I don't understand how the policy evaluation routine could be solved efficiently for potentially con convex function classes. Wouldn't it be better to simply state that in general the problem is not solvable ?
> > Then, it is good to mention that in simpler cases finding a solution is possible as you explained in the answer to my review.
> >
> > This would really help to understand in which settings MAMEX is also computationally efficient.
> >
> > Best,
> > Reviewer FdZN

---

> > > ### Author Response · Authors · 2023-11-21
> > > **Response to the Reviewer FdZN**
> > >
> > > By replacing the optimistic value function $\overline{V}_i^{(t),\pi}$ in Algorithm 4 (DOPMD) of [1] by the regularized payoff function in (3.1), one can get an approximated CCE by mirror descent using FTRL analysis in Section 6.6 in [4]. To be more specific, using the FTRL analysis, the bound of (a) in Section I.3 of [1] can be transformed to the following inequality
> > >
> > > $$\max_{\pi_i \in \Pi_i}\sum_{t=1}^T [\overline{V}\_i(\pi_i \times \pi_{-i}^t) - \overline{V}\_i(\Lambda_i^t \times \pi_{-i}^t)] \le O(H\sqrt{\log |\Pi_i|T} ).$$
> > > Then by Azuma-Hoeffding's inequality, we can have
> > > $$ \begin{aligned}\max_{\pi_i \in \Pi_i}\sum_{t=1}^T [\overline{V}\_i(\pi_i \times \Lambda_{-i}^t) - \overline{V}\_i(\Lambda_i^t \times \Lambda_{-i}^t)] &\le \max_{\pi_i \in \Pi_i}\sum_{t=1}^T [\overline{V}\_i(\pi_i \times \Lambda_{-i}^t) - \overline{V}\_i(\Lambda_i^t \times \Lambda_{-i}^t)] + O\left(H\sqrt{T\log (\sum_{i \in [n]}|\Pi_i|/\delta)}\right)\le O\left(H\sqrt{T\log (\sum_{i \in [n]}|\Pi_i|/\delta)}\right).\end{aligned}$$
> > >
> > > From the inequality above, the mixture policy $\Lambda^{out}=\frac{1}{T}\sum_{t=1}^T \Lambda^t_1\times \cdots \times \Lambda_n^t$ is the approximated CCE by
> > > $$\max_{\pi_i \in \Pi_i}\overline{V}\_i(\pi_i \times \Lambda_{-i}^{out})-\overline{V}\_i(\Lambda_i^{out} \times \Lambda_{-i}^{out}) \le \max_{\pi_i \in \Pi_i}\sum_{t=1}^T [\overline{V}\_i(\pi_i \times \Lambda_{-i}^t) - \overline{V}\_i(\Lambda_i^t \times \Lambda_{-i}^t)]\le O\left(H\sqrt{\log (\sum_{i \in [n]}|\Pi_i|/\delta)/T}\right)$$
> > > if $T$ is large.
> > >
> > > In general, the DOPMD is not computationally efficient since the calculation of the payoff function $\overline{V}$ is as hard as solving nonconvex optimization. However, there are some simple cases in which the algorithm can be computationally efficient.
> > >
> > > For example, if we assume the class of value function as a linear function class $V_{f}^{(i),\pi}(s) = \langle Q_f^{(i)}(s,a),\pi\rangle$ and $Q_f^{(i)}(s,a) = \langle \phi^{(i)}(s,a), \theta_f\rangle$, and the empirical loss function $L^{(i),k}(f,\pi,\tau)$ is a quadratic function of $\theta_f$. Then, the calculation of the payoff function in (3.1) can be implemented efficiently because it is given by maximizing a concave quadratic function of $\theta_f$. Moreover, the sampling procedure in Line 3 of DOPMD can be implemented efficiently by Langevin dynamics ([2]) as long as we can obtain the gradient of $\overline{V}\_i(\pi)$ in (3.1) with respect to $\pi$, i.e., the policy gradient. By using Langevine dynamics, we can sample $\Lambda_i^{t+1} \propto_{\pi_i} \Lambda_i^t \cdot \exp(\eta_i \cdot \overline{V}\_i(\pi_i\times \pi_{-i}^t))$ in Line 6 of DOPMD by the following iteration:
> > >
> > > $$\pi^{(k+1)} = \pi^{(k)} - \nabla_{\pi^{(k)}} \log (\Lambda_i^0)-\alpha \sum_{t'=1}^t\nabla_{\pi^{(k)}} (\eta_i \cdot \overline{V}\_i(\pi^{(k)}\times \pi_{-i}^t)) + \sqrt{2\alpha}\epsilon^{(k)},$$
> > >
> > > where $\alpha$ is the step-size and $\varepsilon^{(k)}$ is a  sequence of standard normal vectors.
> > >
> > > Hence, it avoids calculating  $\overline{V}\_i(\pi)$ for all policies $\pi \in \Pi^{\text{pur}}$. In fact, we can parameterize the policy $\pi_i$ as $\pi_\theta$, and thus the loss function $L^{(i),k}(f,\pi,\tau)$ is quadratic in $\theta$ and $V_{f}^{(i),k,\pi}(\rho)$ is linear in $\theta$. Then, the regularized payoff function $\overline{V}\_i(\pi_i \times \pi_{-i}^t)$ is always a differentiable function with respect to $\pi_i$, and the policy distribution $\Lambda_i^{t}$ is always log-concave, which guarantees that the Langevin dynamics converges to the true distribution [3].
> > >
> > >
> > > *References:*
> > >
> > > [1]. Wang et al.  Breaking the curse of multiagency: Provably efficient decentralized multi-agent rl with function approximation. preprint 2023.
> > >
> > > [2]. Avetik Karagulyan. Sampling with the Langevin Monte-Carlo. Probability [math.PR]. Institut Polytechnique de Paris, 2021. English.
> > >
> > > [3]. Dalalyan A. Further and stronger analogy between sampling and optimization: Langevin Monte Carlo and gradient descent. COLT 2017.
> > >
> > > [4]. F Orabona. A modern introduction to online learning. preprint 2019.

---

> > > > ### Comment · Reviewer_FdZN · 2023-11-22
> > > >
> > > > Dear Authors,
> > > >
> > > > Thanks a lot for your additional explanations.
> > > >
> > > > I think that it would be nice to add in the paper the remark that the algorithm can be computationally efficient in the case $Q_f^{(i)}$ is a linear function.
> > > >
> > > > I keep my positive score.
> > > >
> > > > Best,
> > > >
> > > > Reviewer

---

### Official Review · Reviewer_hiyJ · 2023-10-28

**Soundness:** 4 excellent
**Presentation:** 4 excellent
**Contribution:** 3 good
**Rating:** 6
**Confidence:** 3

**Summary:**

The paper provides a general recipe for solving multi-agent reinforcement learning through the notion of decoupling coefficient. The framework allows for various function approximations and encompasses both model-based and model-free approaches. Theoretical results show that if a game enjoys finite MADC then there exists a centralized algorithm for $\sqrt{K}$ type regret.

**Strengths:**

1. The framework proposed provides a general recipe for solving multi-agent reinforcement learning under various function approximations, and the bounds obtained under this framework are sample efficient. I believe this is the first framework that includes both model-based and model-free methods.
2. Compared to the previous works (MA - DMSO), the framework only requires solving a single objective optimization, which makes it much more feasible.

**Weaknesses:**

While I like the results presented in this paper, it is not very clear to me what the relationship between MADC and the existing measures (such as MADMSO, the single agent decoupling coefficient). I believe the paper would benefit from more discussion and comparison on that. Please see the question raised below for more details on this point.

**Questions:**

1. When $n = 1$, the problems are reduced to a single agent RL. In this case, is MADC reduced to DC, or is it more general than DC? (since it is not restricted to greedy policy)
2. If MADC reduces to DC when $n=1$, does an RL instance with finite DC imply the MG version of the RL instance also has finite MADC (and thus can be solved efficiently?)
3. How should one compare MADC and MADMSO? We know that tabular MG and linear mixture MG have finite MADMSO and MADC. Are there any classes of games that have finite MADC but not finite MADMSO or vice versa?
4. It seems that the current framework does not distinguish NE/CCE/CE. It seems that as long as the NE/CCE/CE equilibrium oracle is efficient, it takes the same amount of samples to find NE/CCE/CE?  As NE is a much stricter notion of equilibrium than CCE, for example, on might expect more samples to be needed (or in other words, harder) to learn NE. So are the results tight for each of NE/CCE/CE or is it an upper bound for the worst-case regret needed for finding any equilibrium?

---

> ### Author Response · Authors · 2023-11-18
> **Response to the Reviewer hiyJ**
>
> Thanks for your response. We will address your questions below.
>
> > 1. When $n=1$, the problems are reduced to a single agent RL. In this case, is MADC reduced to DC, or is it more general than DC? (since it is not restricted to greedy policy)
>
> It is more general than DC since the policy $\pi^k$ in (2.3) is not restricted to the greedy policy. In other words, the MADC in $n=1$ is not less than DC. A function class of MDP with small DC can also have large MADC for $n=1$.
>
> > 2. If MADC reduces to DC when $n=1$, does an RL instance with finite DC imply the MG version of the RL instance also has finite MADC (and thus can be solved efficiently?)
>
>
> There is no nontrivial example in which MADC is infinite since MADC in tabular MG is also finite ($O(|S||A|)$) when states and actions are finite.  However, an RL instance with a small DC cannot imply the MG version of the RL instance also has a small MADC, since the policy $\pi^k$ is not restricted to the greedy policy. Hence, it needs to further check whether the multi-agent version of the RL instance satisfies the Equation (2.3) for every policy sequences $\{\pi^k\}_{k \in [K]}$.
>
> > 3. How should one compare MADC and MADMSO?
>
> In general, we do not see a clear relation between MADC and MADMSO, in the sense that our complexity measures cannot be reduced to theirs and vice versa. However, when applying to some specific examples, the inducing results are comparable. For example, when applying to the linear MG with dimension $d$,  the paper [3] achieves a $\widetilde{O}(n^2d^2H^4\max_{i=1}^n \log |\Pi_i|/\varepsilon^2)$ sample complexity, while our paper can achieve a $\widetilde{O}(d^2H^2 + n^2H^2(\max_{i=1}^n\log |\Pi_i|)^2/\varepsilon^2)$ sample complexity.
>  Compared to [3],  our result has a higher degree in term $\max_{i=1}^n \log|\Pi_i|$. When $\max_{i=1}^n \log |\Pi_i|$ is relatively small and can be ignored, our $\widetilde{O}((n^2+d^2)H^2/\varepsilon^2)$ sample complexity is smaller than the sample complexity result $\widetilde{O}(n^2d^2H^2/\varepsilon^2)$ in [3]. In addition,  our complexity measure also captures model-free settings, while the algorithm and the complexity measure MA-DEC in [3] heavily depend on the model and cannot be directly applied to the model-free setting.
>
> > 4. It seems that the current framework does not distinguish NE/CCE/CE. It seems that as long as the NE/CCE/CE equilibrium oracle is efficient, it takes the same amount of samples to find NE/CCE/CE? As NE is a much stricter notion of equilibrium than CCE, for example, on might expect more samples to be needed (or in other words, harder) to learn NE. So are the results tight for each of NE/CCE/CE or is it an upper bound for the worst-case regret needed for finding any equilibrium?
>
>
>
> The results is not tight for each of NE/CCE/CE. For example, one can use the decentralized function approximation to learn CCE with polynomial sample complexity for tabular general-sum MG ([1]), while any general framework that learns NE/CCE/CE ([2,3,4,5] and our paper) suffers an exponential term $\prod_{i=1}^n |A_i|$ for the tabular general-sum MG due to the intrinsic difficulty of learning NE.
>
>
> Instead of getting the optimal result in one particular equilibrium, our goal is to provide an unified algorithmic framework that learns NE/CCE/CE for both model-based and model-free RL under the general function approximation.
>
>
> *References:*
>
> [1]. Wang et al.  Breaking the curse of multiagency: Provably efficient decentralized multi-agent rl with function approximation. preprint 2023.
>
> [2]. Chen et al. Unified algorithms for rl with decision-estimation coefficients: No-regret, pac, and reward-free learning. preprint 2022.
>
> [3]. Foster et al. On the complexity of multi-agent decision making: From learning in games to partial monitoring. COLT 2023.
>
> [4]. Liu et al. Sample-efficient reinforcement learning of partially observable markov games. NeurIPS 2022.
>
> [5]. Jin et al. V-Learning--A Simple, Efficient, Decentralized Algorithm for Multiagent RL. ICLR Workshop 2022.
>
> [6]. Liu et al. Maximize to Explore: One Objective Function Fusing Estimation, Planning, and Exploration, NeurIPS 2023.

---

> > ### Comment · Reviewer_hiyJ · 2023-11-21
> > **Follow up questions**
> >
> > Thank you for your response.
> >
> > I wonder if MADC offers advantages (better sample complexities) with general function approximations when compared to the other methods? From just the linear function approximation example, it is still unclear whether MADC is advantageous (in terms of sample complexity) under general function approximations.

---

> > > ### Author Response · Authors · 2023-11-22
> > > **Response to the Follow up**
> > >
> > > Thanks for your follow-up. For the model-free RL problem, the main advantage is that we provide the first sample complexity results for NE/CCE/CE in the general-sum MG under the general function approximation.
> > >
> > > To be more specific, the previous works that study all three equilibria under general function approximation can apply **only on the two-player zero-sum MG**. This is because their complexity measures are based on the Bellman optimality operator ([4,5]), i.e.,
> > > $$\mathcal{T}\_h Q_{h+1}(s,a,b) = r(s,a,b)+\mathbb{E}\_{s'\sim \mathbb{P}\_h(s'\mid s,a,b)}[\max_{\pi'}\min_{\upsilon'}V_{h+1}^{\pi'\times \upsilon'}(s')].$$
> > >
> > > However, in general-sum MGs, this concept is not well-defined in Markov Games. **Unlike the zero-sum MG, there is no unique optimal equilibria in the general-sum MG, and thus the Bellman optimality operator cannot be defined in a similar way.** Compared to them, our advantage is to use arbitrary policy sequence in the MADC, to handle the general-sum MG under the general function approximation using the Bellman operator defined on any $\pi$ and agent $i$, i.e.
> > >
> > > $$\mathcal{T}\_h^{(i),\pi} Q_{h+1}(s,a) = r_h^{(i)}(s,a) + \mathbb{E}\_{s'\sim \mathbb{P}\_h(s'\mid s,a)} \langle Q_{h+1}(s',\cdot), \pi_{h+1}(\cdot \mid s')\rangle_{\mathcal{A}}.$$
> > >
> > >
> > > Moreover, in recent years, there is one work ([1]) to study the model-free RL problem in the general-sum MG under the general function approximation. However, their technique can **only be used to learn CCE** and cannot be easily extended to NE/CE. The main advantage of our paper is to propose **a unified treatment of (a) both model-based and model-free RL problems, and (b) learning NE/CE/CCE** from the optimization perspective. Due to the different decentralized function approximation style, the sample complexity results of our paper and [1] is hard to compare.
> > >
> > > For the model-based RL problem, recent works ([2,3]) convey the sample complexity results under the general function approximation. However, except for some specific examples like linear MG, **they do not induce direct sample complexity results on more general complexity measures like multi-agent witness rank** in our paper. So it is unclear how to compare the sample complexity between [2,3] and our results under a more general complexity measure. Compared to them, our main advantage is to provide the sample complexity results for the model-free RL problems.
> > >
> > > *References:*
> > >
> > > [1]. Wang et al.  Breaking the curse of multiagency: Provably efficient decentralized multi-agent rl with function approximation. preprint 2023.
> > >
> > > [2]. Chen et al. Unified algorithms for rl with decision-estimation coefficients: No-regret, pac, and reward-free learning. preprint 2022.
> > >
> > > [3]. Foster et al. On the complexity of multi-agent decision making: From learning in games to partial monitoring. COLT 2023.
> > >
> > > [4]. Huang et al. Towards general function approximation in zero-sum markov games. preprint 2021.
> > >
> > > [5]. Xiong et al. A self-play posterior sampling algorithm for zero-sum Markov games. ICML 2022.

---

### Official Review · Reviewer_JQUp · 2023-10-30

**Soundness:** 3 good
**Presentation:** 3 good
**Contribution:** 2 fair
**Rating:** 6
**Confidence:** 4

**Summary:**

This paper extends the decoupling coefficient from the single agent setting to the multi-agent setting for general-sum Markov games, termed MADC. The authors then propose an extension of the Maximize-to-Explore family of algorithms from the single agent setting to general-sum Markov games to solve for a variety of equilibrium concepts. By doing so, they construct an approach that relies only on solving an optimization problem and computing the equilibrium of a normal-form game. The authors prove their approach achieves sublinear regret (i.e., exploitability of the equilibrium) where the MADC appears as a constant.

**Strengths:**

The authors propose a general approach for learning a variety of equilibrium concepts across Markov games. They also define the multi-agent decoupling coefficient (MADC), and show how it relates theoretically to convergence rates to equilibria. All of this is done assuming the difficult setting with function approximation (either the transition kernel or the action-value functions are directly modelled).

**Weaknesses:**

I would like to see the authors compare / contrast their approach with PSRO [1, 2]. PSRO also consists of two components (single agent optimization to compute a best-response and computing the equilibrium of a normal-form game), applies to Markov games, leverages function approximation, and can learn CCE, CE, and NE.

[1] Lanctot, Marc, et al. "A unified game-theoretic approach to multiagent reinforcement learning." Advances in neural information processing systems 30 (2017).

[2] Marris, Luke, et al. "Multi-agent training beyond zero-sum with correlated equilibrium meta-solvers." International Conference on Machine Learning. PMLR, 2021.

**Questions:**

**Note I am increasing my score conditioned on some discussion of the limitations of scaling this approach to practical sized games (see my last follow-up comment) as well as including a discussion of a comparison with PSRO in related work**

- Pure Policy: I was confused by the definition of pure policies. You state that the set of pure policies is a subset of each agent's local policies, but this is still to vague. I assume a pure policy is a deterministic mapping from states to actions, but can you clarify / confirm this?
- Definition 2.4: It might help to state an intuitive definition in words as well, i.e., the $\delta$-covering number is the minimum number of balls of radius $\delta$ required to cover a set.
- Equation 3.1: Can you state somewhere that $\eta > 0$ is a hyperparameter that you are introducing?
- Can you please bring Algorithm 1 into the main body?
- You say "We prove that MAMEX achieves a sublinear regret for learning NE/CCE/CE in classes with small MADCs", but haven't you proven sublinear regret regardless of how large the MADC is (as long as it is finite, Assumption 2.7)? Do you have an interesting example in which MADC is infinite?
- Practicality: If you are going to consider all pure joint policies in a corresponding NFG (called meta-game in PSRO), why bother with function approximation of a value function? Why is it important to have the value function when you're already going to compute an equilibrium of this enormous game (assuming computing an equilibrium is more expensive than reading all the payoff entries)? Why not just deploy the equilibrium policy and be done rather than continue to iterate to learn the best approximation to the value function? In other words, if I have an oracle to solve an NFG of the size you suggest, I can just ask it to return the equilibrium policy assuming the entries in the payoff tensor are the *actual* Monte-Carlo returns of the pure-policies in the Markov game. Is there some setting you have in mind where it is cheaper to approximate a value function and avoid calculating returns for every joint pure policy? Sorry if I'm missing something, but it feels like something big is being swept under the rug here.

---

> ### Author Response · Authors · 2023-11-18
> **Response to the Reviewer JQUp**
>
> Thanks for your response. We will address your questions below.
>
> > 1. Comparison between MAMEX and PSRO
>
> The PSRO learns the equilibrium in the following way: At first, every player chooses a uniform policy as their strategy. The algorithm then calculates the equilibrium by a meta-solver, and trains an oracle that outputs the best response $\pi_i$ of the equilibrium for player $i$. After that, the algorithm adds $\pi_i$ into the strategy space of the player $i$. Last, the algorithm simulates all the new joint policy and construct a  new normal-form game for the next iteration. The algorithm can finally converge to an equilibrium.   However, in each iteration, it should simulate all the new joint policies and estimate the return. Consequently, the sample complexity increases exponentially as the iteration rounds increase.
>
>
> Different from PSRO, MAMEX utilizes the function approximation technique to the value function. The precise characterization of the structure of the value function can help us to evaluate the policy without actually simulating the environment with more samples. To be more specific, at each round, instead of simulating the environment and getting a Monte-Carlo return of each joint policy, MAMEX only needs to solve a regularized optimization problem over the function space of the value function. The solution of the optimization problem is used to be a payoff for the normal-form game. Since solving this optimization subproblem does not need additional samples, MAMEX bypasses the requirement for exponential samples to simulate the environment and estimate the value for each joint policy $\pi$. This characteristic enhances its sample efficiency in comparison to PSRO.
>
>
>
>
>
> > 2. Pure Policy set:
>
> The pure policy $\Pi_i$ for each player $i$ is a concrete policy set, so an agent can follow any mixture of the pure policy $\Delta(\Pi_i)$. The equilibrium is defined over the mixture of these pure policies. That is, each pure policy can be viewed as a pure strategy of the normal form game, and each joint policy can be viewed as a mixed strategy.
>
> For example, if the pure policy set is the deterministic policy set, the agent can execute any mixture of the deterministic policy. The equilibrium is then defined on the mixture of the deterministic policy, which reduces to the standard equilibrium defined in a normal-form game.
>
> If the pure policy set for each player only consists of the log linear policy $\Pi_i = \\{\pi_\theta: \pi_\theta(\cdot \mid s) = \text{Softmax}(\theta^T\phi(s,\cdot)), \theta \in \mathbb{R}^d, \|\theta\|_2 \le 1\\}$, then one agent can only execute the mixture of the log linear policy. Now the equilibrium is defined in a $|\Pi_1| \times |\Pi_2| \times \cdot \times |\Pi_n|$ multi-player normal-form game, where each log linear policy in $\Pi_i$ is a strategy for player $i$, and each joint log linear policy has a payoff.
>
> > 3. Can you state somewhere that $\eta > 0$ is a hyperparameter that you are introducing?
>
> In Theorem 3.1, we choose $\eta = 4/\sqrt{K}$.
>
> > 4. Can you please bring Algorithm 1 into the main body?
>
> Thanks for your reminder. We will put Algorithm 1 in the main body in our revised version.
>
> > 5. You say "We prove that MAMEX achieves a sublinear regret for learning NE/CCE/CE in classes with small MADCs", but haven't you proven sublinear regret regardless of how large the MADC is (as long as it is finite, Assumption 2.7)? Do you have an interesting example in which MADC is infinite?
>
>
> Thanks for your reminder. It is better to say "We prove that MAMEX achieves a sublinear regret for learning NE/CCE/CE, and the regret is smaller when the MADC of the function class is smaller."  In other words, the bound becomes vacuous when MADC is infinity.
>
> There is no nontrivial example in which MADC is infinite since MADC in tabular MG is also finite ($O(|S||A|)$).  Of course, an MG with infinite states and no structure has infinite MADC.

---

> ### Author Response · Authors · 2023-11-18
> **Response to the Reviewer JQUp**
>
> > 6.  Why bother with the function approximation of a value function? Why not just deploy the equilibrium policy and be done rather than continue to iterate to learn the best approximation to the value function? In other words, if I have an oracle to solve an NFG of the size you suggest, I can just ask it to return the equilibrium policy assuming the entries in the payoff tensor are the actual Monte-Carlo returns of the pure policies in the Markov game.
>
>  Note that the reward functions and transition kernels of the MG are unknown. If we estimate the payoff tensor using Monte-Carlo returns, we need to learn that for **all joint policies**. In other words, we have to enumerate every joint policy in $\Pi$ and collect Monte Carlo samples. To construct accurate estimators, the sample complexity is proportional to $|\Pi|$, which is usually exponential in non-parameterized case $(O(|A|^{|S|}))$ , making the algorithm sample-inefficient.
>
>
>
>
> In contrast, our algorithm uses function approximation on the value function to derive $V_{f}^\pi$ without estimating the function for all policies, thus leading to a sample complexity that only has the term $O(\log(|\Pi|))$. Such a difference also shows why exploration is necessary. By estimating the value functions smartly, balancing exploration and exploitation, we ensure that the size of $|\Pi|$ only through $\log|\Pi|$.
>
> To avoid estimating the payoff function for all joint pure policy, RSRO try to recursively add the policies into the policy space, and compute the payoff function of the new policy in each iteration. However, they do not provide a theoretical guarantee of the iteration rounds, the final sample complexity and the accuracy of the output.
>
> > 7. Is there some setting you have in mind where it is cheaper to approximate a value function and avoid calculating returns for every joint pure policy?
>
> MAMEX approximates the value function, calculates the payoff function and derives the equilibrium at Lines 3 and 4. If we regard Lines 3 and 4 in MAMEX as an oracle, MAMEX is oracle efficient. For CCE, the oracle can be approximately implemented by mirror descent [1], where the time of calculating $V(\pi)$ depends on $L$ and $L$ is the number of iterations. Hence, the computational complexity of this oracle depends only on the number of iterations rather than the joint pure policy space $|\Pi|$, and MAMEX can be calculated efficiently.
>
> *References:*
>
> [1]. Wang et al.  Breaking the curse of multiagency: Provably efficient decentralized multi-agent rl with function approximation. preprint 2023.

---

> > ### Comment · Reviewer_JQUp · 2023-11-20
> > **Follow-Up**
> >
> > Thank you for your response. I think I have a better understanding of your work now. Overall, I think I see this now as a pure theory paper, not quite ready for practical applications (which is OKAY, I'm not saying this is a requirement for acceptance). The fact that the paper focuses on "SAMPLE-EFFICIENT MULTI-AGENT RL" led me to try to look for some practical algorithm one could apply. That being said, I do think it's important to describe a roadmap towards practical application. Maybe one route forward to making this practical would be to supply an algorithm for learning an efficient $1/K$-cover (with representational guarantees) as you mentioned to pi75.
> >
> > Re PSRO, I understand your point that PSRO typically approximates entries in the NFG payoff tensor via simulation of joint policies. Your proposed MAMEX approach uses simulated joint policies to learn a value function which can then be used to predict entries in the NFG payoff tensor (possibly with fewer samples). That being said, PSRO begins with a subgame NFG of tractable size. I realize Double-Oracle/PSRO's convergence guarantees are vacuous (you need to build the entire NFG to guarantee convergence). But from a practical perspective, even if you were given the entire NFG for free (no samples needed), this NFG is so large, no oracle could return a solution in practical amount of time. Also, given your abstract "our algorithm only requires an equilibrium-solving oracle and an oracle that solves regularized supervised learning" which PSRO also satisfies, I feel you must cite PSRO (and possibly EGTA by Welling et al [1]) and compare/contrast with it somewhere. I would appreciate some discussion of these practical considerations and limitations of your approach in the paper. And what future research would you propose to make this approach practical?
> >
> > Also, I read your response above to reviewer pi75 which helped clarify the cardinality of the policy-space to me. In the naive case, each player's pure action space is of size $\vert \Pi_i \vert = \vert A_i \vert^{S}$ where I assume $S$ is the number of steps in the game? If you use a $1/K$-cover of the policy space, you can maintain a finite pure action space, which allows you to formulate an NFG. Note your description of the log linear policy above is defined over a continuous convex set, e.g., $\vert \theta \vert_2 \le 1$, and does not directly admit an NFG game. In general, I found the discussion of the cardinality of the action space in the paper to be lacking. Note that for most (IMO) Markov games, these action spaces will be so large that any approach that considers the NFG (even with a $1/K$-cover policy space) will be intractable.
> >
> > [1] Michael P. Wellman. 2006. Methods for Empirical Game-Theoretic Analysis. In Proceedings, The Twenty-First National Conference on Artificial Intelligence and the Eighteenth Innovative Applications of Artificial Intelligence Conference, July 16-20, 2006, Boston, Massachusetts, USA. 1552–1556.

---

> > > ### Author Response · Authors · 2023-11-21
> > > **Response to the Follow-Up**
> > >
> > > >1. A roadmap to the practical application/practical considerations and limitations/what future research would you propose to make this approach practical
> > >
> > > A possible avenue to the practical application is to let each agent use the PPO ([3]) to update the policy, and use regularized single-objective optimization to estimate $\overline{V}\_i(\pi)$ to update the value estimate. To be more specific,  for the model-based RL problem, we can assume the policy $\pi_\theta$ and the transition kernel $f$ are both deep neural networks. Then, the value update can utilize the techniques in [1], which maximizes the log-likelihood of the model with a value function bias as
> > >
> > > $$\max_\theta \max_\pi\mathbb{E}[\log \mathbb{P}\_f(s',r\mid s,a)] + \eta' \cdot \mathbb{E}\_{s \sim \rho}V_{f}^{\pi}(s), $$ where $\rho$ is the initial distribution.
> > > For the model-free RL problem, we can parameterize the state-action value function $Q_\theta^{(i)}(s,a)$ and minimize the standard TD error with a value function maximization bias as
> > > $$ \min_\theta \max_\pi \mathbb{E}[(r^{(i)}(s,a)+\gamma Q_\theta^{(i)}(s',a') - Q_\theta^{(i)}(s,a))^2] - \eta'\mathbb{E}\_{s,a\sim \pi(\cdot \mid s)}[Q_\theta^{(i)}(s,a)]. $$
> > >
> > > For the policy update, each agent can run a PPO algorithm to derive a CCE, as long as the gradient of the policy is approachable. Then the output joint policy will form an approximated CCE ([2]). This is because PPO is an adversarial no-regret algorithm with respect to adaptive adversaries, and if each agent follows a no-regret algorithm with respect to adaptive adversaries, the final output policy can converge to a CCE ([2]).
> > >
> > > However, for Nash Equilibrium, calculating the equilibrium is PPAD-complete. So MAMEX is computationally inefficient for learning NE.
> > >
> > > >2. Comparison with PSRO and EGTA
> > >
> > >  The comparisons of EGTA are shown below. We have added the comparisons of PSRO and EGTA in the related work section in Appendix A.
> > >
> > > The EGTA uses a graph to represent the deviation for all profiles, and then identify the Nash equilibrium and whether a  profile is relatively stable. Note if we want to evaluate all profiles and construct the whole graph,  we need many samples to estimate the payoff functions for each profile, and then identify all the deviations and construct the graph.
> > >
> > > The authors also mention that if one cannot evaluate all the profiles because of the lack of samples, one possible approach is to apply machine learning or regression techniques to fit a payoff function over the entire profile space given the available data. In fact, we use the function approximation technique to derive an estimate of all the profiles without estimating all the profiles. Thus, our approach can be considered as a learning approach to evaluate all the profiles without simulating environments using many samples.
> > >
> > > >3. Discussion of the cardinality of the action space in the paper is lacking
> > >
> > > For the normal case, at one step $h$, the cardinality of the policy space for the agent $i$, is $|A_i|^{|S|}$, where $A_i$ is the number of actions of the agent $i$, and the $S$ is the number of states. Hence, the cardinality of the whole policy space for the agent $i$ should be $O(|A_i| ^{|S|H})$ (The original response lacks a term $H$). If we choose the $\varepsilon$-cover of the parameterized policy class as
> > >
> > > $\Pi_i^{\text{pur}} = \\{\pi_\theta: \pi_{h,\theta}(\cdot \mid s) = \text{Softmax}(\langle \theta, \phi_{h,i}(s,a)\rangle), ||\theta||\_2 \le 1, ||\phi_{h,i}(s,a)|| \le 1, \theta \in \mathbb{R}^d\\}$,
> > >
> > >  the final cardinality should be $(1/\varepsilon)^{d}$. (The $\varepsilon$-cover of the parameterized policy class can be directly selected by the discretization of the $\theta$.) The sample complexity of MAMEX is still polynomial, as its upper bound only involves $\log|\Pi|$. In the normal case, the induced NFG is large since the cardinality of the policy is large, so we can try to use a no-regret algorithm to derive the CCE instead of solving the large NFG.
> > >
> > >
> > >  *References:*
> > >
> > >  [1]. Liu et al. Maximize to Explore: One Objective Function Fusing Estimation, Planning, and Exploration, NeurIPS 2023.
> > >
> > >  [2]. N. Cesa-Bianchi and G. Lugosi. Prediction, Learning, and Games. Cambridge University Press, 2006.
> > >
> > >  [3]. Cai et al. Provably efficient exploration in policy optimization. ICML 2020.

---

### Official Review · Reviewer_cnZv · 2023-11-05

**Soundness:** 3 good
**Presentation:** 3 good
**Contribution:** 3 good
**Rating:** 6
**Confidence:** 4

**Summary:**

This work studies online learning in general-sum Markov games with general function approximation, and focuses on the centralized learning paradigm. They extend the regret decoupling idea that has been used for the single-player case, and give a new notion, called multi-agent decoupling coefficient (MADC) to quantify the sample complexity. They propose a (conceptually) simple centralized algorithm called multi-agent maximize-to-explore (MAMEX) and show that it is able to achieve a regret upper bound related to MADC.

**Strengths:**

- The work provides a very general and conceptually simple algorithm to deal with centralized multi-agent reinforcement learning. It nicely extends the prior work on the single-player case.
- The writing is quite good.

**Weaknesses:**

- To me, there lacks motivation to study equilibrium learning in a centralized manner, particularly when it does not consider any global value optimization.  Equilibrium seems to be a concept under which selfish players cannot make unilateral move, and is usually used to characterize the steady state when every player plays independently and selfishly. However, if the players are in coordination, perhaps they can aim higher, such as higher social welfare. Can you give more motivations on centralized learning NE/CE/CCE without considering social welfare?  A related questions is: why is the Reg_{NE/CE/CCE} a meaningful measure for the quality of a centralized MARL algorithm?
- There are some previous works also on quantifying sample complexity of multi-agent RL, including Wang et al. 2023, Foster et al., 2023, Chen et al. 2022b. In this work, the comparisons to these previous works are all about the "simplicity" of the objective. The comparisons on "sample complexity", however, are missing.  How to compare the new notion MADC with the complexity measure proposed in these works? Do they have any relations?

**Questions:**

See the weakness part.

---

> ### Author Response · Authors · 2023-11-18
> **Response to the Reviewer cnZv**
>
> Thanks for your response. We will address your questions below.
>
> > 1. Motivation to study equilibrium learning in a centralized manner
>
> Centralized or decentralized learning is a separate question from whether the game is cooperative or competitive. For example, in a noncooperative game, each agent might first run a centralized RL algorithm on a game environment to learn the equilibrium policy $\hat \pi$, and then deploy her component $\hat \pi^{(i)}$. Here this environment takes the joint policy as the input and outputs the trajectory. Therefore, it is crucial to investigate the equilibrium for a multi-agent RL problem, which has been also studied in recent years ([1,2,3,4,6,7]).
>
> To be more specific, there are three motivations for learning the equilibrium in a centralized manner:
>
> * AlphaZero ([5]): It learns a Nash equilibrium in a two-player zero-sum Markov Game to learn how to compete with the opponent. The **centralized** RL algorithm is deployed to the two-player zero-sum Markov game and a policy $\pi^{(1)}$ and $\pi^{(2)}$ are learned. After the algorithm converges, the policy $\pi^{(1)}$ is deployed to play against human.
>
> * Financial Market: When $n$ market participants choose the best strategies based on the equilibrium, the market may reach a relatively stable state. So they have enough motivation to learn an equilibrium $(\pi^{(1)},\cdots,\pi^{(n)})$ by following a centralized algorithm, and the $i$-th market participants take the policy $\pi^{(i)}$.
>
> Furthermore, if the goal of the agents is to achieve social welfare, our algorithm can be modified to learn the policy with optimal social welfare. In particular, we only need to change the equilibrium-solving oracle to an oracle that maximizes the social welfare
>
> > 2. comparisons in the "sample complexity" side and relations between several complexity measures
>
> * Compared to the previous works that study the general function approximation of the general-sum MGs ([1,2,3]).
>
> [2] and [3] both study the model-based RL problem with centralized function approximation. Since [2] only studies the PAC learning setting with sample complexity results instead of the regret result,  to compare with them, we use our sample complexity result in Corollary D.4 to give a comparison. To learn a $\varepsilon$-NE/CCE/CE, both our results and their results have the form $O(\text{poly}(d_{\text{parameter}},\log |\Pi|, \log |M|)/\varepsilon^2)$, where $M$ is the model and $d_{\text{parameter}}$ is the complexity measure (MADC in our paper, MADEC in [2,3]). When applying to the tabular MG, both our work and their works suffer an exponential term $\prod_{i=1}^n A_i$, due to the centralized function approximation. When applying to the linear MG with dimension $d$,  the paper [2] achieves a $\widetilde{O}(dH^2\log |M|/\varepsilon^2)$ sample complexity, the paper [3] achieves a $\widetilde{O}(n^2d^2H^4\max_{i=1}^n \log |\Pi_i|/\varepsilon^2)$ sample complexity, while our paper can achieve a $\widetilde{O}(d^2H^2 + n^2H^2(\max_{i=1}^n\log |\Pi_i|)^2/\varepsilon^2)$ sample complexity. Compared to [2], the term $\log|M| \ge\Omega( A)$ (using an appropriate discretization ([8])) usually suffers the curse of multi-agency, while our result avoids this term and does not suffer the curse of multi-agency. Compared to [3],  our result has a higher degree in term $\max_{i=1}^n \log|\Pi_i|$. When $\max_{i=1}^n \log |\Pi_i|$ is small and can be ignored, our $\widetilde{O}((n^2+d^2)H^2/\varepsilon^2)$ sample complexity is smaller than the  result $\widetilde{O}(n^2d^2H^2/\varepsilon^2)$ in [3].
>
> When applying to the linear mixture MG, the work [2] derives a $\widetilde{O}(d^2H^3/\varepsilon^2)$ sample complexity, our paper achieves a smaller $\widetilde{O}((n^2+d^2)H^2/\varepsilon^2)$ sample complexity, and [3] does not derive an accurate result.
>
> Moreover, both [2] and [3] cannot be easily extended to the model-free settings since (a) their algorithms involve an optimization subproblem (Line 4 of Algorithm 9 in [2], Line 5 of Algorithm 1 in [3]) or a sampling procedure (Line 6 of Algorithm 9 in [2]) over the model space, and (b) their essential complexity measures (MDEC in [2], MA-DEC in [3]) heavily depends on the model. In contrast, our complexity measure MADC is flexible to be applied to both model-free RL and model-based RL.
>
> The recent work [1] uses a decentralized function approximation, so they provide a polynomial sample complexity result for learning CCE for the tabular general-sum MG. However, they do not provide a regret result, and the function approximation results are not directly comparable due to the different decentralized function approximation.
>
> * Relations
>
> We do not see a clear relation in general, in the sense that our complexity measures cannot be reduced to theirs and vice versa.   However, when applying to some specific examples, the regret of our work and previous works is comparable.   In addition,  our complexity measure also captures model-free settings.

---

> ### Author Response · Authors · 2023-11-18
> **Response to the Reviewer cnZv**
>
> *References:*
>
> [1]. Wang et al.  Breaking the curse of multiagency: Provably efficient decentralized multi-agent rl with function approximation. preprint 2023.
>
> [2]. Chen et al. Unified algorithms for rl with decision-estimation coefficients: No-regret, pac, and reward-free learning. preprint 2022.
>
> [3]. Foster et al. On the complexity of multi-agent decision making: From learning in games to partial monitoring. COLT 2023.
>
> [4]. Jiang et al. Offline congestion games: How feedback type affects data coverage requirement, ICLR 2023.
>
> [5]. Sliver et al. Mastering the game of Go with deep neural networks and tree search. Nature 2016.
>
> [6]. Liu et al. Sample-efficient reinforcement learning of partially observable markov games. NeurIPS 2022.
>
> [7]. Jin et al. V-Learning--A Simple, Efficient, Decentralized Algorithm for Multiagent RL. ICLR Workshop 2022.
>
> [8]. Liu et al. When Is Partially Observable Reinforcement Learning Not Scary? COLT 2022.

---

> > ### Comment · Reviewer_cnZv · 2023-11-22
> >
> > Thanks for answering my questions. Please include a complete comparison with [1,2,3] (as that in your response) in the paper.

---

### Official Review · Reviewer_pi75 · 2023-11-06

**Soundness:** 3 good
**Presentation:** 3 good
**Contribution:** 3 good
**Rating:** 6
**Confidence:** 3

**Summary:**

The paper discusses the challenges and solutions in the context of Multi-Agent Reinforcement Learning (MARL) for general-sum Markov Games (MGs) with a focus on efficient learning of equilibrium concepts such as Nash equilibrium (NE), correlated equilibrium (CE), and coarse correlated equilibrium (CCE). The authors address the complexities of using function approximation in general-sum MGs, which is different from single-agent RL or two-agent zero-sum MGs.

The paper introduces a unified algorithmic framework called Multi-Agent Maximize-to-EXplore (MAMEX) for general-sum MGs with general function approximation. MAMEX extends the Maximize-to-Explore (MEX) framework to MGs by optimizing over the policy space, and updating the joint policy of all agents to achieve a desired equilibrium. It combines an equilibrium solver for normal-form games defined over the policy space and a regularized optimization problem for payoff functions over the hypothesis space.

The paper defines a complexity measure called the Multi-Agent Decoupling Coefficient (MADC) to capture the exploration-exploitation tradeoff in MARL. MAMEX is shown to be sample-efficient in finding NE/CCE/CE in MGs with small MADCs, covering a wide range of MG instances, including those with low Bellman eluder dimensions, bilinear classes, and low witness ranks.

The paper's contributions include the development of MAMEX, a unified algorithmic framework for model-free and model-based MARL, and the introduction of the MADC complexity measure to quantify the exploration difficulty in MGs with function approximation.

The paper aims to provide a foundation for addressing the challenges of efficient MARL in general-sum MGs with a unified approach, covering both model-free and model-based methods.

**Strengths:**

- The paper is generally clear.

- The authors provide a novel efficient framework to compute CCE, CE, and NE in general-sum MGs.

- The authors propose a novel interesting complexity measure MADC, which captures the exploration-exploitation tradeoff for general-sum MGs.

- The final regret of the algorithm depends on the introduced MADC measure.

**Weaknesses:**

- The main weakness of the paper is how to perform the policy evaluation step. It seems to me that it will be computationally expensive to construct an estimator for each pure strategy for each player. Can the authors explain this more in detail? How big is the policy space of the pure strategy? If we are replacing the set with a 1/K-cover how much are we losing?

- There are some typos in the paper (e.g. page 8 solveing )

- It is not easy to understand the paper without looking at the appendix. For example, it would be better to include the definition of $\{l^{i,s}\}_{i \in [n]}$ in the main paper.

- How do the authors compare the regret results with previous SotA algorithms?

**Questions:**

See weaknesses

---

> ### Author Response · Authors · 2023-11-18
> **Response to the Reviewer pi75**
>
> Thanks for your response. We will address your questions below.
> > 1. The main weakness of the paper is how to perform the policy evaluation step. It seems to me that it will be computationally expensive to construct an estimator for each pure strategy for each player. Can the authors explain this more in detail?
>
> The policy evaluation step refers to Lines 3 and 4 in MAMEX (Algorithm 1). If we regard these two lines as an oracle, MAMEX is oracle-efficient. The computational complexity of this oracle depends on the type of equilibrium. For CCE, the oracle can be approximately implemented by mirror descent [1], where the time of calculating $V(\pi)$ depends on $L$ and $L$ is the number of iterations. Hence, the computational complexity of this oracle depends only on the number of iterations rather than the joint pure policy space $|\Pi|$, and MAMEX can be calculated efficiently. For Nash Equilibrium, calculating the equilibrium is PPAD-complete. So MAMEX is computationally inefficient for learning NE.
>
> > 2. How big is the policy space of the pure strategy?
>
> It depends on the selection of the pure strategy. If we consider all deterministic policies, the cardinality of the policy space of player $i$ is $|\Pi_i| = |A_i|^{|S|}$. Moreover, if we use the parameterized policy class, the size of the policy space can be smaller. For example, if we consider a $1/K$-cover of the log linear policies, the policy space is $K^d$, where $d$ is the dimension of the log linear policy class.
>
> > 3. If we are replacing the set with a 1/K-cover how much are we losing?
>
> We take the Nash equilibrium in the example below.
> If we consider a $1/K$-cover $\widetilde{\Pi}\_i$ for the policy space $\Pi_i$, then for any $\pi' \in \widetilde{\Pi}\_i$,
> there exists a policy $\pi \in \Pi_i$ such that $||\pi(\cdot \mid s) - \pi'(\cdot \mid s)||\_1\le 1/K$.
> Then for an policy $\Gamma$ in $\otimes_{i=1}^n \widetilde{\Pi}\_i$, then for each player,  $i$, the regret
> $$\max_{\pi_i \in \Delta(\Pi_i)}V_{1,i}^{\pi_i\times \Gamma_{-i}}(\rho) - V_{1,i}^{\Gamma}(\rho) \le V_{1,i}^{\widetilde{\pi}\_i\times \Gamma_{-i}}(\rho) - V_{1,i}^{\Gamma}(\rho) + O(1/K)\le \max_{\pi' \in \Delta(\widetilde{\Pi}\_i)}V_{1,i}^{\pi_i'\times \Gamma_{-i}}(\rho) - V_{1,i}^{\Gamma}(\rho).$$
> The inequality holds by denoting $\pi_i^* = \arg\max_{\pi \in \Delta(\pi_i)}V_{1,i}^{\pi_i'\times \Gamma_{-i}}(\rho)$, then there exists a $\widetilde{\pi}\_i \in \widetilde{\Pi}\_i$ such that
> $$V_{1,i}^{\pi_i^* \times \Gamma_{-i}}(\rho)  \le V_{1,i}^{\widetilde{\pi}\_i\times \Gamma_{-i}}(\rho) + O(1/K).$$
> Thus the difference between the regret defined on the original policy space and the regret defined on the $1/K$-cover is $O(1/K)*nK = O(n)$, which can be dominated by other terms.

---

> ### Author Response · Authors · 2023-11-18
> **Response to the Reviewer pi75**
>
> > 4. How do the authors compare the regret results with previous SotA algorithms?
>
> * Compared to the previous papers that study the general function approximation of the general-sum MGs ([1,2,3]).
>
> Two papers [2] and [3] both study the model-based RL problem with centralized function approximation. Since [2] only studies the PAC learning setting with sample complexity results instead of the regret result,  to compare with them, we use our sample complexity result in Corollary D.4 to give a comparison. To learn a $\varepsilon$-NE/CCE/CE, both our results and their results have the form $O(\text{poly}(d_{\text{parameter}},\log |\Pi|, \log |M|)/\varepsilon^2)$, where $M$ is the model and $d_{\text{parameter}}$ is the complexity measure (MADC in our paper, MADEC in [2,3]). When applying to the tabular MG, both our work and their works suffer an exponential term $\prod_{i=1}^n A_i$, due to the centralized function approximation. When applying to the linear MG with dimension $d$,  the paper [2] achieves a $\widetilde{O}(dH^2\log |M|/\varepsilon^2)$ sample complexity, the paper [3] achieves a $\widetilde{O}(n^2d^2H^4\max_{i=1}^n \log |\Pi_i|/\varepsilon^2)$ sample complexity, while our paper can achieve a $\widetilde{O}(d^2H^2 + n^2H^2(\max_{i=1}^n\log |\Pi_i|)^2/\varepsilon^2)$ sample complexity. Compared to [2], the term $\log|M| \ge\Omega( A)$ (using an appropriate discretization ([8])) usually suffers the curse of multi-agency, while our result avoids this term and does not suffer the curse of multi-agency. Compared to [3],  our result has a higher degree in term $\max_{i=1}^n \log|\Pi_i|$. When $\max_{i=1}^n \log |\Pi_i|$ is relatively small and can be ignored, our $\widetilde{O}((n^2+d^2)H^2/\varepsilon^2)$ sample complexity is smaller than the  result $\widetilde{O}(n^2d^2H^2/\varepsilon^2)$ in [3].
>
> When applying to the linear mixture MG, the work [2] derives a $\widetilde{O}(d^2H^3/\varepsilon^2)$ sample complexity, our paper achieves a smaller $\widetilde{O}((n^2+d^2)H^2/\varepsilon^2)$ sample complexity, and [3] does not derive an accurate result.
>
> Moreover, both [2] and [3] cannot be easily extended to the model-free settings since (a) their algorithms involve an optimization subproblem (Line 4 of Algorithm 9 in [2], Line 5 of Algorithm 1 in [3]) or a sampling procedure (Line 6 of Algorithm 9 in [2]) over the model space, and (b) their essential complexity measures (MDEC in [2], MA-DEC in [3]) heavily depends on the model. In contrast, our complexity measure MADC is flexible to be applied to both model-free RL and model-based RL.
>
>
>
> * Compared to the previous SoTA algorithm for specific examples.
>
> For zero-sum linear MG, the previous work [4] provides a $\widetilde{O}(d^{3/2}H^2\sqrt{K})$ regret when $\sum_{h=1}^H r_h \in [0,H]$, while MAMEX achieves a $\widetilde{O}(dH^3\sqrt{K}+H^3\sqrt{K}\log |\Pi|)$ regret. When the pure policy class for each player is the log linear policy class with dimension $d_\pi$, MAMEX yields a $\widetilde{O}((d+nd_\pi)H^3\sqrt{K})$ regret.
>
> For zero-sum linear mixture MG, the previous work [5] provides a minimax-optimal $\widetilde{O}(dH\sqrt{K})$ regret, while MAMEX yields a $\widetilde{O}(dH^5\sqrt{K})$ regret. Thus our regret matches their results in terms of $d$ and $K$, with an additional factor $H^4$.
>
> > 5. typos and appendix
>
> Thanks for your reminder. We will fix them in our next version. If we have additional space, we will move the definition of $\ell$ and the algorithm code into the main body.
>
> *References:*
>
> [1]. Wang et al.  Breaking the curse of multiagency: Provably efficient decentralized multi-agent rl with function approximation. preprint 2023.
>
> [2]. Chen et al. Unified algorithms for rl with decision-estimation coefficients: No-regret, pac, and reward-free learning. preprint 2022.
>
> [3]. Foster et al. On the complexity of multi-agent decision making: From learning in games to partial monitoring. COLT 2023.
>
> [4]. Xie et al. Learning zero-sum simultaneousmove markov games using function approximation and correlated equilibrium. COLT 2020.
>
> [5]. Chen et al.  Almost optimal algorithms for two-player zerosum linear mixture markov games. ICALT 2022.

---

### Author Response · Authors · 2023-11-18
**Global Response**

Thanks for the detailed response of all reviewers. We have changed the pdf file to our revised version. Our revised version has been modified mainly in the following points. All the changes are highlighted in blue.

* We move the Algorithm 1 and the definition of discrepancy function $\ell^{(i),s}$ into the main body for clarity. Instead, we put the notation part and the definition of the covering number into the Appendix.

* We fix a small error in the proof of Collroary E.4.

---

### Meta-Review · Area_Chair_X7LP · 2023-12-04

**Metareview:**

The paper proposes a unified algorithmic framework that is sample efficient in terms of function approximation for learning Nash Equilibrium, Coarse Correlated Equilibrium, and Correlated Equilibrium, for both model-based and model-free MARL problems, assume that a new defined complexity measure called MADC is small. The idea generalizes a prior work by Liu et al that was done for single agent and two-player zero-sum games. The main challenge is that NE, CCE, CE are not unique for multi-agent settings as in the case of single-agent and two-agent zero-sum, so new ideas had to be developed. The reviewers agreed that the paper is worth publishing. However, a lot of things need to be addressed in the camera ready version of the paper. The oracle to compute NE is computationally intractable (PPAD-hard) so a remark must be added. Moreover, for the tabular case, due to centralized training, the method does not avoid the curse of multi-agent systems (seems to depend on the product of actions of all agents). Finally, even for CCE, CE, is not clear if the oracle is efficiently implementable as the number of possible actions per player in the induced normal form game is of order $|S|^{|A_i|}$. There has been some discussion that it is possible (during rebuttal), as step 3 of the algorithm is not needed (no need to compute all possible values of the induced normal form game). Please also add to your camera ready version the response to FdZN, as it is not clear if the result can have practical implications.

**Justification For Why Not Higher Score:**

The paper is based on an idea of a prior work with some extra ideas to include multi-agent settings.

**Justification For Why Not Lower Score:**

The score was unanimous among the reviewers. All agreed that the paper is worth publishing, though they were not super enthusiastic about it.

---

### Decision · Program_Chairs · 2024-01-16

Accept (poster)